# Cohesin and CTCF control the dynamics of chromosome folding

Pia Mach [1,2,9], Pavel I. Kos [1,9], Yinxiu Zhan [1,9], Julie Cramard [1], Simon Gaudin[1,3,4], Jana Tünnermann[1,2], Edoardo Marchi[5,6], Jan Eglinger [1], Jessica Zuin[1], Mariya Kryzhanovska[1], Sebastien Smallwood[1], Laurent Gelman[1], Gregory Roth[1], Elphège P. Nora [7,8], Guido Tiana [5,6] & Luca Giorgetti [1]✉

In mammals, interactions between sequences within topologically associating domains enable control of gene expression across large genomic distances. Yet it is unknown how frequently such contacts occur, how long they last and how they depend on the dynamics of chromosome folding and loop extrusion activity of cohesin. By imaging chromosomal locations at high spatial and temporal resolution in living cells, we show that interactions within topologically associating domains are transient and occur frequently during the course of a cell cycle. Interactions become more frequent and longer in the presence of convergent CTCF sites, resulting in suppression of variability in chromosome folding across time. Supported by physical models of chromosome dynamics, our data suggest that CTCF-anchored loops last around 10 min. Our results show that long-range transcriptional regulation might rely on transient physical proximity, and that cohesin and CTCF stabilize highly dynamic chromosome structures, facilitating selected subsets of chromosomal interactions.

In mammalian cells, interactions between chromosomal sequences play important roles in fundamental processes such as DNA replication[1], repair[2] and transcriptional regulation by distal enhancers[3]. Chromosome conformation capture (3C) methods, which measure physical proximity between genomic sequences in fixed cells, revealed that chromosomal contacts are organized into submegabase domains of preferential interactions known as topologically associating domains (TADs)[4,5] whose boundaries can functionally insulate regulatory sequences[3]. TADs mainly arise from nested interactions between convergently oriented binding sites of the DNA-binding protein CTCF, which are established as chromatin-bound CTCF arrests the loop extrusion activity of the cohesin complex[6–10].

Determining the timing and duration of chromosomal interactions within TADs and their relationship with CTCF and cohesin is key to understanding how enhancers communicate with promoters[11,12]. Single-cell analyses of chromosome structure in fixed cells[4,13–15], chromosome tracing experiments[16–19], in vitro[9,10,20] and live-cell[21] measurements of CTCF and cohesin dynamics, and polymer simulations[6,15,22], as well as live-cell imaging of chromosomal locations and nascent RNA[23,24], all suggested that TADs and CTCF loops are dynamic structures whose temporal evolution might be governed by the kinetics of loop extrusion[25]. Recent live-cell measurements of a CTCF loop connecting two opposite TAD boundaries in mouse embryonic stem cells (mESCs) provided direct evidence that this is the case, and revealed that cohesin-mediated loops between CTCF sites located 500 kilobases (kb) away last 10–30 min (ref. [26]). However, it is still unclear if contacts between sequences separated by genomic distances where enhancers and promoters interact within the same TAD occur on the timescale of seconds, minutes or hours. We also have little knowledge on whether and how rates and durations of such contacts are modulated by loop extrusion. We finally do not know if cohesin increases chromosome mobility and thus favors the encounters between genomic sequences

[1]Friedrich Miescher Institute for Biomedical Research, Basel, Switzerland. [2]University of Basel, Basel, Switzerland. [3]École Normale Supérieure de Lyon, Lyon, France. [4]Université Claude Bernard Lyon I, Lyon, France. [5]Università degli Studi di Milano, Milan, Italy. [6]INFN, Milan, Italy. [7]Cardiovascular Research Institute, University of California San Francisco, San Francisco, CA, USA. [8]Department of Biochemistry and Biophysics, University of California San Francisco, San Francisco, CA, USA. [9]These authors contributed equally: Pia Mach, Pavel I. Kos, Yinxiu Zhan. ✉e-mail: luca.giorgetti@fmi.ch

by reeling them into loops, or if instead it provides constraints that decrease mobility and prolong the duration of such encounters. Both scenarios have been suggested to be possible theoretically[27,28], but it is unclear which effect dominates in living cells.

Here we use live-cell fluorescence microscopy to measure chromosome dynamics and its dependence on cohesin and CTCF in mESCs. By combining two live-cell imaging strategies with polymer simulations, we reveal that loops extruded by cohesin constrain global chromosome motion, while also increasing the temporal frequencies and durations of physical encounters between sequences inside the same TAD. Convergent CTCF sites substantially stabilize contacts through cohesin-mediated CTCF-anchored loops that last around 5–15 min on average. Our results support the notion that chromosome structure within single TADs is highly dynamic during the span of a cell cycle and thus that long-range transcriptional regulation might rely on transient physical proximity between genomic sequences. They also reveal how contact dynamics and the temporal variability in chromosome folding are modulated by cohesin and CTCF in single living cells and provide a quantitative framework for understanding the role of folding dynamics in fundamental biological processes.

## Results

### Cohesin decreases chromosome mobility independently of CTCF

To study how cohesin and CTCF influence the global dynamics of the chromatin fiber independently of local chromatin state and structural differences, we examined the dynamic properties of large numbers of random genomic locations in living cells. We generated clonal mESC lines carrying multiple random integrations of an array of ~140 repeats of the bacterial Tet operator sequence (TetO) using piggyBac transposition[29]. These can be visualized upon binding of Tet repressor (TetR) fused to the red fluorescent protein tdTomato. To compare the motion of genomic locations that either block or allow the loop extrusion activity of cohesin, the TetO array was adjacent to three CTCF motifs (3 × CTCF) that could be removed by Cre-assisted recombination (Fig. 1a). Motifs were selected based on high CTCF enrichment in chromatin immunoprecipitation followed by sequencing (ChIP–seq) and each was confirmed to be bound by CTCF in nearly 100% of alleles at any time in mESCs using dual-enzyme single-molecule footprinting[30] (R. Grand and D. Schübeler, personal communication), thus providing a close experimental representative of an 'impermeable' loop extrusion barrier.

3 × CTCF-TetO sequences were introduced in mESCs that stably expressed *Os*Tir1 and where the endogenous *Rad21*, *Wapl* or *Ctcf* genes were targeted with an auxin-inducible degron (AID) peptide fused to eGFP[31,32]. This resulted in several mESC clones (three per degron condition) with different sets of genomic insertions of the 3 × CTCF-TetO

cassette, where over 95% of any of the AID-tagged proteins could be rapidly depleted upon addition of auxin (Fig. 1b and Extended Data Fig. 1a). This allowed us to study chromosome dynamics following acute depletion of factors affecting cohesin-mediated chromosome structure (Extended Data Fig. 1b) at previously reported time points (90 min for RAD21 (ref. [33]), 6 h for CTCF[31] and 24 h for WAPL[32]) that minimize secondary effects such as defects in cell-cycle progression (Extended Data Fig. 1c).

Mapping TetO insertion sites revealed 10–20 insertions per cell line, with on average 1–2 heterozygous insertions per chromosome without any strong bias towards active or inactive chromatin (Extended Data Fig. 1d,e). Insertions were on average 10 kb away from the nearest endogenous CTCF binding sites (Extended Data Fig. 1f). 4C sequencing (4C-seq) confirmed that insertion of 3 × CTCF-TetO cassettes often led to the formation of ectopic interactions with endogenous CTCF sites, which were lost upon removal of 3 × CTCF sites or depletion of RAD21 (Extended Data Fig. 1g,h).

To measure the dynamics of 3 × CTCF-TetO insertions, we acquired three-dimensional (3D) movies (one z-stack of 10 μm every 10 s for 30 min) using highly inclined and laminated optical sheet microscopy[34] (Fig. 1b and Supplementary Video 1). This resulted in ~270 cells per condition with over 8,000 trajectories from three clonal lines imaged with 3–4 biological replicates per condition. Detection and localization of TetO arrays as subdiffraction fluorescent signals[35] enabled reconstruction of trajectories of individual genomic insertions (Fig. 1c and Methods). We then studied their mean squared displacement (MSD) as a function of time after correcting each trajectory for the confounding effect of cell movement, which we inferred from the collective displacement of all insertions in each nucleus (Fig. 1d, Extended Data Fig. 2a and Methods). Independently of the degron background, in untreated cells, genomic locations underwent on average a subdiffusive motion whose anomalous exponent (~0.6) and generalized diffusion coefficients (D) (~$1.2 \times 10^{-2}$ μm² s⁻ᵅ) were in line with previous studies of specific genomic loci[36,37] (Fig. 1d and Extended Data Fig. 2b). The MSD of radial distances (radial MSD) between insertions within the same nuclei showed the same scaling although statistics were less robust for long time intervals due to the shorter trajectories that could be built based on pairwise distances (Fig. 1d). Interestingly, removal of 3 × CTCF sites (Extended Data Fig. 1g) or degradation of CTCF (6-h auxin treatment) did not have a significant impact on MSD averaged over all genomic locations nor on its distribution across trajectories and cells (Fig. 1e,f, results for single clones in Extended Data Fig. 2c, P values in Extended Data Fig. 2e).

By contrast, acute depletion of RAD21 (90-min auxin treatment) led to a significant increase in mobility both in the presence (Fig. 1g) and absence of 3 × CTCF sites (Extended Data Fig. 2d), with only a very minor impact on anomalous exponents (Extended Data Fig. 2b,e,

**Fig. 1 | Cohesin slows down chromosome dynamics in living cells. a**, Clonal mESC lines containing random TetO arrays flanked by 3 × CTCF motifs and expressing TetR-tdTomato. Constructs were integrated using piggyBac transposition in mESCs allowing auxin-inducible degradation of GFP-tagged RAD21, WAPL or CTCF. ITR, inverted terminal repeats. **b**, Representative images of RAD21-AID-eGFP cells containing 3 × CTCF-TetO imaged before or after 90 min of auxin treatment (exposure time eGFP and tdTomato: 50 ms, deconvolved, maximum intensity projection, bicubic interpolation, *n* = 3 replicates). **c**, Left, time series of TetR-tdTomato signal over 30 min (maximum intensity projection, time interval dt = 10 s, color-coded for intensity changes over time). Right, magnification with overlay of TetR-tdTomato signal with reconstructed trajectories of individual TetO arrays. **d**, Left, cell motion is approximated as the average roto-translational motion of TetO signals within the same nucleus. Right, MSD averaged over trajectories within one nucleus (mean ± s.e.m.) before (cyan, *n* = 77) and after (blue, *n* = 77) cell motion and localization error correction. Green, radial MSD of pairs of operator arrays within the same nucleus (mean ± s.e.m., *n* = 491 pairs). **e**, Left, MSD (mean ± s.e.m.) in mESC lines before

(blue, 310 cells, 13,537 trajectories) or after (red, 271 cells, 11,082 trajectories) Cre-mediated removal of 3 × CTCF sites. Three replicates per cell line and three lines per condition were analyzed and merged here and in all following MSD graphs. P values (two-sided Student's *t*-test) for all panels shown in Extended Data Fig. 2e. Right, schematic representation of Cre-mediated removal of CTCF sites. **f**, Left, same as in **e** but in mESC lines with 3 × CTCF-TetO arrays, before (blue, 323 cells, 9,829 trajectories) or after (red, 365 cells, 12,495 trajectories) CTCF degradation (6 h of auxin treatment). Right, schematic representation of auxin-induced CTCF degradation. **g**, MSD (mean ± s.e.m.) of 3 × CTCF-TetO insertions before (blue, 310 cells, 13,537 trajectories) or after (red, 240 cells, 8,788 trajectories) RAD21 degradation (90 min of auxin). **h**, MSD (mean ± s.e.m.) of 3 × CTCF-TetO before (blue, 336 cells, 6,687 trajectories) or after (red, 350 cells, 6,717 trajectories) WAPL degradation (24 h of auxin). **i**, Fold changes in generalized diffusion coefficients (D) and scaling exponents (α) in untreated cells compared with cells where degradation of CTCF, RAD21 and WAPL or removal of CTCF motifs (3 × CTCF) occurred.

*P* values in Extended Data Fig. 2e). In the presence of wild-type levels of RAD21, generalized diffusion coefficients were on average ~30% lower than in depleted cells, where RAD21 levels were low enough to prevent formation of cohesin-mediated structures (compare with Extended Data Fig. 1b). This outcome was consistent across three clonal cell lines with different TetO insertion sites and the small differences in the magnitude of the effect were likely due to location-dependent effects (Extended Data Fig. 2f). Importantly, the effect was specific for

RAD21 degradation as we did not observe any changes in MSD behavior in control cell lines expressing *Os*Tir1 but no AID-tag (Extended Data Fig. 2g). In addition, depletion of WAPL (24-h auxin treatment), which results in higher levels of DNA-bound cohesin[32], caused a substantial decrease in chromosome mobility (Fig. 1h and Extended Data Fig. 2e). Together, these results indicate that increasing levels of DNA-bound cohesin decrease chromosome mobility, with only very minor effects (if any) mediated by the presence of even strong CTCF motifs (Fig. 1i).

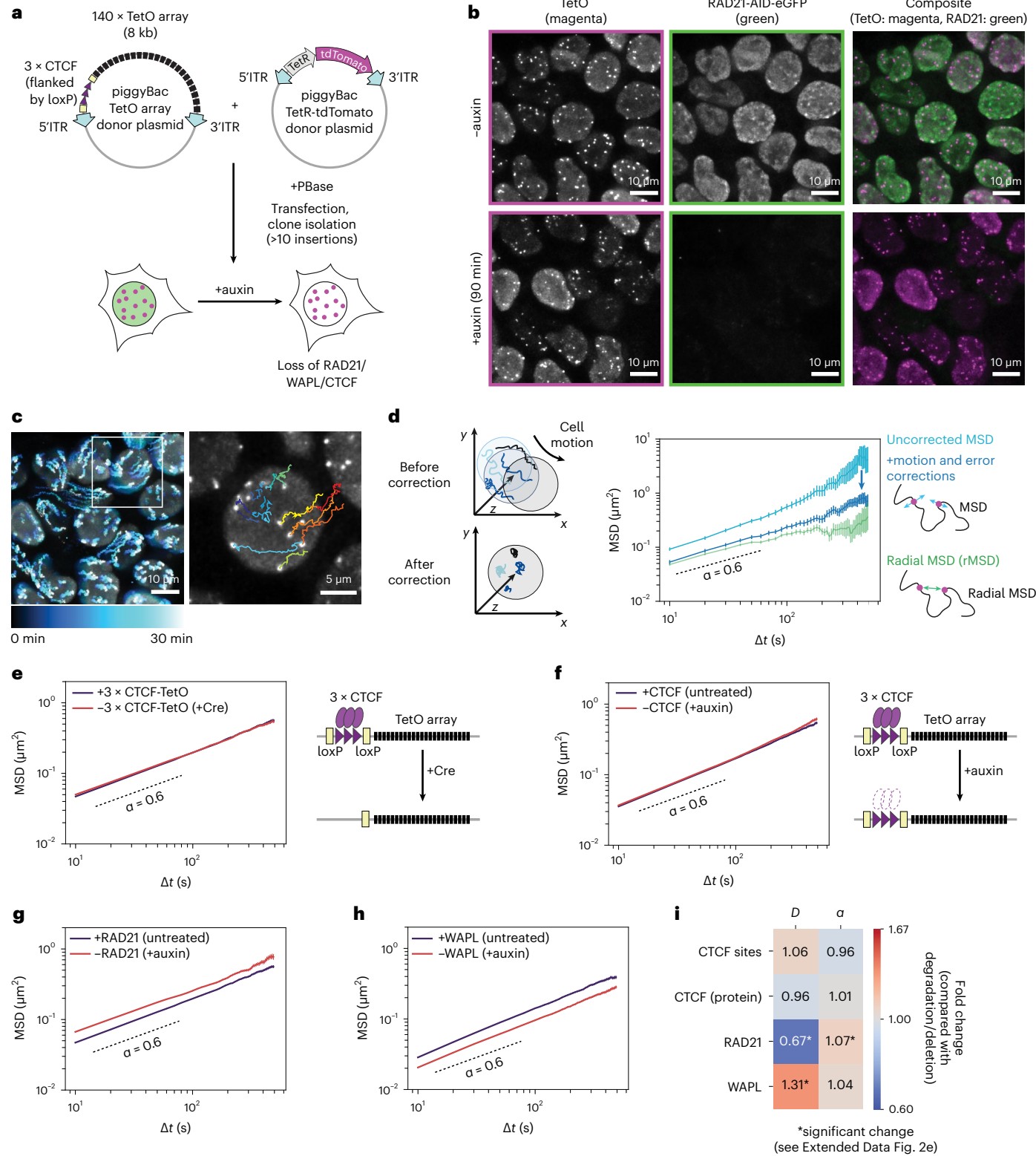

## Loop extrusion can explain reduced chromosome dynamics

We next used polymer simulations to determine if loop extrusion alone could explain the observed global reduction in chromosome dynamics in the presence of cohesin and minimal effects from CTCF. We simulated the dynamics of a polymer with excluded volume, with or without loop extrusion and extrusion barriers whose linear arrangement and orientation were sampled from endogenous CTCF sites (Fig. 2a and Extended Data Fig. 3a). To mimic random insertion of 3 × CTCF sites, we also simulated the same polymers with additional loop extrusion barriers separated by 800 kb which were inserted at random positions in the polymer (magnified area in Fig. 2a). To emphasize their potential effects on chromosome dynamics, all barriers in the simulations were impermeable to loop extruders. Every monomer represented 8 kb of chromatin, corresponding to the genomic size of the TetO array. Simulation steps were approximated to real-time units by matching the time needed for a monomer to move by its own diameter with the time required by the TetO array to move by its estimated mean physical size (Methods). We sampled an extremely large range of extruder residence times and loading rates (4 orders of magnitude each) centered around a residence time of ~30 min and extruder densities of ~20 per Mb (in line with previous measurements[38,39]), and using two extrusion speeds corresponding to in vivo and in vitro estimates (~0.1 kb s$^{-1}$ and ~1 kb s$^{-1}$, respectively)[20,38] (Fig. 2b and Extended Data Fig. 3b).

In the absence of loop extrusion, the polymer underwent subdiffusive behavior with anomalous exponent of ~0.6 (Fig. 2c), as expected from simple polymers with excluded volume[40–42] (see Supplementary Information) and compatible with our experimental results on random TetO insertions (Fig. 1e). Strikingly, in line with experimentally measured effects of RAD21 (Fig. 1g), introduction of loop extrusion led to lower generalized diffusion coefficients and minor effects on anomalous exponents, independently of loading rate and residence time (Fig. 2c,d), extrusion speed (Extended Data Fig. 3c) or the presence of extrusion barriers (Fig. 2e,f). Interestingly, for extruder residence times of 5.5–11 min and unloading rates corresponding to extruder linear densities of ~20 per Mb, the predicted decrease in generalized diffusion coefficients was in quantitative agreement with the experimentally observed value of ~30% (Fig. 2d,f; extruder densities as in Fig. 2b; compare with Fig. 1g). Also, consistently with WAPL depletion experiments (Fig. 1h), increasing extruder residence times systematically resulted in larger reductions in generalized diffusion coefficients (Fig. 2d,f and Extended Data Fig. 3c).

Importantly, addition of barriers in the presence of loop extrusion led to substantially smaller changes in polymer dynamics compared with the effect of loop extrusion itself even when probed directly on the barriers (Fig. 2g,h and Extended Data Figs. 3c and 4a,b), in agreement with our experimental finding that CTCF degradation had no strong effect on MSDs of TetO insertions (Fig. 1e,f and Extended Data Fig. 2c,e). Similarly, insertion of additional barriers had little impact on MSD (Fig. 2i,j), thus recapitulating the negligible effect of removal of 3 × CTCF sites (Fig. 1e). Polymer simulations thus strongly support the notion that the observed decrease in chromosome mobility and lack of effects from CTCF is a macroscopic manifestation of the physical constraints imposed by cohesin in living cells.

## Cohesin and CTCF constrain the dynamics of sequences in *cis*

We next asked how cohesin and CTCF impact the reciprocal motion of two genomic sequences located on the same DNA molecule. To this aim, we simulated the dynamics of a polymer carrying two convergent impermeable extrusion barriers mimicking strong CTCF motifs separated by ~150 kb (Fig. 3a). This is comparable to median distances between convergent CTCF sites within TADs genome-wide in mESCs (141 kb, Methods) and also to the estimated average separation between enhancers and promoters in human cells (~160 kb)[43]. Simulations performed with extrusion parameters recapitulating the dynamic effects of RAD21 depletion (black square in Fig. 2f) predicted that radial MSDs should be lowest in the presence of loop extrusion and barriers (Fig. 3b) due to the formation of transient loops anchored by the barriers (Fig. 3c). Similar to MSDs (Fig. 2), radial MSDs should increase upon removal of extrusion barriers and become maximal when loop extrusion is also removed (Fig. 3b).

Importantly, simulations also predicted that scaling exponents of radial MSD curves should be considerably smaller (~0.2) than those we previously observed for TetO arrays separated by several Mb or located on different chromosomes (~0.6, Fig. 1d). This is because correlations in the motion of two monomers are stronger when they are located closer along the polymer. Indeed, simulations predicted that scaling exponents fitted from radial MSD curves at short times should increase with increasing genomic distance and approach 0.6 for loci separated by several Mb (consistent with radial MSDs of randomly inserted TetO arrays) (Extended Data Fig. 5a) before saturating to stationary values at longer times. This holds true also without loop extrusion (theoretical analysis in Supplementary Information and simulations in Extended Data Fig. 5b).

To test these predictions, we turned to a live-cell imaging approach allowing us to measure the radial dynamics of two sequences located within the same TAD, in the presence and absence of cohesin and/or strong CTCF sites. We engineered mESCs carrying targeted integrations of two orthogonal operator arrays: ~140× TetO and 120× LacO separated by 150 kb (Fig. 4a), which could be visualized upon binding of TetR-tdTomato and a weak DNA-binding variant of LacI fused to eGFP (LacI**-eGFP)[44]. To minimize confounding effects from additional regulatory sequences such as active genes or enhancers, we targeted the arrays into a 560-kb 'neutral' TAD on chromosome 15 where we previously removed internal CTCF sites[3] (Fig. 4a). The two operator arrays were directly adjacent to excisable 3 × CTCF site cassettes arranged in a convergent orientation (Fig. 4a). Cell lines were verified by Nanopore Cas9-targeted sequencing (nCATS)[45] to contain a single copy of each targeting cassette (Extended Data Fig. 5c). We additionally targeted the endogenous *Rad21* locus with a C-terminal HaloTag-FKBP fusion allowing the inducible degradation of RAD21 upon treatment with dTAG-13 (ref. [46]) as confirmed by severely decreased protein levels (>95% after 2-h treatment, Extended Data Fig. 5d).

**Fig. 2 | Loop extrusion generally slows down polymer motion. a**, Representative snapshots of conformations and simulated contact maps for a polymer model with excluded volume and increasingly complex models with loop extruders, extrusion barriers sampled from CTCF motifs within 9 Mb on chromosome 15 (Chr15:7–16 Mb) and additional randomly distributed extrusion barriers. For the system with additional barriers, the contact map is presented aside with magnification of the contact map of the system without additional barriers to highlight the differences. **b**, Simulated contact maps (with loop extrusion and extrusion barriers) for polymers with two extrusion speeds (1 kb s$^{-1}$ and 0.1 kb s$^{-1}$) and different combinations of extruder loading rates and residence times. The resulting linear densities of extruders (number per Mb) are shown in the bottom left corner of each contact map. **c**, Effect of extruders. MSDs of polymers with (red line) or without (gray dashed line) loop extruders in the absence of extrusion barriers (loading rate 0.6 (Mb × min)$^{-1}$ and residence time 5.5 min, corresponds to black square in panel **d**). Black dashed curve represents $\alpha = 0.6$ as an eye guide. **d**, Effect of extruders. Ratios of generalized diffusion coefficients and anomalous exponents between the two conditions shown in panel **c**. Black square, set of parameters whose corresponding MSDs are shown in panel **c**. **e**, MSDs of polymers with (blue line) or without (gray dashed line) both extruders and barriers. Same parameters as in panel **c**. **f**, Same as panel **d** for cases illustrated in panel **e**. **g**, MSDs of polymers with loop extruders in the presence (blue) or absence (red) of extrusion barriers. Same parameters as in panels **c** and **e**. **h**, Same as panels **d** and **f** but for cases illustrated in panel **g**. **i**, MSDs of polymers either with (light blue) or without (red) additional randomly inserted extrusion barriers. Same parameters as in panels **c**, **e**, **g**. **j**, Same as panels **d**, **f** and **h** but for cases illustrated in panel **i**.

Capture-C with tiled oligonucleotides revealed that integration of operator arrays themselves did not lead to detectable changes in chromosome structure (Extended Data Fig. 5e). Convergent 3 × CTCF sites, however, led to the formation of a new CTCF-mediated interaction within the TAD (2.8× increase in contact probability after correcting the confounding contribution of the wild-type allele) (Fig. 4b), which

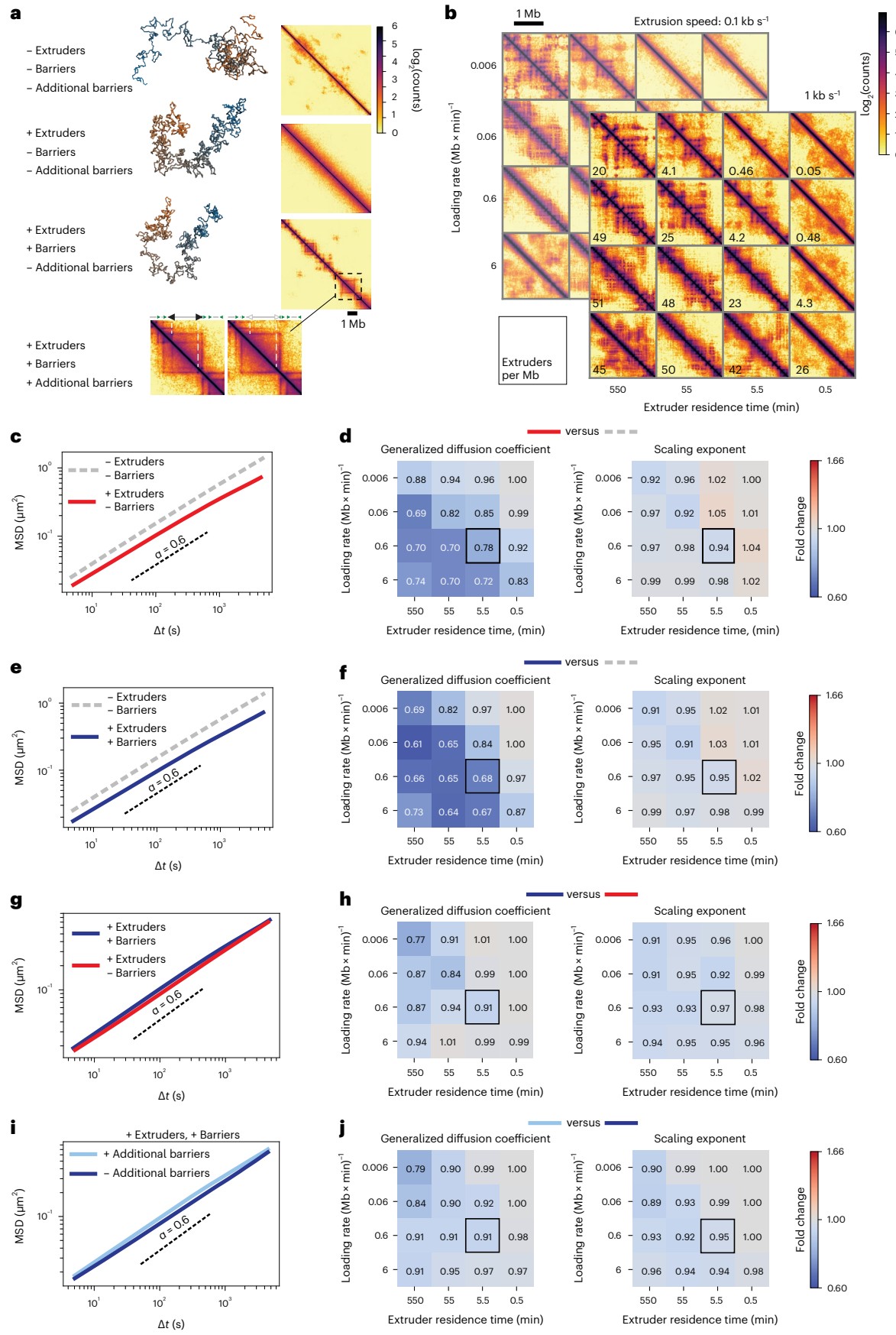

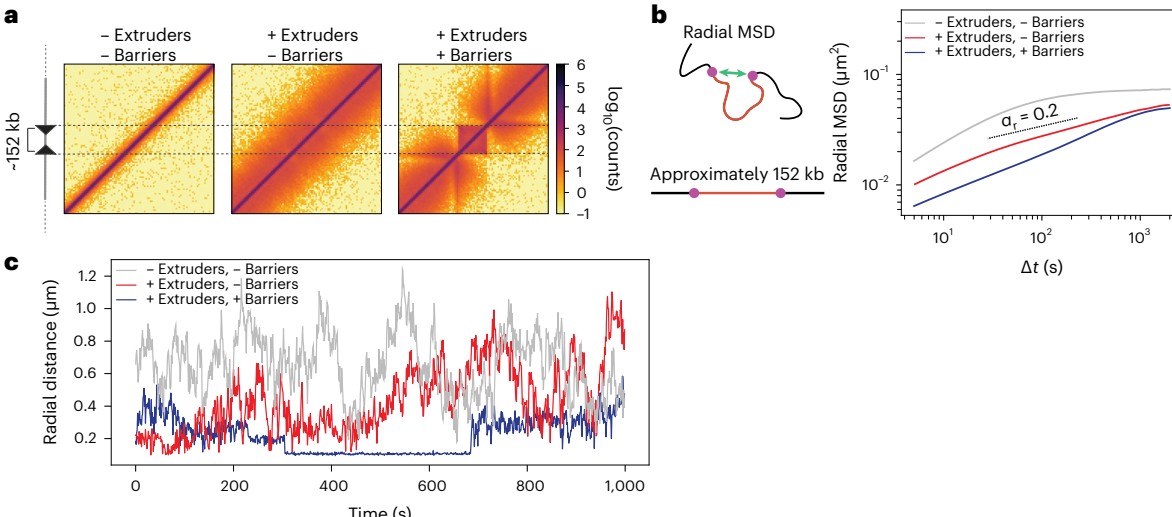

**Fig. 3 | Convergent CTCF sites further constrain polymer dynamics.**
**a**, Simulated contact maps of a region spanning the equivalent of 800 kb for a polymer chain without loop extrusion, with loop extruders and with convergent extrusion barriers separated by the equivalent of 152 kb. **b**, Radial MSD of the two monomers separated by the equivalent of 152 kb in the three conditions from panel **a**. Dashed line is an exponent of 0.2 as a guide to the eye ($\alpha_r$ indicates the slope of radial MSDs). Loop extrusion parameters as in Fig. 2c. **c**, Representative examples of distances between the two monomers in simulations with or without loop extrusion and extrusion barriers. The flat stretch in the trajectory with extrusion and barriers corresponds to a loop anchored by the two barriers.

was lost upon RAD21 depletion along with all other CTCF-mediated interactions across the locus (Extended Data Fig. 5f).

We imaged cells for 3 h every 30 s in three dimensions (Fig. 4c and Supplementary Video 2), either in the presence or absence of RAD21, and measured distances between the two arrays over time (Fig. 4d, $n$ = 3–7 biological replicates for each condition, on average 220 cells per condition, Supplementary Table 1 and Methods). Doublet signals corresponding to replicated alleles occurred in a very minor fraction (3%, Methods) of trajectories, compatible with the late-replication profile of the 'neutral' TAD and the cell-cycle distribution (Extended Data Figs. 5g and 6a). In these cases, only trajectories that were initially closest across channels were considered. After correction of chromatic aberrations (Methods and Extended Data Fig. 6b), we estimated our experimental uncertainty on radial distances to be ~130 nm by measuring pairwise distances in control cells where multiple TetO insertions were simultaneously bound by both TetR-tdTomato and TetR-eGFP (Extended Data Fig. 6c–e).

In agreement with model predictions for locations separated by 150 kb, radial MSDs of the two arrays showed scaling exponents close to 0.2, much smaller than those observed with randomly inserted TetO arrays (Extended Data Fig. 7a,b). Also in line with model predictions for

(Extended Data Fig. 5a), presence of RAD21 and 3 × CTCF sites led to the most constrained radial mobility, whereas RAD21 degradation and deletion of CTCF sites resulted in the least constrained motion (Extended Data Fig. 7c). These measurements thus verified the model prediction that genomic sequences located at short distances (150 kb) experience stronger physical constraints than sequences located at larger genomic distances[26] (Fig. 1 and Extended Data Fig. 7b), and that loop extrusion provides constraints that are further reinforced by convergent CTCF sites.

Consistently with their more constrained radial MSD behavior, we finally observed that distances between TetO and LacO signals were smallest in the presence of convergent CTCF sites and cohesin. In these conditions, distances between TetO and LacO arrays tended to remain close to the ~130-nm experimental uncertainty with only occasional fluctuations toward larger values in the course of the 3 h of imaging (Fig. 4d,e). Removal of 3 × CTCF sites led to increased radial distances and variability within single trajectories, which were further increased upon degradation of RAD21, irrespective of the presence or absence of CTCF sites (Fig. 4d,e and Supplementary Video 3). Thus, constraints imposed by extruding cohesin and convergent CTCF sites reduce not only average physical distances between sequences but also

**Fig. 4 | Cohesin and CTCF reduce variability in chromosome folding dynamics.** **a**, Top, insertion of TetO and LacO arrays separated by 150 kb within a 'neutral' TAD on chromosome 15 in mESCs. Flanking 3 × CTCF sites can be excised by Cre and Flp recombinases. Arrays are visualized by binding of LacI**-eGFP and TetR-tdTomato, respectively. Bottom, tiled Capture-C map (6.4-kb resolution) and genomic datasets in mESCs in a region in 2.6 Mb surrounding the engineered TAD. Capture-C was performed in cells where arrays were flanked by 3 × CTCF sites. Dashed lines, positions of LacO and TetO insertions. **b**, Capture-C maps in mESC lines with (left) or without (middle) 3 × CTCF sites flanking TetO and LacO arrays, and differential map (right, +3 × CTCF versus −3 × CTCF, Methods) highlighting interactions formed between convergent 3 × CTCF sites (arrows). **c**, Top, representative fluorescence microscopy images of mESCs with 3 × CTCF-LacO and TetO-3 × CTCF insertions. Bottom, magnified view with time series overlay of LacI**-eGFP and TetR-tdTomato signals (exposure time 50 ms, deconvolved, maximum intensity projection, bicubic interpolation). **d**, Representative trajectories of TetO-LacO radial distances with or without

convergent 3 × CTCF sites, either before or after degradation of RAD21 (2 h of dTag-13) (dt = 30 s). **e**, Distribution of TetO-LacO radial distances in the four experimental conditions (+3 × CTCF sites/+RAD21: $n$ = 152 cells, 4 pooled replicates; −3 × CTCF sites/+RAD21: $n$ = 214 cells, 4 pooled replicates; +3 × CTCF sites/−RAD21: $n$ = 248 cells, 7 pooled replicates; −3 × CTCF sites/−RAD21: $n$ = 277 cells, 6 pooled replicates). **f**, Distributions of variance over mean within single trajectories across the four experimental conditions (no. of cells as in panel **e**). Boxes, lower and upper quartiles (Q1 and Q3, respectively). Whiskers denote 1.5 × interquartile region (IQR) below Q1 and above Q3. $P$ values are calculated using two-sided Kolmogorov–Smirnov test. NS, not significant; **$P < 0.01$; ****$P < 0.0001$. Exact $P$ values can be found in Supplementary Table 2. Outliers are not shown. **g**, Distribution of jump step size (changes in TetO-LacO radial distance) across increasing time intervals for the four experimental conditions (no. of cells as in panel **e**). Boxes, lower and upper quartiles (Q1 and Q3, respectively). Whiskers, 1.5 × IQR below Q1 and above Q3. Outliers are not shown.

their variability in time (Fig. 4f), also supported by analysis of distance changes (jumps) as a function of time (Fig. 4g).

Finally, to test whether the effects of cohesin on chromosome motion would be different in the presence of active transcription at nearby locations, we measured looping dynamics this time in a parental mESC line before the removal of resistance cassettes. In this line, both the TetO and LacO arrays were immediately flanked by mouse *Pgk1* promoters[47] driving the transcription of resistance genes (Extended Data Fig. 6f,g). In line with previous studies[48,49], we found that active transcription led to slightly decreased radial MSD. Cohesin depletion

resulted in similar amounts of increased radial mobility irrespective of the presence or absence of active promoters (Extended Data Fig. 7c).

## Chromosomal contacts are transient

We next set off to quantify changes in distances over time and determine whether despite the experimental uncertainty on 3D distances (Extended Data Fig. 6e) we could observe transitions between two states: a 'proximal' state with small radial distances (presumably including cohesin-mediated loops between convergent CTCF sites), and a generic 'distal' state with larger spatial distances corresponding to

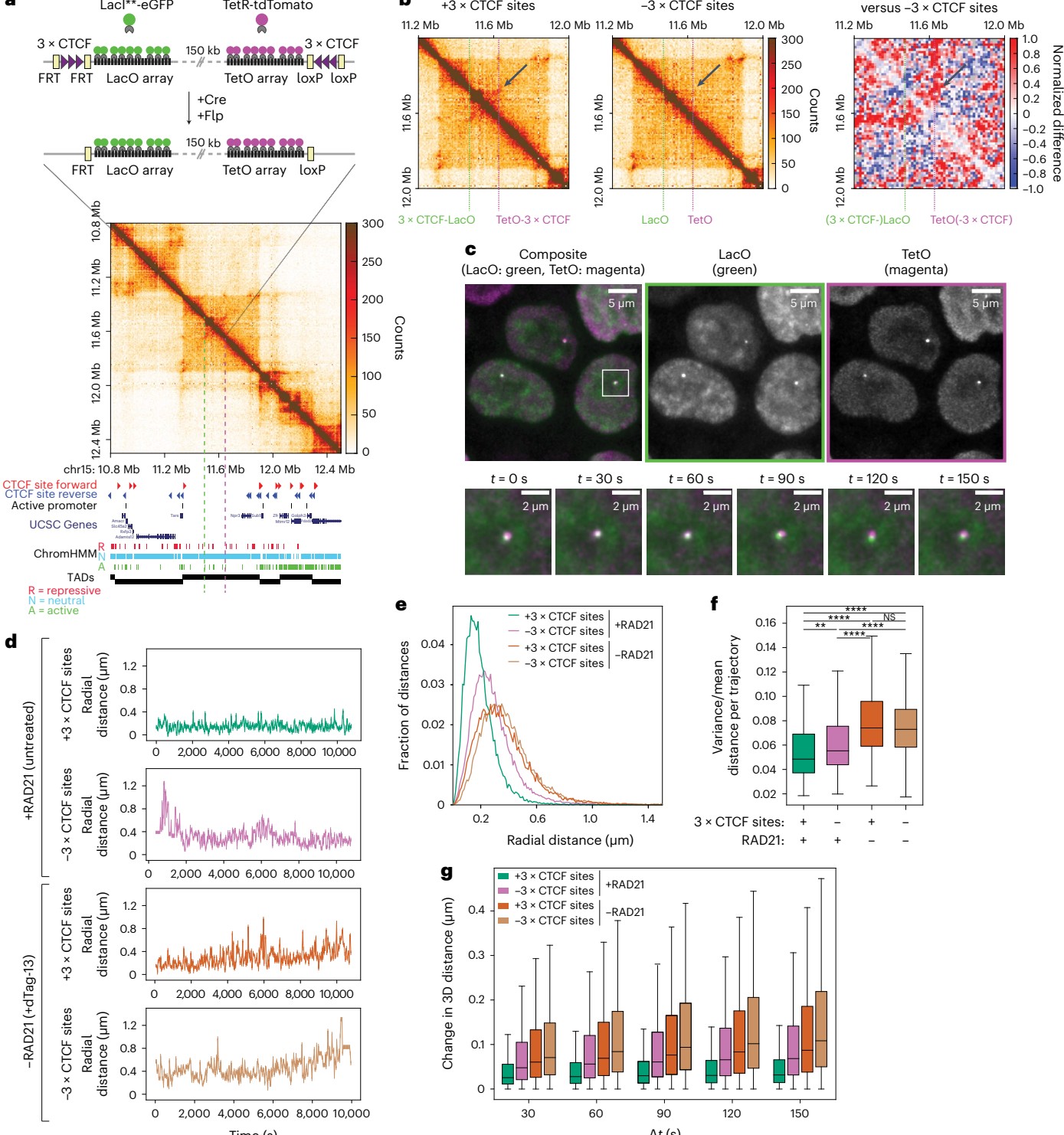

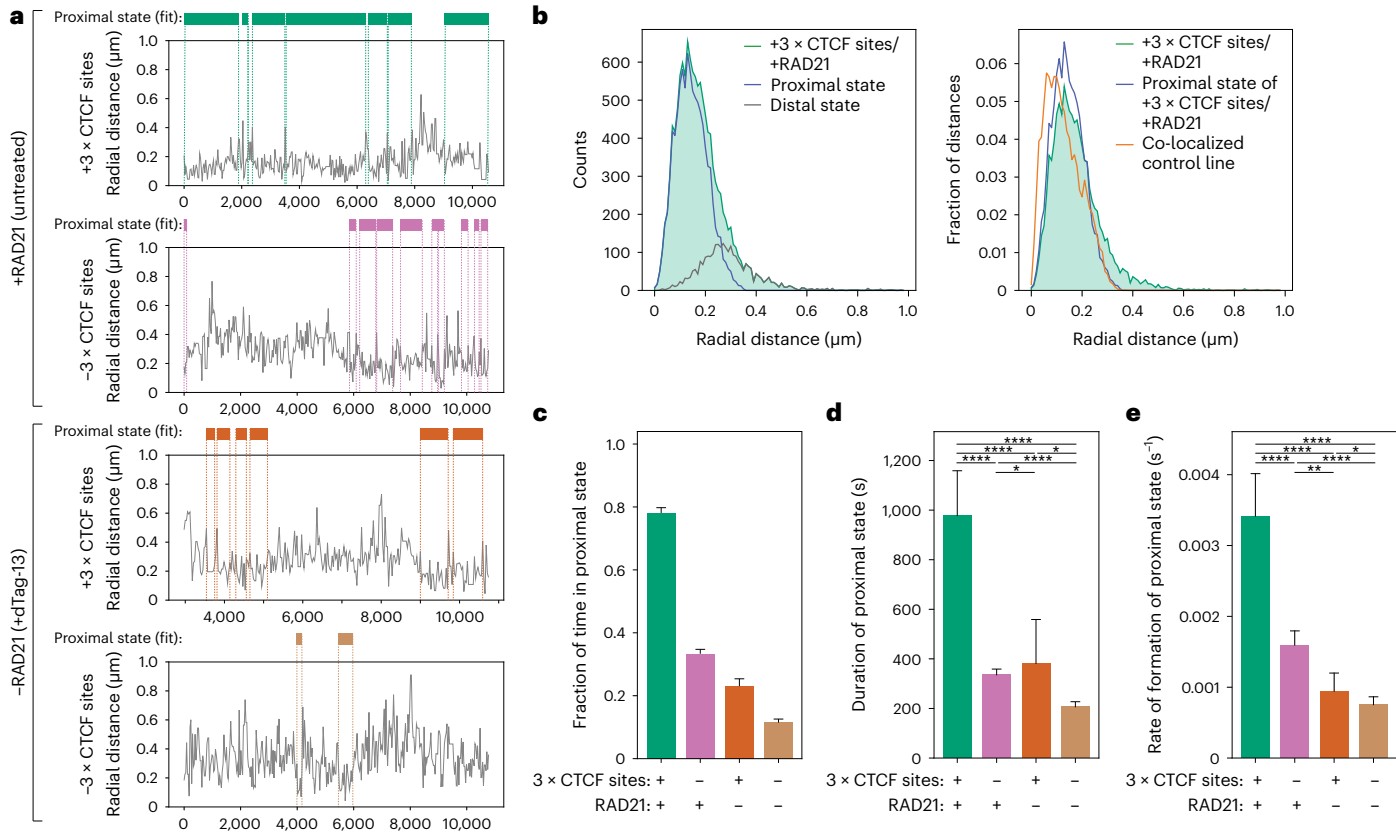

**Fig. 5 | Cohesin and CTCF control contact dynamics inside a TAD.**
**a**, Representative trajectories of radial distance (gray) and occurrences of the proximal state called by HMM (colored bars). The HMM was fitted on data with convergent 3 × CTCF sites and RAD21 (top left) to find the proximal state which was then imposed on the other three samples. **b**, Left, radial distance distribution in cells with convergent 3 × CTCF sites and RAD21 overlaid with those of proximal and distal states called by HMM on the same sample. Right, same as in the left panel but normalized and with the additional display of the distance distribution from a control cell line where TetO and LacO signals perfectly co-localize. **c**, Fraction of time spent in the proximal state called by HMM in the four experimental conditions (no. of replicates as indicated in

Fig. 4e). Shown are averages across experimental conditions; error bars represent bootstrapped (n = 10,000) standard deviations. **d**, Average durations of proximal states (mean ± 95% confidence interval (CI), n = 680 (−3 × CTCF/+RAD21); n = 287 (+3 × CTCF/+RAD21); n = 268 (−3 × CTCF/−RAD21); n = 114 (+3 × CTCF/−RAD21)). P values (two-sided Kolmogorov–Smirnov): *P < 0.05; **P < 0.01; ***P < 0.001; ****P < 0.0001. Exact P values can be found in Supplementary Table 2. **e**, Average rates of contact formation−time elapsed between the end of a proximal state and the beginning of the next (mean ± 95% CI, n = 726 (−3 × CTCF/+RAD21); n = 323 (+3 × CTCF/+RAD21); n = 268 (−3 × CTCF/−RAD21); n = 138 (+3 × CTCF/−RAD21)). P values as in panel **d**.

other configurations of the chromatin fiber. This was motivated by the expectation that any polymer with site-specific attractive interactions, such as those mediated by cohesin at convergent CTCF sites, should in principle result in two-state thermodynamic behavior. We thus fitted a two-state hidden Markov model (HMM) on the ensemble of trajectories obtained in cells where both convergent 3 × CTCF sites and RAD21 were present (Fig. 5a,b). Interestingly, distances in the proximal state inferred by HMM largely overlapped with those detected on perfectly colocalizing signals in control experiments where TetR-eGFP and TetR-tdTomato were bound to the same set of randomly inserted TetO arrays (149 versus 130 nm on average, respectively) (Fig. 5b and Extended Data Fig. 6e). The proximal state thus corresponds to configurations of the chromatin fiber where the two arrays were in very close physical proximity, also including (but not restricted to) cohesin-mediated loops between CTCF sites. For simplicity, we refer to the proximal state interchangeably as 'contact', without implying a direct molecular interaction between the two DNA fibers. Radial distances in the distal state (288 nm on average) instead were similar to those measured in cells where both CTCF sites had been removed (291 nm) (Extended Data Fig. 8a,b). Thus, the distal state largely overlapped with chromosome conformations where specific cohesin-mediated CTCF loops were lost.

We next fitted the HMM to all experimental conditions while keeping the same proximal state as in cells with 3 × CTCF sites and RAD21 (Fig. 5a). This showed that in the presence of RAD21, the LacO and TetO arrays spent ~78% of the time in contact (that is, in the proximal state) when 3 × CTCF sites were present. This was 2.3× higher than the 33% of time they spent in contact when the 3 × CTCF sites were removed (Fig. 5c), in agreement with the corresponding 2.8-fold difference in contact probability inferred from Capture-C (Fig. 4b). The fraction of time spent in contact decreased markedly upon depletion of RAD21 to ~23% in the presence of 3 × CTCF sites and 11% in the absence (Fig. 5c). Both the average duration of contacts and their rate of formation were maximal in the presence of RAD21 and 3 × CTCF sites, where they lasted around 16 min and reformed every 5 min on average (Fig. 5d,e). Contacts became substantially shorter (6 min) and rarer (one every 10 min) when 3 × CTCF sites were removed, and even more so upon RAD21 depletion (lasting 2 min and occurring every 22 min on average). Interestingly, these results were not affected by the presence of actively transcribed promoters in the immediately flanking regions (Extended Data Fig. 8b–f), in line with the lack of changes in contact probability measured in Capture-C (Extended Data Fig. 6g). Thus, both cohesin and CTCF impact both the duration and the probability of

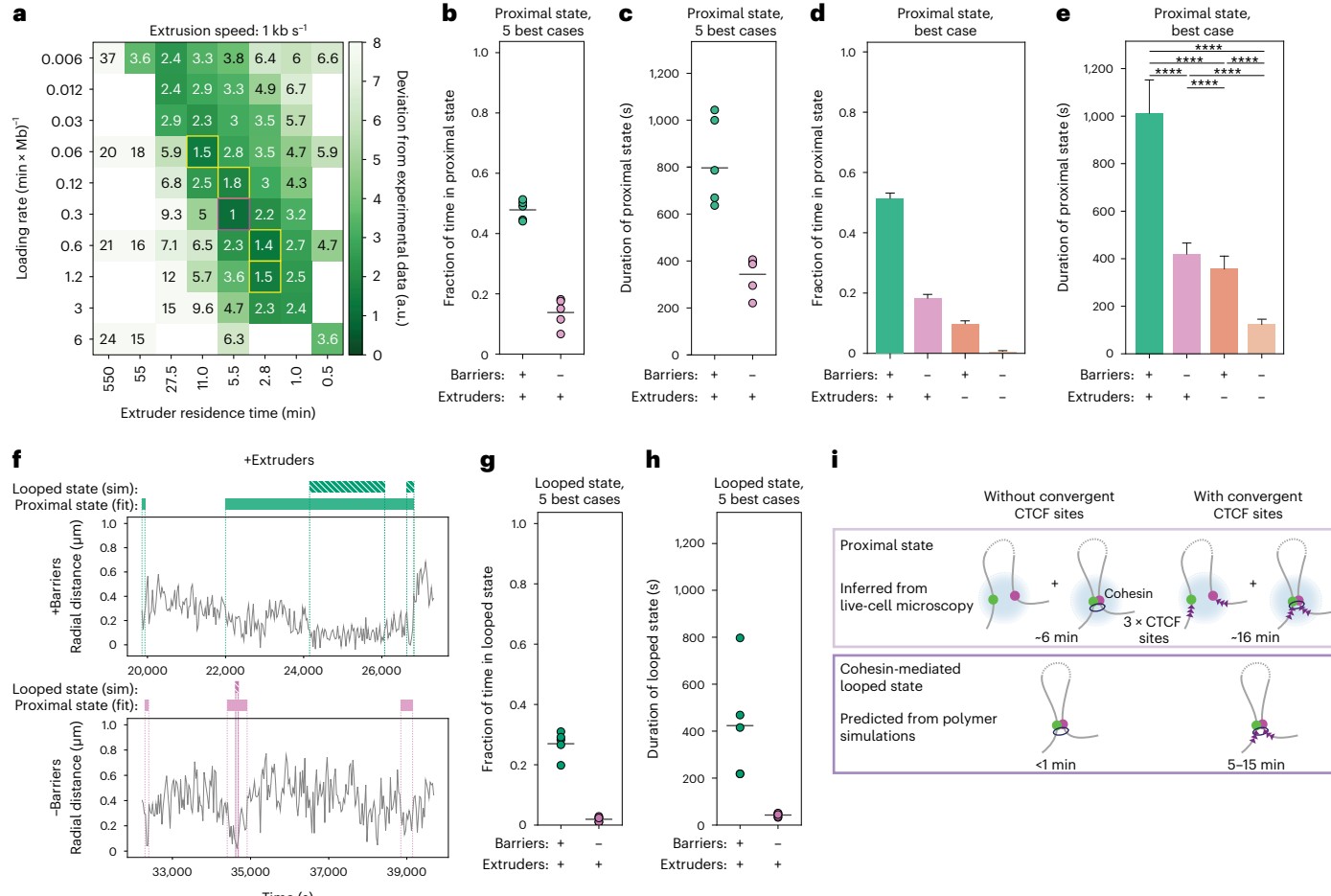

**Fig. 6 | Estimation of frequency and duration of cohesin-mediated CTCF loops. a**, Levels of agreement between simulations and experimental data as a function of loop extrusion parameters (here shown with extrusion speed 1 kb s$^{-1}$). The score represents the deviations of the distance, duration and fraction of time spent in the proximal state with those experimentally observed in the presence of RAD21 with or without 3 × CTCF sites (Methods). Magenta square, parameter set maximizing the agreement with experimental values. Yellow squares, four additional second-best parameter sets. **b**, Fraction of time spent in the proximal state called by HMM on simulations with the five best-matching parameters (magenta and yellow squares in panel **a** for +Extruder case, Methods). **c**, Average duration (mean ± 95% CI) of proximal state called by HMM on simulations with the five best-matching parameters. **d**, Fraction of time spent in the proximal state called by HMM on simulations (over $n$ = 15,880 time points) for the best-matching parameter set in the presence of extruders (+) or low levels (−) of extruders, either with or without extrusion barriers. Shown are averages across experimental conditions; error bars represent bootstrapped ($n$ = 10,000) standard deviations. **e**, Average duration of the proximal state (mean ± 95% CI, over $n$ = 15,880 time points) either in the presence of extruders (+) or low levels of extruders (−), either with or without extrusion barriers. Two-sided Kolmogorov–Smirnov $P$ values can be found in Supplementary Table 2. **f**, Representative trajectories of radial distances (gray), contact states called by HMM (full bar) and looped states in the underlying polymer conformations (striped bars) from +Extruders/+Barriers (top) and +Extruders/−Barriers simulations (bottom) with best-matching parameters (magenta square in panel **a**). **g**, Fraction of time spent in the looped state based on simulations with the five best-matching parameters. **h**, Average duration of the looped state based on simulations with the five best-matching parameters (mean ± 95% CI). **i**, Scheme summarizing the durations of proximal and looped states in the presence and absence of 3 × CTCF sites. a.u., arbitrary unit; sim, simulation.

formation of chromosomal contact events between loci separated by 150 kb within an 'empty' TAD.

To understand if these results could be rationalized in terms of loop extrusion, we compared them with polymer simulations with convergent impermeable loop extrusion barriers separated by ~150 kb. Simulations were performed using loop extrusion parameters spanning a finer-grained 25-fold range around experimentally realistic values that reproduced the dynamic effect of RAD21 degradation (compare with Fig. 2d, black square) and with both in vitro and in vivo estimates of extrusion speeds[20,38]. In a large region of the parameter space, distances between convergent barriers were bimodally distributed, supporting the expectation that the polymer can be approximated as a two-state system (Extended Data Fig. 9a). To allow direct comparison with experimental distance-based HMM states, we applied random

errors matching experimental uncertainty levels to radial distances generated by the models (Extended Data Fig. 9b). We called proximal and distal states using the same HMM strategy as with experimental data. Importantly, for a large number of parameter combinations, distances in the proximal state largely overlapped with the corresponding distribution observed experimentally in the presence of convergent CTCF sites and cohesin (Extended Data Fig. 9c and Supplementary Fig. 1a).

We then compared the distance, duration and fraction of time spent in the proximal state with those experimentally observed in the presence of RAD21 with or without 3 × CTCF sites. We found that their similarity was maximal for extruder densities ranging from 8 to 32 per Mb (Supplementary Fig. 1e) and residence times of 2.8–11 min, with extrusion speeds of both 0.1 and 1 kb s$^{-1}$, all of which

were in the range of previous estimations of experimental values[20,21,38,50] (Fig. 6a and Extended Data Fig. 9d). Considering the five best-matching scenarios (red- and yellow-marked values in Fig. 6a), the two locations spent 45–55% of the time in the proximal state with an average contact duration of around 10–17 min, which reduced to 18% and 8 min in the absence of extrusion barriers (Fig. 6b,c). Similar to the effects observed experimentally upon depletion of RAD21, decreasing extruder densities (for example, by decreasing loading rates) led to decreased fractions of time and shorter durations of the proximal state (Fig. 6d,e, shown for the best case, general trends in Supplementary Fig. 1d–d). Thus, the duration and the fraction of time spent in the proximal state, and most importantly how these quantities change upon removing cohesin and/or CTCF sites, can be understood in terms of a simple loop extrusion model.

The HMM-based proximal state likely provides an overestimation of the duration of underlying CTCF-CTCF loops mediated by stalled cohesins, since it also contains a fraction of CTCF-independent proximity events that cannot be distinguished from loops. To estimate the duration and times the two loci spent in a cohesin-mediated CTCF-CTCF looped conformation, we quantified occurrences in the simulated polymer where the two monomers formed the base of an extruded loop (Fig. 6f and Methods). As expected, these events were rarer and shorter than contacts detected by HMM on polymer simulations (Fig. 6b,c), with two monomers spending ~20–31% of time at a loop base for 5–15 min on average in the presence of extrusion barriers (Fig. 6g,h). Finally, transient cohesin-dependent loops that are not stabilized by CTCF sites should occur much more rarely (1–3% of the time) and lasted less than a minute on average (Fig. 6g,h). Comparison of polymer simulations with HMM states thus suggests that the dynamics of chromosome contacts detected at a range of 150 nm are generated by faster and rarer cohesin-mediated CTCF loops (Fig. 6i).

## Discussion

Our study provides quantitative measurements of chromosome folding dynamics in living cells and reveals how they are controlled by cohesin and CTCF. Two experimental strategies allow us to minimize biological variation from specific regulatory and structural genomic contexts and enable direct comparison with polymer models. By studying large numbers of random genomic locations, we average over local differences in chromosome mobility and reveal the global dynamic effects of cohesin. By visualizing and manipulating two locations within a 'neutral' genomic environment, we unravel how cohesin and CTCF impact chromosome looping within a single TAD. We show that although higher extrusion speeds could in principle result in increased chromosome motion (Extended Data Fig. 9e,f), physiological extrusion rates rather generate transient constraints that decrease chromosome dynamics, in line with previous measurements of histone mobility[49]. Similar to previous reports[51], we observe that constraints introduced by cohesin reduce spatial distances between genomic sequences in cis and increase the chances that they interact. We now, however, reveal that this entails an increase in both the rate of formation and the duration of contacts. Convergently oriented high-affinity CTCF motifs lead to higher contact frequencies and substantially longer contact durations, somewhat similar to the effect of insulator elements in Drosophila[23]. Comparison with polymer simulations reveals that in mESCs this can be understood in terms of stalling of loop-extruding cohesins. This observation also suggests that asymmetries in contact patterns established by CTCF motifs genome-wide might also lead to temporal asymmetries in physical interactions, notably between regulatory sequences. We additionally observe that constraints introduced by cohesin and CTCF sites lead to reduced temporal variability in physical distances, arguing that loop extrusion increases the reproducibility of chromosome folding at selected genomic sites.

Our study also provides estimates of the frequency and duration of chromosomal contacts at genomic-length scales that represent enhancer–promoter communication genome-wide. In our study, contacts are defined by physical distances (~150 nm) that might be comparable to those where signals arise in 3C methods[11]. For sequences separated by 150 kb, such contacts assemble and disassemble over minutes. This provides many opportunities in a single cell-cycle for regulatory sequences in a TAD to contact each other, and suggests that long-range regulation by distal enhancers might rely on transient interactions. We note that despite accurate correction of chromatic aberrations, shorter-range and thus potentially faster proximity events remain inaccessible in our experimental set-up[52]. Estimates based on comparison with polymer simulations further suggest that cohesin-mediated interactions between convergent CTCF sites might last around 5–15 min on average and at least for sequences located 150 kb apart occur around 27% of the time. This is in good agreement with recent estimates of the duration of a 500-kb loop in mESCs (10–30 min on average)[26], which, however, occurs more rarely (3.5–6% of the time). This is in line with the predictions from polymer simulations that increasing the genomic distance between convergent CTCF sites should substantially decrease the frequency of CTCF-mediated interactions, but not their duration (Extended Data Fig. 10). Taken together, our data establish firm quantitative bases for understanding the dynamics of chromosome folding within TADs and provide temporal constraints for mechanistic models of chromosome structure and its impact on fundamental biological processes such as long-range transcriptional regulation.

## Online content

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

## Methods

### Culture of mESC lines

All cell lines are based on the E14Tg2a parental mESC line (karyotype 19, XY, 129/Ola isogenic background; E14 for brevity). E14 CTCF-AID-eGFP (clone EN52.9.1) was published by Nora et al.[31]. E14 WAPL-AID-eGFP and E14 RAD21-AID-eGFP were published by Liu et al.[32]. The latter were kindly provided by Elzo de Wit (Netherlands Cancer Institute). All cell lines for the dual-array imaging approach are based on the double-CTCF knockout cell line described by Zuin et al.[3]. Cells were cultured on gelatin-coated culture plates in Glasgow Minimum Essential Medium (Sigma-Aldrich, G5154) supplemented with 15% fetal calf serum (Eurobio Abcys), 1% L-Glutamine (Thermo Fisher Scientific, 25030024), 1% Sodium Pyruvate MEM (Thermo Fisher Scientific, 11360039), 1% MEM Non-Essential Amino Acids (Thermo Fisher Scientific, 11140035), 100 μM β-mercaptoethanol (Thermo Fisher Scientific, 31350010), 20 U ml$^{-1}$ leukemia inhibitory factor (Miltenyi Biotec, premium grade) in 8% $CO_2$ at 37 °C. Cells were tested for mycoplasma contamination regularly and no contamination was detected. For Hi-C, Capture-Hi-C, 4C-seq, western blot and imaging experiments, cells were cultured in standard E14 medium supplemented with 2i (1 μM MEK inhibitor PDO35901 (Axon, 1408) and 3 μM GSK3 inhibitor CHIR 99021 (Axon, 1386)). For live-cell imaging experiments, cells were cultured in Fluorobrite DMEM (Gibco, A1896701) supplemented with 15% fetal calf serum (Eurobio Abcys), 1% L-Glutamine (Thermo Fisher Scientific, 25030024), 1% Sodium Pyruvate MEM (Thermo Fisher Scientific, 11360039), 1% MEM Non-Essential Amino Acids (Thermo Fisher Scientific, 11140035), 100 μM β-mercaptoethanol (Thermo Fisher Scientific, 31350010), 20 U ml$^{-1}$ leukemia inhibitory factor (Miltenyi Biotec, premium grade) and with 2i inhibitors (1 μM MEK inhibitor PDO35901 (Axon, 1408) and 3 μM GSK3 inhibitor CHIR 99021 (Axon, 1386)).

### Generation of mESC lines carrying random integrations of TetO arrays

To generate clonal cell lines carrying random integrations of the TetO array in the degron cell lines (E14 Rad-AID-eGFP, E14 CTCF-AID-eGFP and E14 WAPL-AID-EGFP), 0.5 × 10$^6$ cells were transfected with 2 μg of PB-3 × CTCF-TetO vector, 200 ng of PB-TetR-tdTomato and 200 ng of pBroad3_hyPBase_IRES_tagRFPt (ref. [53]) with Lipofectamine3000 (Thermo Fisher Scientific, L3000008) according to the manufacturer's recommendations. Cells were cultured in standard E14 medium for 5 d and subsequently sorted by FACS for fluorescence emission at 581 nm (tdTomato) on 96-well plates to isolate clonal lines. Sorted cells were kept for 2 d in standard E14 medium supplemented by 100 μg μl$^{-1}$ primocin (InvivoGen, ant-pm-1) and 10 μM ROCK inhibitor (STEMCELL Technologies, Y-27632). At 10 d after sorting, the plates were duplicated by detaching with accutase (Sigma-Aldrich, A6964) and re-seeding in full E14 culture medium. One-third of the cells were replated onto Corning High-Content Imaging Glass Bottom Microplates (96-well, Corning, 4580). At 2 d after re-seeding, clonal lines were screened by microscopy for >10 insertions of TetO per cell and a good signal-to-noise ratio (SNR). Selected clones were expanded and genotyped by PCR for the absence of random integration of the piggyBac itself. Primers used for genotyping are listed in Supplementary Table 3.

### Generation of dual-array (TetO-LacO) mESC line

Integration of the TetO array into the genomic locus on chr15:11,647,372: the vector containing the guide RNA (gRNA) sequence was available from a previous study (PX459-chr15_gRNA/Cas93). The gRNA sequence can be found in Supplementary Table 3. E14 mESCs already containing a double-knockout for CTCF sites (clone D6 in ref. [3]) were transfected with the targeting vector pMK-3 × CTCF-TetO-Rox-PuroR-Rox and the gRNA vector PX459-chr15_gRNA/Cas9 using nucleofection with the Amaxa 4D-Nucleofector X-Unit and the P3 Primary Cell 4D-Nucleofector X Kit (Lonza, V4XP-3024 KT). Then, 2 × 10$^6$ cells were nucleofected with 1 μg of TetO targeting vector and 1 μg of PX459-ch15_gRNA/Cas9

as described above and treated with 1 μg ml$^{-1}$ puromycin (InvivoGen, ant-pr-1) 48 h after transfection for 3 d to select cells for insertion of the TetO cassette. Cells were then cultured in standard E14 medium for an additional 7 d and subsequently sorted by FACS on 96-well plates as described above to isolate clonal lines. At 10 d after sorting, the plates were duplicated by detaching with accutase (Sigma-Aldrich, A6964) and re-seeding in full E14 culture medium. Genomic DNA was extracted on-plate by lysing cells with lysis buffer (100 mM Tris-HCl pH 8.0, 5 mM EDTA, 0.2% SDS, 50 mM NaCl and 1 mg ml$^{-1}$ proteinase K (Macherey-Nagel, 740506)) and 0.05 mg ml$^{-1}$ RNase A (Thermo Fisher Scientific, EN0531) and subsequent isopropanol precipitation. Individual cell lines were analyzed by genotyping PCR to determine heterozygous insertion of the TetO cassette. Cell lines showing the corrected genotype were selected and expanded. Primers used for genotyping are listed in Supplementary Table 3. Targeted nanopore sequencing with Cas9-guided adapter ligation[45] was performed on expanded clones to confirm single-copy insertion of the TetO cassette. Clone 2G5 was used for further engineering. Integration of the LacO array into the genomic locus on chr15:11,496,908: the gRNA sequence for the CRISPR–Cas9 knock-in of the LacO cassette was designed using the online tool https://eu.idtdna.com/site/order/designtool/index/CRISPR_SEQUENCE and purchased from Microsynth AG. The gRNA sequence can be found in Supplementary Table 3. The gRNA sequence was cloned into the PX330 plasmid (Addgene, no. 58778) using the BsaI restriction site. The clonal line carrying the TetO cassette (clone 2G5) was transfected with the targeting vector pUC19-ITR-NeoR-ITR-3 × CTCF-LacO and the gRNA vector pX330-chr15_LacO_gRNA/Cas9 using nucleofection with the Amaxa 4D-Nucleofector X-Unit and the P3 Primary Cell 4D-Nucleofector X Kit (Lonza, V4XP-3024 KT). A total of 2 × 10$^6$ cells were harvested using accutase (Sigma Aldrich, A6964) and resuspended in 100 μl transfection solution (82 μl primary solution, 18 μl supplement, 15 μg targeting vector and 5 μg of gRNA vector) and transferred to a single Nucleocuvette (Lonza). Nucleofection was performed using the protocol CG110. Transfected cells were directly seeded in pre-warmed E14 standard medium. At 48 h after transfection, 250 μg ml$^{-1}$ G418 (InvivoGen, ant-gn-1) was added to the medium for 3 d to select cells for insertion of the LacO cassette. Cells were sorted and genotyped as described for the TetO integration. Primers used for genotyping are listed in Supplementary Table 3. Cell lines showing the corrected genotype were selected and expanded. Expanded clones were transiently transfected with 200 ng of PB-TetR-tdTomato and 200 ng of PB-LacI-eGFP using Lipofectamine3000 according to the manufacturer's instructions (Thermo Fisher Scientific, L3000008) and 2 d after transfection validated for heterozygous insertion of the LacO cassette on the same allele as the TetO by microscopy. Targeted nanopore sequencing with Cas9-guided adapter ligation[45] was performed on correct clones to confirm single-copy insertion of the LacO cassette. Clone 1F11 was used for further engineering. To visualize the operator arrays in live-cell imaging and remove the puromycin resistance gene used for selection during integration, 0.5 × 10$^6$ E14 TetO-LacO cells (clone 1F11) were transfected with 200 ng of PB-TetR-tdTomato, 200 ng of PB-LacI-eGFP and 200 ng of pBroad3_hyPBase_IRES_tagRFPt (ref. [54]) with Lipofectamine3000 (Thermo Fisher Scientific, L3000008) according to the manufacturer's instructions. At 7 d after transfection the cells were sorted (as described previously) for fluorescence emission at 507 nm (eGFP) and 581 nm (tdTomato). Sorted cells were cultured and genotyped as described for the random TetO integration. Primers used for genotyping are listed in Supplementary Table 3. Cell lines showing the corrected genotyping pattern were selected and expanded and a good and comparable SNR was selected for by microscopy. Clones 1B4 (+PuroR) and 2C10 (−PuroR) were used for further engineering.

### Live-cell imaging

First, 35-mm glass-bottom dishes (Mattek, P35G-1.5-14-C) were coated with 1–2 μg ml$^{-1}$ Laminin (Sigma-Aldrich, L2020) in PBS at

37 °C overnight. Cells ($1 \times 10^6$) were seeded in Fluorobrite medium (as described above) 24 h before imaging. For targeted degradation of RAD21, WAPL or CTCF in the degron cell lines, the medium was exchanged to medium containing 500 µM auxin (Sigma-Aldrich, I5148-2G) at the respective time required for complete degradation of the protein target before imaging (RAD21: 90 min, WAPL: 24 h, CTCF: 6 h). For targeted depletion of RAD21 using the FKBP degron system (dual-array cell lines), cells were cultured in Fluorobrite medium containing 500 nM dTAG-13 (Sigma-Aldrich, SML2601-1MG) 2 h before imaging. For fixed cell measurements to estimate the localization error, $1 \times 10^6$ cells were seeded onto Mattek dishes and incubated for 24 h at 37 °C, 8% $CO_2$. The medium was removed and the cells were fixed in 4% paraformaldehyde (Electron Microscopy Sciences, 15710) in PBS for 10 min at room temperature. The cells were washed three times in PBS and Fluorobrite medium was added to the Mattek dish to achieve comparable background fluorescence levels. Cells were imaged with a Nikon Eclipse Ti-E inverted widefield microscope equipped with a Total Internal Reflection Microscopy iLAS2 module (Roper Scientific), a Perfect Focus System (Nikon) and motorized Z-Piezo stage (ASI) using a CFI APO TIRF 100 ×1.49 NA oil immersion objective (Nikon). The microscope was operating in highly inclined and laminated optical sheet mode[34]. Excitation sources were a 48-nm, 200-mW Toptica iBEAM SMART laser and a 561-nm, 200-mW Coherent Sapphire laser. Images were collected on two precisely aligned back-illuminated Evolve 512 Delta EMCCD cameras with a pixel size of $16 \times 16$ µm$^2$ (Photometrics). Cells were maintained at 37 °C and 8% $CO_2$ using an enclosed microscope environmental control set-up (The BOX and The CUBE, Life Science Instruments). Before the acquisition of movies for the dual-array set-up, TetraSpeck Microspheres, 0.1-µm beads (Thermo Fisher Scientific, T7279), were imaged to allow for correction of chromatic aberrations during image processing and analysis. Movies for measurement of random TetO integrations in degron cell lines were acquired every 10 s (exposure time: 50 ms) in 34 z-planes (10-µm stack, distance between consecutive z planes = 300 nm) with the Visiview software (Visiview 4.4.0.12, Visitron). Images for measurement of cell lines with the dual-array set-up were acquired every 30 s, with an exposure time of 50 ms, respectively, each in a sequential mode with 21 z-planes (6-µm stack, dz = 300 nm). For the measurement of the time it takes the operator arrays to displace by their own size, images were acquired continuously on a single focal plane over 10 s every 0.1 s with exposure times of 50 ms.

## Image processing
Raw images were deconvolved using the Huygens Remote Manager and a classical maximum likelihood estimation algorithm with a theoretical point-spread function. The initial SNRs were estimated from the images and images were deconvolved until one of the following stopping criteria was reached: the maximum number of iterations was performed (for random integrations: 20 cycles, for tdTomato and eGFP; in dual-color set-up: 15 cycles for tdTomato signal, 5 cycles for GFP signal) or a quality change criterion below 0.001 was returned. Representative image series shown in the main figures were deconvolved as described above, adjusted to display the same brightness and contrast, and interpolated using a bicubic interpolation. Movies were corrected for bleaching over time using an exponential fit. The two-dimensional (2D) projection of intensity changes over time was created using the Temporal Color Code in Fiji v.2.0. (https://github.com/fiji/fiji/blob/master/plugins/Scripts/Image/Hyperstacks/Temporal-Color-Code.ijm).

## Spot detection and localization of multi operator data
Our field of view typically contains approximately 25 mESC nuclei. Despite the fact that our mESC lines are clonal, background nuclear fluorescence intensities in each cell can vary substantially. This poses challenges to conventional threshold-dependent algorithms for spot detection and localization which perform unevenly across cells with different background intensities. To overcome these limitations, we implemented a two-step procedure for 3D spot detection and localization. To detect spots, we used deepBlink v.0.1.1 (ref. [35]), a convolutional neural network-based spot detection and localization algorithm in two dimensions, which has been shown to be able to deal with different background intensities and to detect spots in a threshold-independent manner. To enhance our detection efficiency, we employed custom models trained on a combination of the following datasets: smFISH and SunTag datasets provided by deepBlink and in-house manually curated live-cell imaging images. To detect 3D spots, we applied deepBlink to all z-stacks separately followed by linkage of the spots across z-stacks using Trackpy[53]. The precise 3D coordinates of the spots were then determined using 3D Gaussian fitting using a voxel of size $6 \times 6 \times 4$ pixels centered at the spot in the brightest z-stack. deepBlink models can be found at https://github.com/zhanyinx/SPT_analysis/tree/main/models. The parameters and models used for each cell line can be found in Supplementary Table 4. All scripts used for the analysis can be found at https://github.com/zhanyinx/SPT_analysis/.

## Tracking and cell motion correction of multi operator data
3D spots coordinates are fed into TrackMate for tracking using linear assignment problem (LAP) tracker. Each track is assigned to manually annotated cell masks (from max z-projection of frame 93) using a custom script (https://github.com/zhanyinx/SPT_analysis/blob/main/source/spot_detection_tracking/assign_cellids.py), which uses the majority rule. Motion correction is then performed using a roto-translation model. Specifically, for each pair of consecutive time frames, a set of matching spots in every cell is determined by solving the LAP using the Euclidean distance between spots as a measure of distance. Only spots that match across two consecutive frames are then used to estimate the roto-translation model which is then applied to correct for nuclear motion (six matching spots on average across all time frames, trajectories and movies, with a minimum of four spots per pair of time frames). All scripts used for the analysis can be found at https://github.com/zhanyinx/SPT_analysis/.

## MSD analysis of multi operator data
Tracks with fewer than ten spots are filtered out for follow-up analysis. To calculate the MSD, we first calculate the time-averaged MSD for each trajectory. We then calculate the ensemble average (across trajectories) MSD by pooling all replicates. The ensemble average is done in log space. We corrected the localization error effect on the MSD curve by estimating the standard deviation of the error distribution using fixed images as described by Kepten et al.[55]. To calculate the scaling ($\alpha$) and the generalized diffusion coefficient ($D$) of each MSD curve, we fitted the ensemble average of the log-time average MSD between 10 and 100 s. To test the significance of differences between conditions, we fitted $\alpha$ and diffusion coefficient for each cell. The $P$ value is calculated using Student's $t$-test (two-sided). Since we are always comparing two conditions whose cell-cycle profiles are similar, we ignore the effect of sister chromatids. All scripts used for the analysis can be found at https://github.com/zhanyinx/SPT_analysis/. The specific Fiji and relative plug-ins can be found at https://github.com/giorgettilab/Mach_et_al_chromosome_dynamics/tree/master/Fiji.

## Chromatic aberration correction of dual-color data
To correct for chromatic aberration we took 3D image stacks of TetraSpeck Microspheres, 0.1-µm beads (Thermo Fisher Scientific, T7279), adsorbed on MatTek dishes in $1 \times$ PBS at the beginning of every imaging session and used them to correct the corresponding set of movies. After detecting signals from single beads in each channel using deepBlink and determining their 3D location by Gaussian fitting, we first identified spots that are shared across channels by solving the LAP using the Euclidean distance between spots. We then used the common set of bead signals to compute a 3D roto-translation that we

finally applied to *xyz* positions. This procedure corrects for *x*, *y* and *z* aberrations simultaneously. The same transformations accurately corrected chromatic aberrations in actual experiments in double-labeled mESCs ('Control TetO' in Extended Data Fig. 6g), with the exception of a small residual systematic shift (approximately 40 nm) along the *z* axis ('TetO-LacO case' in Extended Data Fig. 6g), which is likely due to 3D image anisotropies that cannot be measured using '2D' bead images.

### Tracking and MSD analysis of dual-color data

To increase the ability to detect longer tracks, we used an in-house script to stitch multiple tracks belonging to the same cell ([https://github.com/zhanyinx/SPT_analysis/blob/main/source/dual_channel_analysis/utils.py](https://github.com/zhanyinx/SPT_analysis/blob/main/source/dual_channel_analysis/utils.py), stitch function). In short, if two tracks from the same cell overlap more than 50% in time, the shortest one is filtered out. We called cell masks using CellPose[56] on the max *z*-projection of the middle frame of the movie using the GFP channel and used these masks to define cell identity. For tracks with overlaps lower than 50%, the overlapping part of the tracks are randomly removed from one of the two tracks. The resulting tracks are stitched if the distance across the time gap is smaller than 1.6 µm. To match tracks across channels, we used the following measure to calculate the distance between tracks across channels:

$$\frac{\sum_{i=1}^{3} <(x_{1i}(t) - x_{2i}(t))^2>_{t \in T_1 \cap T_2}}{\sqrt{\text{len}(t \in T_1 \cap T_2)}}$$

Where $x_1$ are the coordinates from channel 1 and $x_2$ are the coordinates from channel 2, $T_1$ contains all the time frames from channel 1 and $T_2$ contains all the time frames from channel 2, and len is a function that returns the length of an array. We solved the LAP using the distance measure above to match tracks across channels. Tracks with average distances across channels higher than 1 µm are filtered out. Matched tracks with lower than 25 time points are filtered out. For each matched pair of tracks, we calculate the pairwise distance using the Euclidean distance in three dimensions. We define noisy pairwise distance using the ratio of the pairwise distance in three dimensions and two dimensions. In particular, we defined as noisy the top 5% of this ratio and filtered them out. To calculate the radial MSD, we first calculate the time-averaged radial MSD for each pairwise distance 'trajectory'. We then calculate the ensemble average (across trajectories) of the log of time-averaged radial MSD. We corrected for the radial localization uncertainty by estimating the standard deviation of the error distribution using fixed images as described by Kepten et al.[55]. To calculate the scaling ($\alpha$) and the generalized diffusion coefficient ($D$) of each MSD curve, we fitted the ensemble average time average MSD between 30 and 300 s. Since we are always comparing two conditions whose cell-cycle profiles are similar, we ignore the effect of sister chromatids. All scripts used for the analysis can be found at [https://github.com/zhanyinx/SPT_analysis/](https://github.com/zhanyinx/SPT_analysis/).

### Estimation of experimental uncertainty on radial distance

To estimate our uncertainty in detecting distances across channels, we used a cell line with multiple integration of TetO arrays that can be tagged with TetR-eGFP and TetR-tdTomato. Spot detection is done as for our dual-color lines. We corrected for chromatic aberration using TetraSpeck Microspheres, 0.1-µm beads (Thermo Fisher Scientific, T7279), and then matched spots across channels by solving the LAP using scipy.optimize.linear_sum_assignment function with the Euclidean distance between spots as a measure of distance. Spots across channels with distances higher than a threshold are filtered out to avoid mismatches. We used a threshold of 300 nm for matching the spots registration. We applied a second round of chromatic aberration correction using the set of registered points themselves. The resolution

limit (uncertainty) is then estimated as the average distance between registered spots which corresponds to 130 ± 70 nm.

### HMM for detection of the proximal state

To detect the proximal state in a threshold-independent manner, we used an HMM with two hidden states ('proximal' and 'distal'). We used a Gaussian model for the emission probabilities. Only distance trajectories with less than 20% missing values at any time point are kept. Missing values are filled with the first preceding time point with distance value. To more reliably detect the proximal state, we used all the trajectories from the experimental condition with both cohesin and CTCF sites to train an HMM. We then re-trained an HMM model for each experimental condition by using the proximal state (Gaussian mean and standard deviation) from the experimental condition with both cohesin and CTCF sites. Finally, we applied the experimental condition-specific HMM to every trajectory to estimate the contact duration and rate of contact formation for all the experimental conditions. The HMM model training can be found as a jupyter notebook ([https://github.com/zhanyinx/SPT_analysis/blob/main/notebooks/HMM_experimental_data.ipynb](https://github.com/zhanyinx/SPT_analysis/blob/main/notebooks/HMM_experimental_data.ipynb)). We modified the hmmlearn library to allow fixing proximal state during HMM training. The modified hmmlearn library can be found at [https://github.com/zhanyinx/hmmlearn](https://github.com/zhanyinx/hmmlearn).

### Simulations

Polymer simulations were performed using LAMMPS[57]. We chose Langevin dynamics with the *NVT* thermostat. Arbitrary units were set such that thermal energy $k_B T = 1$, where $k_B$ is the Boltzmann constant and *T* is room temperature, corresponding to 300 K. For every set of parameters, we performed ten independent runs. A run consists of an equilibration part of $10^7$ simulation steps and a production part of $10^8$ simulation steps. For subsequent analysis and calculation of contact maps, we recorded the data every $10^4$ simulation steps. In simulations for Fig. 2, the chain length was 1,125 beads. In simulations for Figs. 5 and 6, the chain length was 1,000 beads. We used PyMOL software (v.2.3.3) to represent snapshots of polymer chain in Fig. 2a. Examples of initial conformations and simulation parameters can be find at [https://github.com/giorgettilab/Mach_et_al_chromosome_dynamics](https://github.com/giorgettilab/Mach_et_al_chromosome_dynamics), in the polymer simulations section.

To simulate the loop extrusion process, we developed and embedded in LAMMPS a package called 'USER-LE'. The loop extrusion model contains extruders and barriers on the polymer. An extruder is represented as an additional sliding bond, which extrudes the loop in a two-sided manner. It can be loaded to the polymer between (*i*) and (*i* + 2) beads with a certain probability only when the bead (*i* + 1) is unoccupied by another extruder and is not a barrier. Each extruder can be unloaded from polymer with a certain probability. Every bead can be occupied by only one extruder. Extruders cannot pass through each other. When extruders meet each other on the polymer, they stall until one of them is released. Every extruder attempts to make an extruding step every *N* simulation steps.

In addition to 'neutral' polymer beads, there are three types of barriers blocking loops coming from the left, from the right and from any direction. These barriers mimic CTCF sites, for which one can define a probability for the loop extruder to go through (the same probability for all barriers). To launch loop extrusion, one should define three fixes with LAMMPS syntax: loading, unloading and loop extrusion. Loading: frequency in number of steps to try to load extruders, types of beads, max distance to create, type of the bond (extruder) to be created, probability to create, seed for pseudorandom generator of numbers, new type of the first beads and new type for the second bead. Unloading: frequency in number of steps to try to unload extruders, type of the bond (extruder), min distance to release bond, probability to release bond, seed for pseudorandom number generator. Loop extrusion: frequency in number of steps to try to move extruders, neutral polymer type, left barrier type, right barrier type, probability to go

through the barrier, type of the bond (extruder) and type of two-sided barrier (optional).

## Statistics and reproducibility

No statistical method was used to predetermine sample size. No data were excluded from the analyses. No randomization was performed as the study did not require sample allocation into different groups. Live-cell imaging experiments were performed in 3–7 biological replicates and all replicates showed consistent results. For Capture-C, Hi-C, 4C-seq, piggyBac insertion site mapping and Nanopore sequencing with Cas9-guided adapter ligation, one biological replicate was performed. For flow cytometry measurements two biological replicates were performed. Western blot analysis and genotyping PCR with subsequent agarose gel electrophoresis were performed with 1–2 biological and 2 technical replicates. Blinding was not possible for data collection in live-cell imaging experiments, as data acquisition required identification of the sample for further processing. Data analysis for live-cell imaging, Capture-C, Hi-C, 4C-seq and piggyBac insertion site mapping were performed in a blinded manner. Blinding was not necessary for the other experiments since the results are quantitative and did not require subjective judgment or interpretation. Whenever Student's *t*-test was used, we formally verified the normality of distributions but assumed variance equality.

## Reporting summary

Further information on research design is available in the Nature Portfolio Reporting Summary linked to this article.

## Data availability

All Capture-C, Hi-C, 4C-seq and integration site mapping sequencing fastq files generated in this study have been uploaded to the Gene Expression Omnibus (GEO) under accession GSE197238. The following public database was used: BSgenome.Mmusculus.UCSC.mm9 (https://bioconductor.org/packages/release/data/annotation/html/BSgenome.Mmusculus.UCSC.mm9.html). The trajectories from imaging data can be found at https://doi.org/10.5281/zenodo.6627715. Source data are provided with this paper.

## Code availability

Custom codes generated in this study are available at: https://github.com/zhanyinx/SPT_analysis/ (image analysis); https://github.com/giorgettilab/Mach_et_al_chromosome_dynamics/ (4C-seq, Hi-C, nanopore, simulation analysis); https://github.com/polly-code/lammps_le (repository with loop extrusion module for the LAMMPS); and https://github.com/zhanyinx/hmmlearn (the modified version of hmmlearn).

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

## Acknowledgements

We thank all members of the Giorgetti laboratory for help with labeling images for training the deepBlink spot-detection algorithm and feedback on the manuscript; A. S. Hansen, L. Mirny and members of their laboratories (in particular S. Grosse-Holz) for useful discussions of the results; E. de Wit for sharing the WAPL-AID-eGFP cells; H. Kohler for assistance with flow cytometry experiments; and R. Grand and D. Schübeler for sharing single-molecule footprinting data. P.M. was supported by a Marie Skłodowska-Curie Innovative Training Network (grant no. 813327 'ChromDesign'). J.T. was supported by a Marie Skłodowska-Curie Innovative Training Network (grant no. 813282 'PEP-NET'). J.Z. was supported by a Marie Skłodowska-Curie grant (no. 748091 – 3DQuant). Research in the Giorgetti laboratory is funded by the Novartis Foundation, the European Research Council (grant no. 759366, 'BioMeTre'), Marie Skłodowska-Curie Innovative Training Networks (grant nos. 813327 'ChromDesign' and 813282 'PEP-NET') under the European Union's Horizon 2020 research and innovation program, and the Swiss National Science Foundation (grant no. 310030_192642).

## Author contributions

P.M. and P.I.K. conceived the study with L. Giorgetti and with the help of discussions with E.P.N. P.M. performed the experiments with help from J.T., J.C., S.G., J.Z. and M.K. P.I.K. developed the code and performed and analyzed polymer simulations with input from E.M. and G.T. Y.Z. and P.I.K. performed image and trajectory analysis with help from E.M., G. R. and G.T. L. Gelman and J.E. contributed to setting up and provided assistance with microscopy and image analysis. S.S. performed piggyBac insertion site mapping and provided assistance with high-throughput sequencing experiments. P.I.K. and Y.Z. analyzed sequencing data. L. Giorgetti wrote the paper with P.M., P.I.K., Y.Z. and G.T. and with input from all the authors.

## Competing interests

All authors declare no competing interests.

## Additional information

**Extended data** is available for this paper at https://doi.org/10.1038/s41588-022-01232-7.

**Correspondence and requests for materials** should be addressed to Luca Giorgetti.

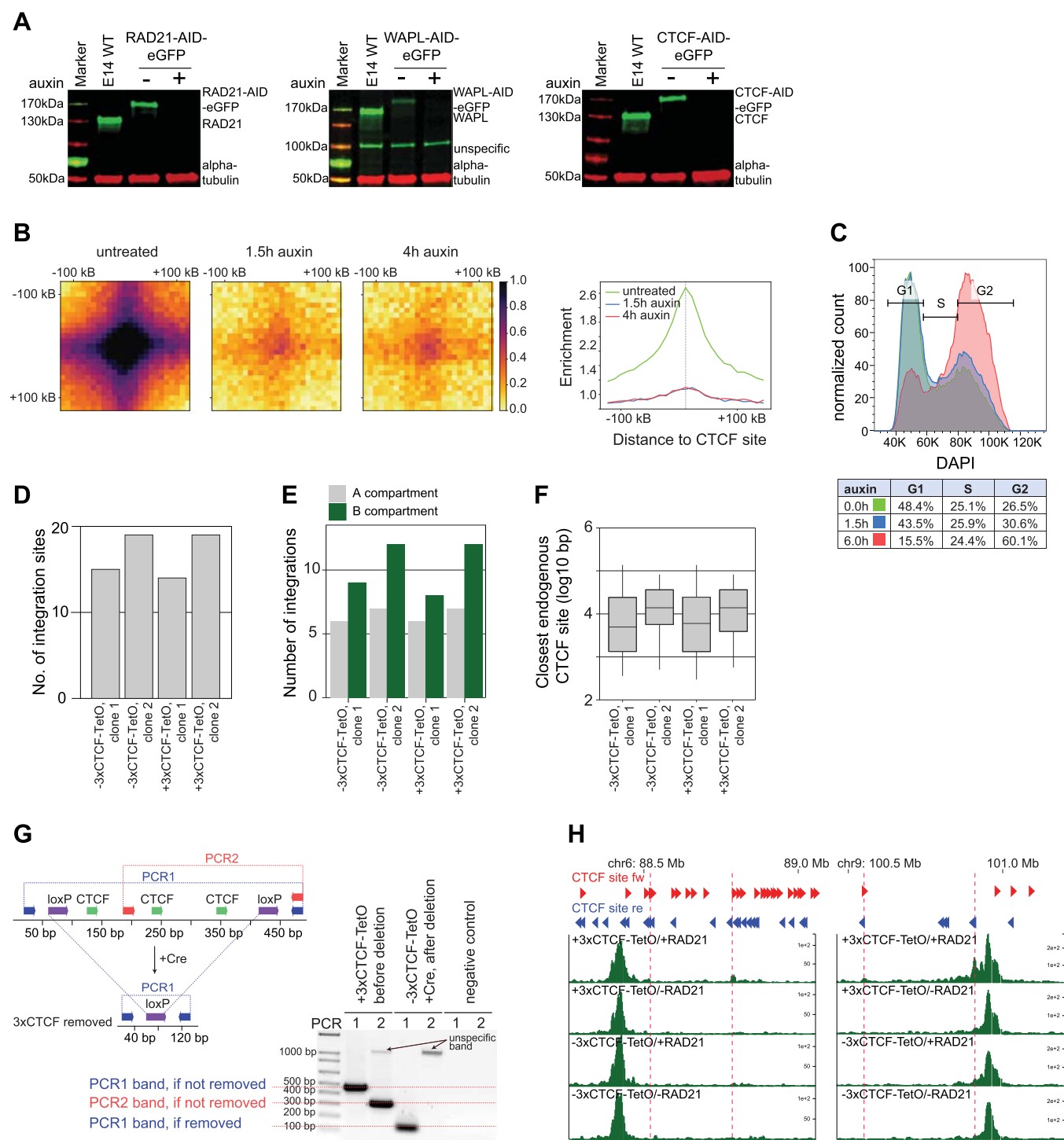

**Extended Data Fig. 1 | Chromosome structure is altered upon degradation of factors involved in loop extrusion. A**. Western Blots showing degradation of RAD21, WAPL and CTCF upon 1.5 h, 24 h and 6 h, respectively. Loading control: $\alpha$-tubulin, n = 1–2 replicates for each cell line. **B**. Left: Average enrichment in Hi-C read counts at CTCF sites based on Hi-C data in RAD21-AID-eGFP cells either untreated (left), treated for 1.5 h (middle) or 4 h (right) with auxin. Right: Differences in enrichment at CTCF peaks. Peaks were called on Hi-C data from untreated cells. **C**. Flow cytometry analysis of fixed cells stained with DAPI showing cell-cycle stage distributions of RAD21-AID-eGFP mESC cultured with serum, LIF and 2i, either before (green) or after 1.5 h (blue) and 6 h (red) auxin treatment. **D**. Integration site numbers in two clones of RAD21-AID-eGFP lines with and without 3xCTCF sites. **E**. Distribution of integration sites from lines shown in panel D that belong to A and B compartments called on distance-normalized Hi-C map (same as panel B). **F**. Integration sites distances from the closest endogenous CTCF site. Boxplot: lower and upper quartiles (Q1 and Q3, respectively); whiskers: 1.5x interquartile region (IQR) below Q1 and above Q3. n = 15 and 19 insertions for -3xCTCF-TetO clones 1 and 2, respectively, n = 14 and 19 insertions for +3xCTCF-TetO clones 1 and 2, respectively. **G**. Example of genotyping PCR upon removal of 3xCTCF sites in a RAD21-AID-eGFP +3xCTCF-TetO clonal line. PCR1 amplifies the entire 3xCTCF cassette and product size changes from 470 bp to 147 bp if the cassettes are successfully removed. PCR2 amplifies half of the 3xCTCF cassette and no product is expected if 3xCTCF cassettes were removed from all insertion sites; otherwise a PCR band of 303 bp is expected. **H**. Representative 4C-seq profiles from insertions on chromosomes 6 and 9 using TetO as a viewpoint showing that 3xCTCF-TetOs lead to the formation of ectopic contacts (dashed red lines) with nearby endogenous CTCF sites in the presence of RAD21. Contacts are lost upon deletion of 3xCTCF cassette (−3xCTCF-TetO) and upon degradation of RAD21 (−RAD21).

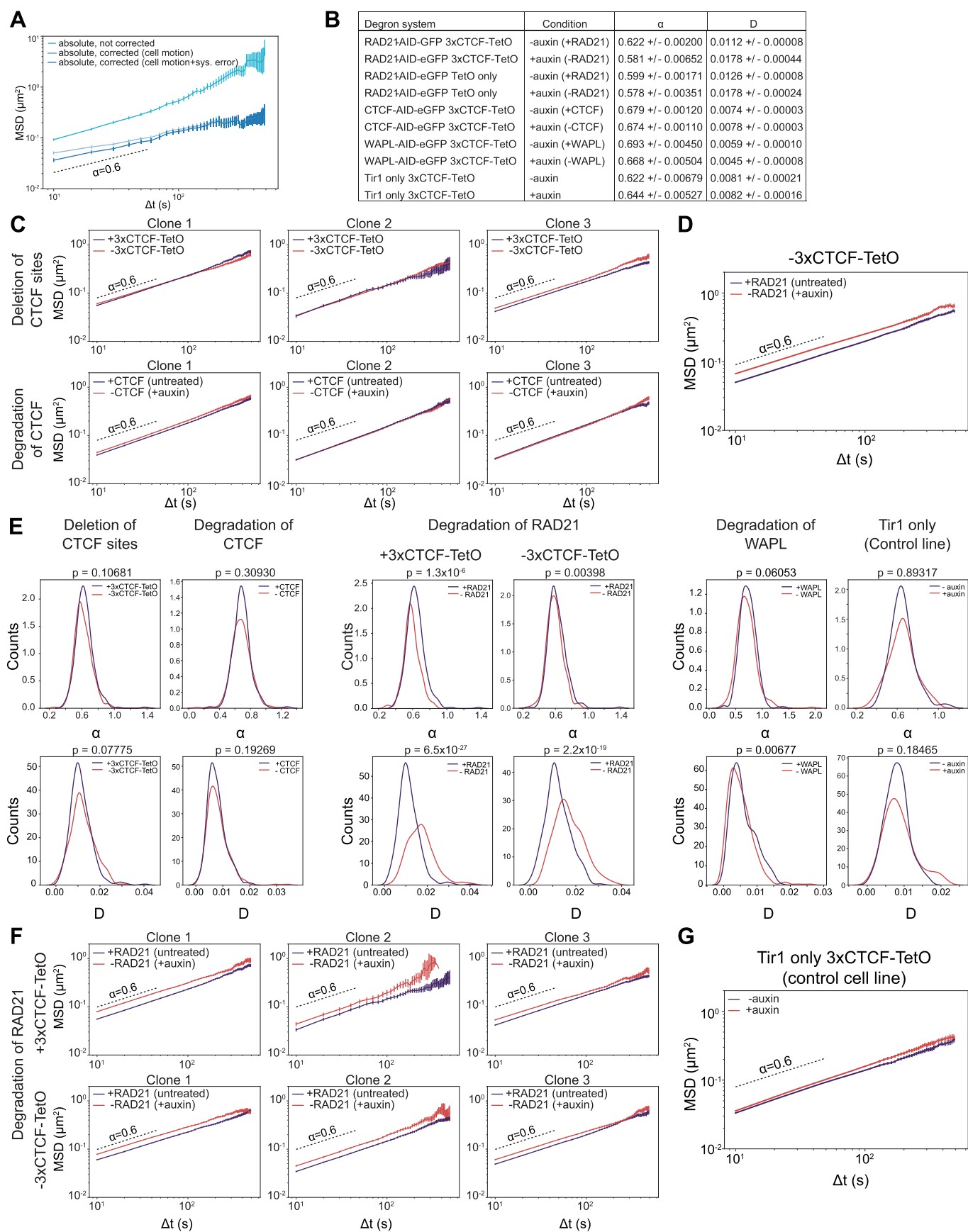

**Extended Data Fig. 2 | See next page for caption.**

**Extended Data Fig. 2 | Chromosome dynamics is modulated by degradation of factors involved in loop extrusion. A**. Mean Square Displacement (MSD) of trajectories from TetO insertions within the same cell (MSD, mean ± s.e.m., n = 45 tracks) before (cyan) and after applying cell motion (light blue, n = 45 tracks) and localisation error correction (dark blue, n = 45 tracks). **B**. Scaling exponents (α) and generalized diffusion coefficients (D) across all conditions and cell lines were fitted by pooling all three biological replicates. Shown are the numbers for the best fit ± error of the fit. **C**. MSD (mean ± s.e.m.) plots for a single clonal cell line (biological replicate) when looking at removal of 3xCTCF sites (top row) next to the array or degrading all CTCF (bottom row). **D**. MSD (mean ± s.e.m.) in the cell lines (n = 3 replicates per clonal cell line, three cell lines) where the 3xCTCF cassette was excised. Shown are the MSDs for cells either depleted of RAD21 for 90 min (red, 266 cells, 9,020 trajectories analyzed) or not (blue, 271 cells, 11,082

trajectories analyzed). Global depletion of RAD21 increases mobility. *p*-values in panel E. **E**. Distributions of α and D fitted based on single trajectory MSD and significance test for differences in generalized diffusion coefficients (D) and scaling exponents (α). The *p*-value is calculated using Student t-test (two-sided) (see Methods). **F**. Same as in C for a single clonal cell line (biological replicate) with integrations with 3xCTCF-TetO (top row) or without 3xCTCF-TetO (bottom row) when degrading RAD21. Global depletion of RAD21 increases mobility. **G**. Same as in D in the cell lines that contain integrations of 3xCTCF-TetO and the Tir1 protein, but do not contain any AID-tag for targeted degradation. MSDs for cells either treated with auxin for 90 min (red, 97 cells, 2,155 trajectories analyzed) or not (blue, 111 cells, 3,711 trajectories analyzed). No significant changes were detected. *p*-values in panel E.

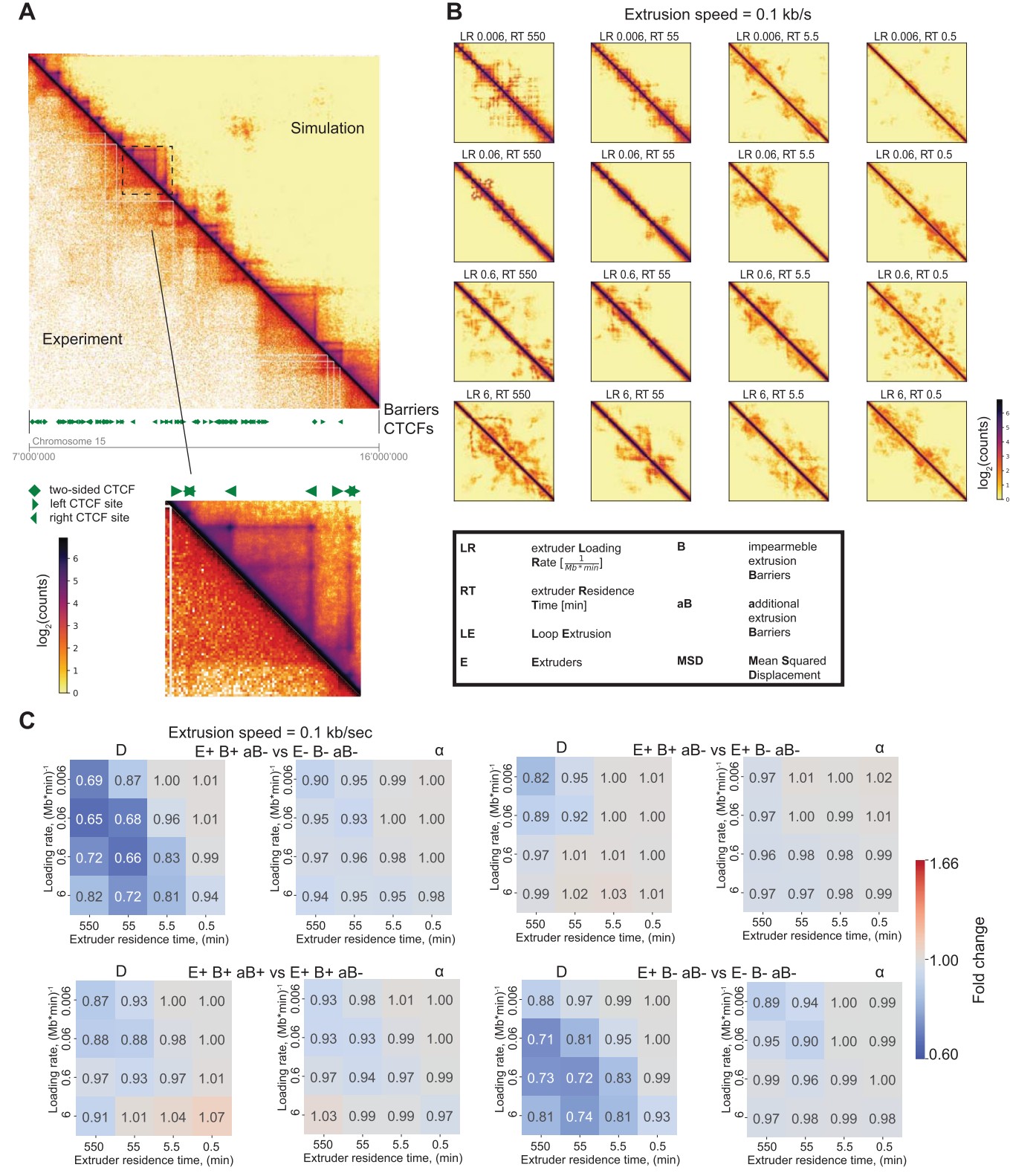

**Extended Data Fig. 3 | Simulations of chromosome dynamics and effects of loop extrusion. A**. Visual comparison of experimental Hi-C contact map with contact maps of simulations at extrusion speed 1 kb/s, extruder loading rate 0.06 (Mb x min)$^{-1}$ and residence time 5.5 min. **B**. Contact maps for the polymer simulations at extrusion speed 0.1 kb/s and barriers from the range 7–16 Mb of chromosome 15. Acronyms used in this figure are indicated in the black box on the right. **C**. Pairwise comparison for conditions indicated in the title of each pair of heatmaps. Pair of heatmaps contains ratios of generalized diffusion coefficients (D) and scaling exponent ($\alpha$), and represents fold change between the conditions.

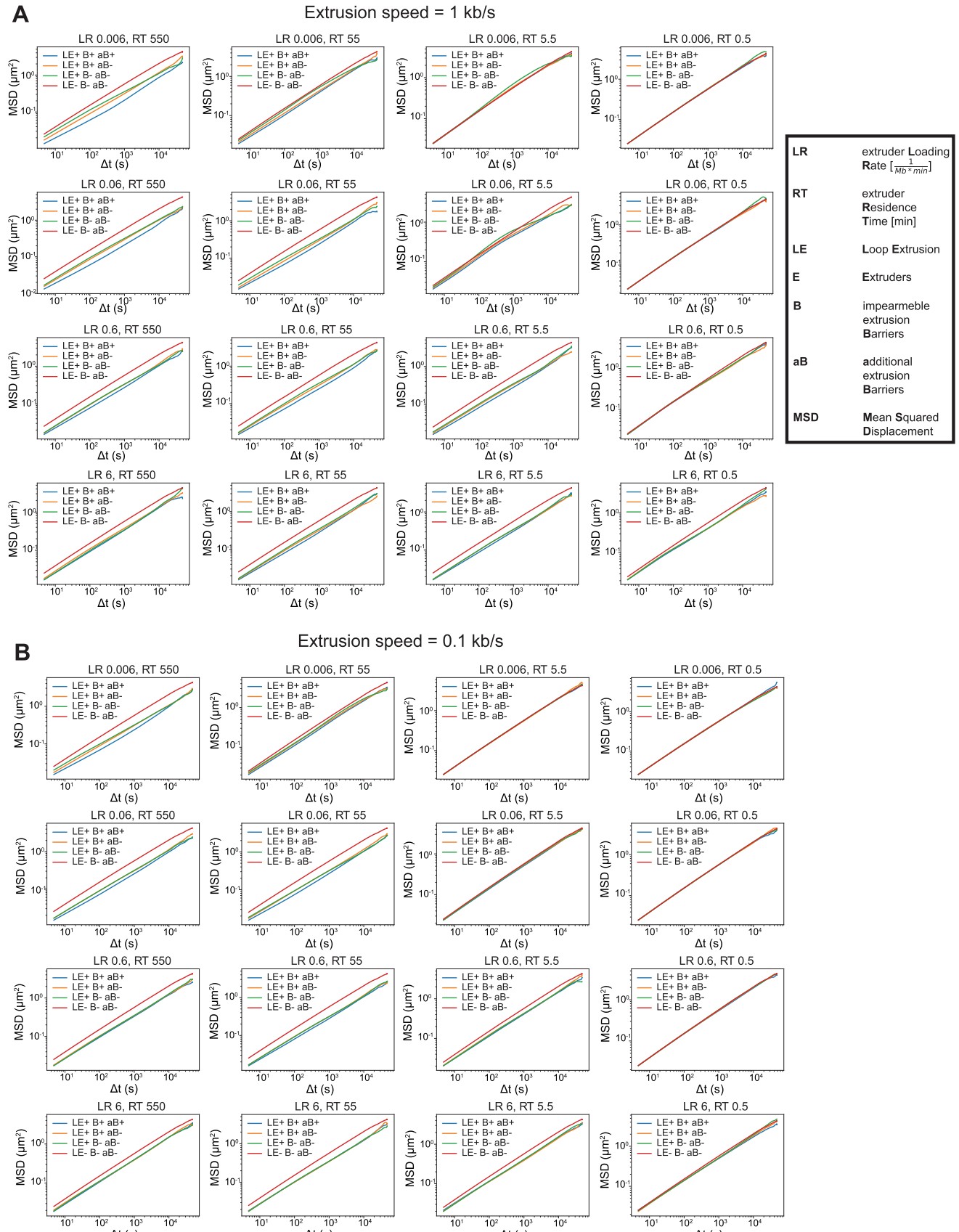

**Extended Data Fig. 4 | MSDs of systems for two extruder speeds. A.** MSDs for all 16 conditions for each set of loop extrusion parameters and extrusion speed of 1 kb/s. **B.** Same as A but for the extrusion speed of 0.1 kb/s.

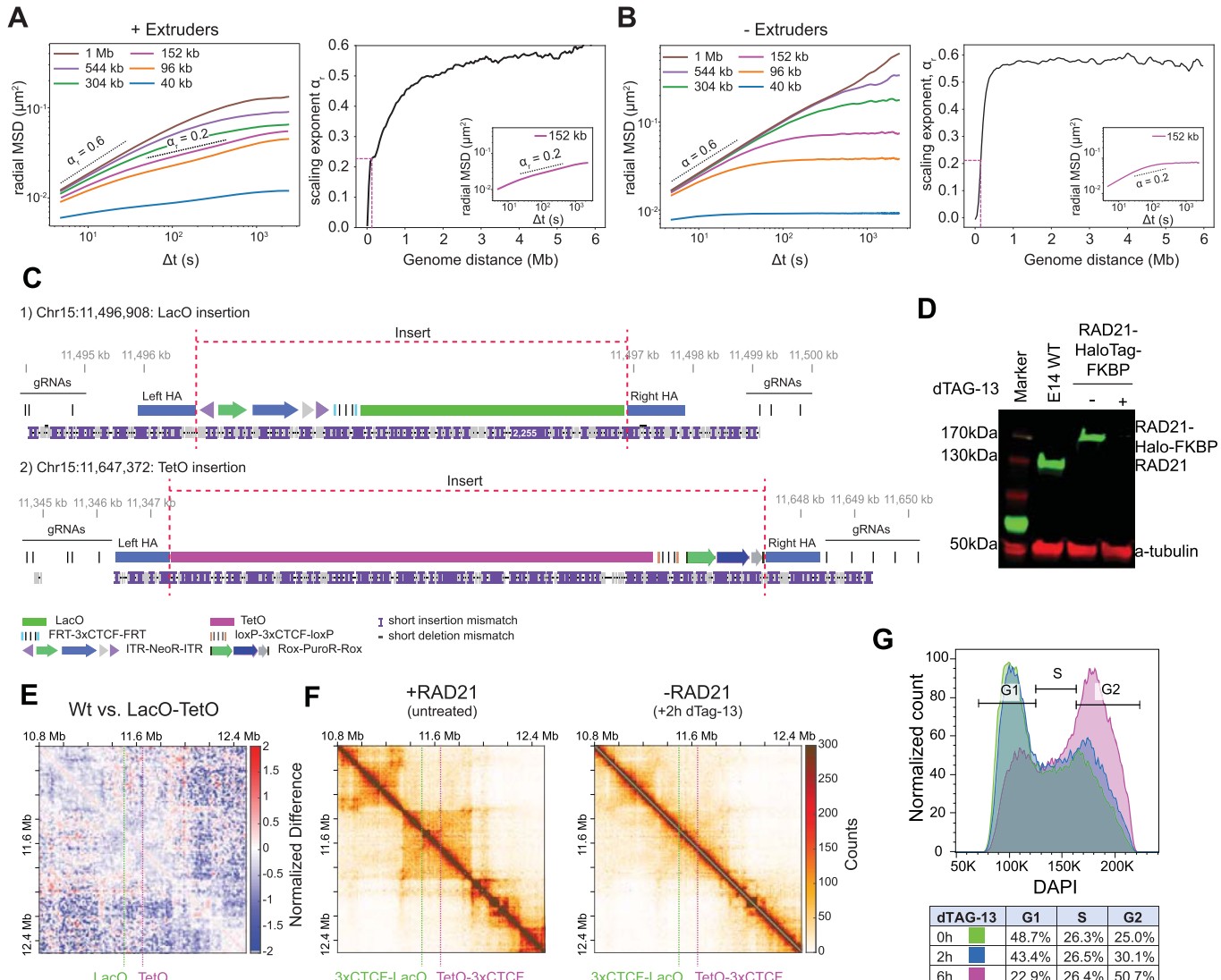

**Extended Data Fig. 5 | Characterization of TetO and LacO array integrations.**
**A**. Left panel: radial MSD of distances between multiple pairs of monomers separated by distances equivalent to 40 kb - 1 Mb for a polymer with loop extrusion but no barriers. Dashed scaling exponents $\alpha = 0.2$ and $\alpha = 0.6$ serve as an eye guide. Right panel: Slopes of radial MSD curves for two loci separated by varying linear distances, estimated from linear fitting between 5 and 60 seconds. Inset: detail of radial MSD and fit for monomers separated by 152 kb. **B**. Left panel: radial MSD of multiple pairs of monomers separated by various distances (40 kb - 1 Mb). Simulations were performed for the polymer without extruders and barriers. Values were averaged with a sliding window without considering the first and last 200 monomers (1.6 Mb). Dashed scaling exponent $\alpha = 0.6$ serves as an eye guide. Right panel: Distance dependency of the scaling exponent ($\alpha$) on the genomic distance between loci. **C**. Integrated Genomic Viewer (IGV) snapshot showing an example of a Nanopore sequencing read mapped to a modified mouse genome including the respective insertions. Reads that spanned from a guide RNA (gRNA) binding site upstream of the left homology arm (left HA) to a gRNA binding site downstream the right homology arm (right HA) confirmed single insertion of the transgene. **D**. Western Blots showing the targeted degradation of RAD21 after 2 h of treatment with 500 nM dTAG-13. Loading control: anti-tubulin, n = 2 replicates. **E**. Differential map at 6.4 kb resolution for the structural differences between a E14 wild-type (WT) and the E14 cell line containing LacO and TetO insertions (see Methods). Dashed lines indicate the insertion sites. No structural changes are detected upon integration of the operator arrays. **F**. Capture-C maps at 6.4 kb resolution in the region on chr15 (10.8 Mb-12.5 Mb) in the untreated cells (left) and in cells treated with 500 nM dTag-13 (left) showing that RAD21 degradation leads to loss of chromosome structure. **G**. Flow cytometry analysis of fixed cells stained with DAPI to show cell cycle stage distribution of E14 RAD21-HaloTag-FKBP cells.

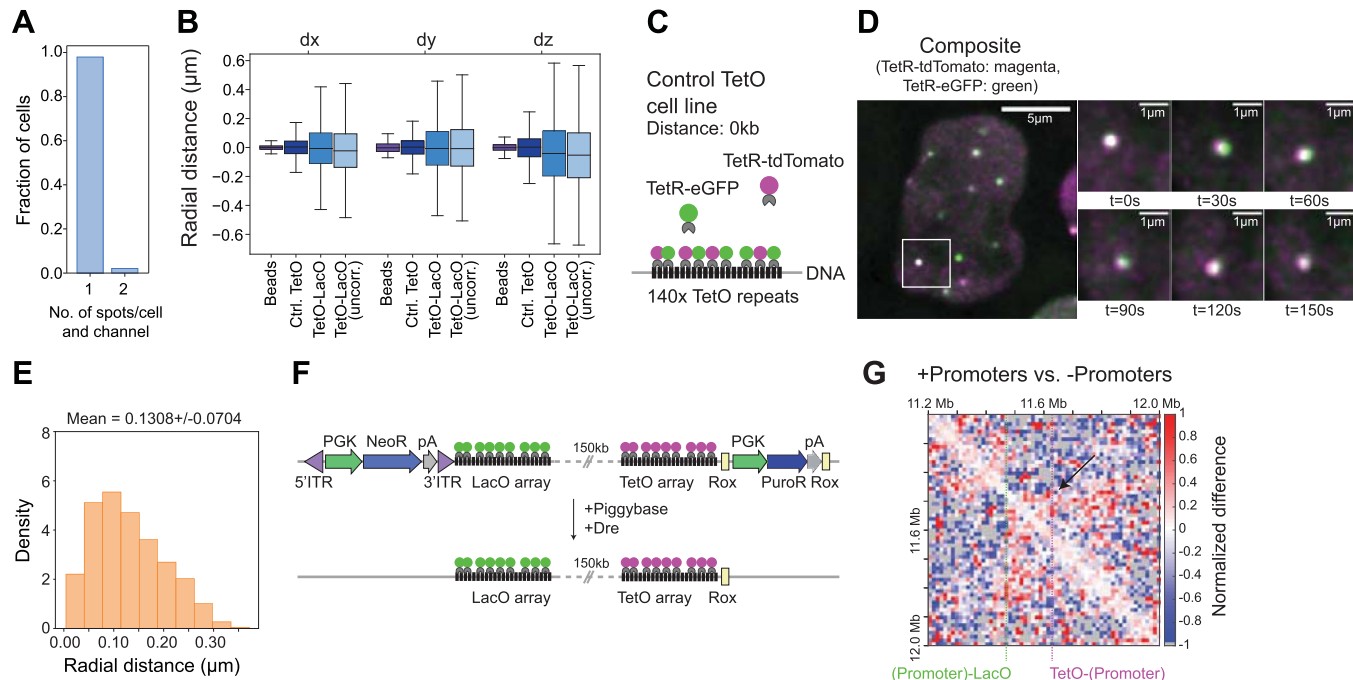

**Extended Data Fig. 6 | Correction of chromatic aberrations and characterization of mESC lines with promoters flanking TetO and LacO arrays. A.** Bar plot showing the number of detected spots per cell per channel for 1,400 manually annotated images subsampled from the images series. In 3% of the images 2 spots per cell are detected indicating the presence of sister-chromatids. **B.** Distribution of pairwise distances in each dimension for co-localized signals measured on beads (n = 2,226 timepoints) or on the control TetO cell line (n = 69,453 timepoints), as well as for chromatic-aberration corrected and uncorrected images from TetO-LacO cell lines (in the presence of cohesin and 3xCTCF sites, n = 848,955 timepoints). Boxplot: boxes denote lower and upper quartiles (Q1 and Q3, respectively); whiskers denote 1.5x the interquartile region (IQR) below Q1 and above Q3. **C.** Schematic representation of the 'Control TetO' cell line that contains multiple TetO array integrations as well as stable integrations of TetR-eGFP and TetR-tdTomato. This allows labeling of each TetO array with two separate fluorophores. **D.** Representative images of the 'Control TetO' cell line. The time series shows a zoomed version of the region indicated by the white square. **E.** Radial distance distribution of the 'Control TetO' cell line as defined in panel C and D showing that the resolution on the 3D distance is ~130 nm. **F.** Schematic representation of cell line containing 3-phosphoglycerate kinase (PGK) promoters driving the expression of resistance gene directly adjacent to the operator arrays. The expression cassettes can be excised using Dre recombination or piggyBac transposition to yield the cell line with operator arrays only (PGK = PGK promoter, NeoR = Neomycin resistance gene, PuroR = Puromycin resistance gene, pA = polyadenylation signal, ITR = inverted terminal repeats for piggyBac recognition, Rox = Rox sites for Dre recombination). **G.** Differential map at 6.4 kb resolution for the structural differences between the E14 cell line containing LacO and TetO insertions with the adjacent promoters vs. the E14 cell line containing the operator arrays only (see Methods). Dashed lines indicate the insertion sites.

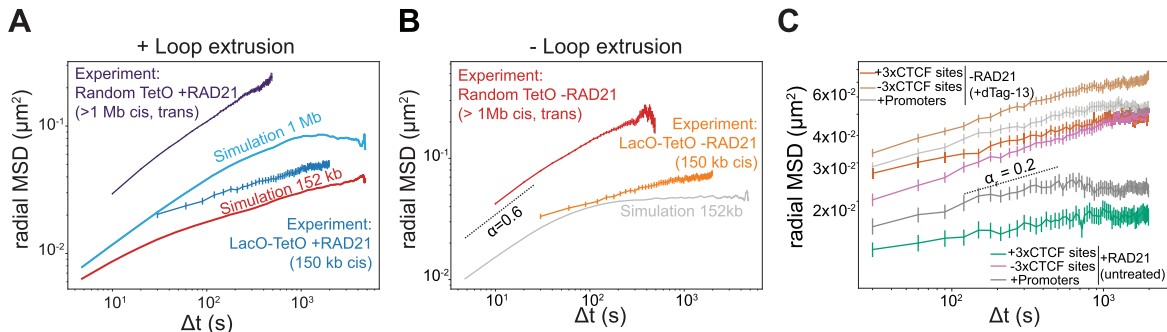

**Extended Data Fig. 7 | Polymer simulations of two genomic locations within the same TAD. A.** Radial MSD of TetO random integrations (mean ± s.e.m., purple, see Fig. 1E, n = 271 cells examined over 3 pooled biological replicates) and of targeted LacO and TetO insertions on Chr15 (mean ± s.e.m., dark blue, n = 214 cells examined over 4 replicates) are compared to model predictions for pairs of loci containing extrusion barriers at a distance of 1 Mb (light blue) and 152 kb (red). Note that random TetO insertions often occur on different chromosomes and thus have larger absolute radial MSD than 1 Mb simulations (but similar scaling). **B.** Radial MSD for cell lines containing multiple random integrations of TetO as shown in Extended Data Fig. 2D (mean ± s.e.m., red, 266 cells examined over 3 pooled replicates) or the targeted integrations of LacO and TetO on chr15 (mean ± s.e.m., orange, n = 277 cells examined over 6 replicates) in the absence of RAD21 compared to the predicted radial MSD of two loci at a distance of 150 kb in the absence of extruders (gray) as predicted from polymer simulations. **C.** Radial MSD of TetO-LacO distances in mESC lines with or without convergent 3xCTCF sites (or promoters, respectively), either before or after treatment with 500 nM dTag-13 for 2 hours to induce degradation of RAD21 (dt = 30 s). radial MSDs are plotted as mean ± s.e.m. over conditions: +CTCF sites/+RAD21: n = 152 cells examined over 4 replicates, −CTCF sites/+RAD21: n = 214 cells examined over 4 replicates, +CTCF sites/−RAD21: n = 248 cells examined over 7 replicates, −CTCF sites/−RAD21: n = 277 cells examined over 6 replicates, +Promoters/+RAD21: n = 155 cells examined over 3 replicates, +Promoters/−RAD21: n = 170 cells examined over 3 replicates.

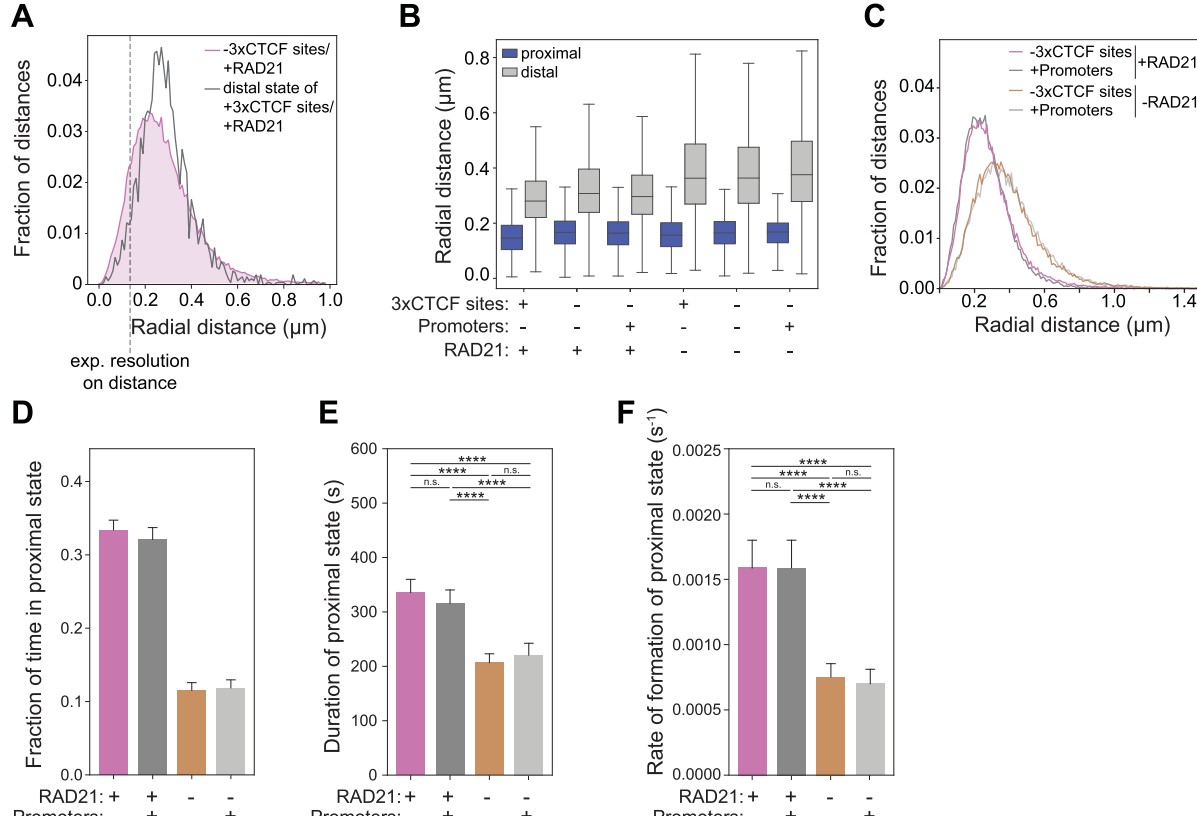

**Extended Data Fig. 8 | Live-cell imaging of two genomic locations within the same TAD. A.** Radial distance distribution for the condition -3xCTCF sites/+RAD21 (magenta) overlaid with the distal state called by HMM on the +3xCTCF sites/+RAD21 (gray) showing that the distal state identified by HMM largely overlaps with the distance distribution of the two loci in the absence of the CTCF sites. **B.** Boxplot for the radial distances for the proximal and distal state called by HMM on all six conditions. The horizontal line indicates the median. Box plots are as in Extended Data Fig. 1F. Boxplot: boxes denote lower and upper quartiles (Q1 and Q3, respectively); whiskers denote 1.5x the interquartile region (IQR) below Q1 and above Q3. **C.** Distribution of TetO-LacO radial distances in the four experimental conditions. −CTCF sites/+RAD21: n = 214 cells examined over 4 replicates, −CTCF sites/−RAD21: n = 277 cells examined over 6 replicates, +Promoters/+RAD21: n = 155 cells examined over 3 replicates, +Promoters/−RAD21: n = 170 cells examined over 3 replicates). **D.** Fraction of time

spent in the proximal state called by HMM in the four experimental conditions comparing +Promoters vs. -Promoters +/−RAD21 (no. of cells is as indicated in panel C). Shown average across experimental conditions and error bars represent bootstrapped (n = 10,000) standard deviations. **E.** Average duration of proximal states (mean ± 95% confidence interval, n = 680 cells (-promoter +RAD21); n = 466 cells (+promoter +RAD21); n = 268 cells (−promoter −RAD21); n = 253 cells (+promoter −RAD21)) for the conditions +Promoters vs. −Promoters, +/− RAD21. p-values (two-sided Kolmogorov-Smirnov): * – p < 0.05, ** – p < 0.01, *** – p < 0.001, **** – p < 0.0001. p-values can be found in Suppl. Table S2. **F.** Average rates of contact formation – time elapsed between the end of a proximal state and the beginning of the next (mean ± 95% confidence interval, n = 726 (-promoter +RAD21); n = 495 (+promoter +RAD21); n = 323 (−promoter −RAD21); n = 296 (+promoter −RAD21))) for the conditions +Promoters vs. −Promoters, +/−RAD21. p-values legend is as in panel E.

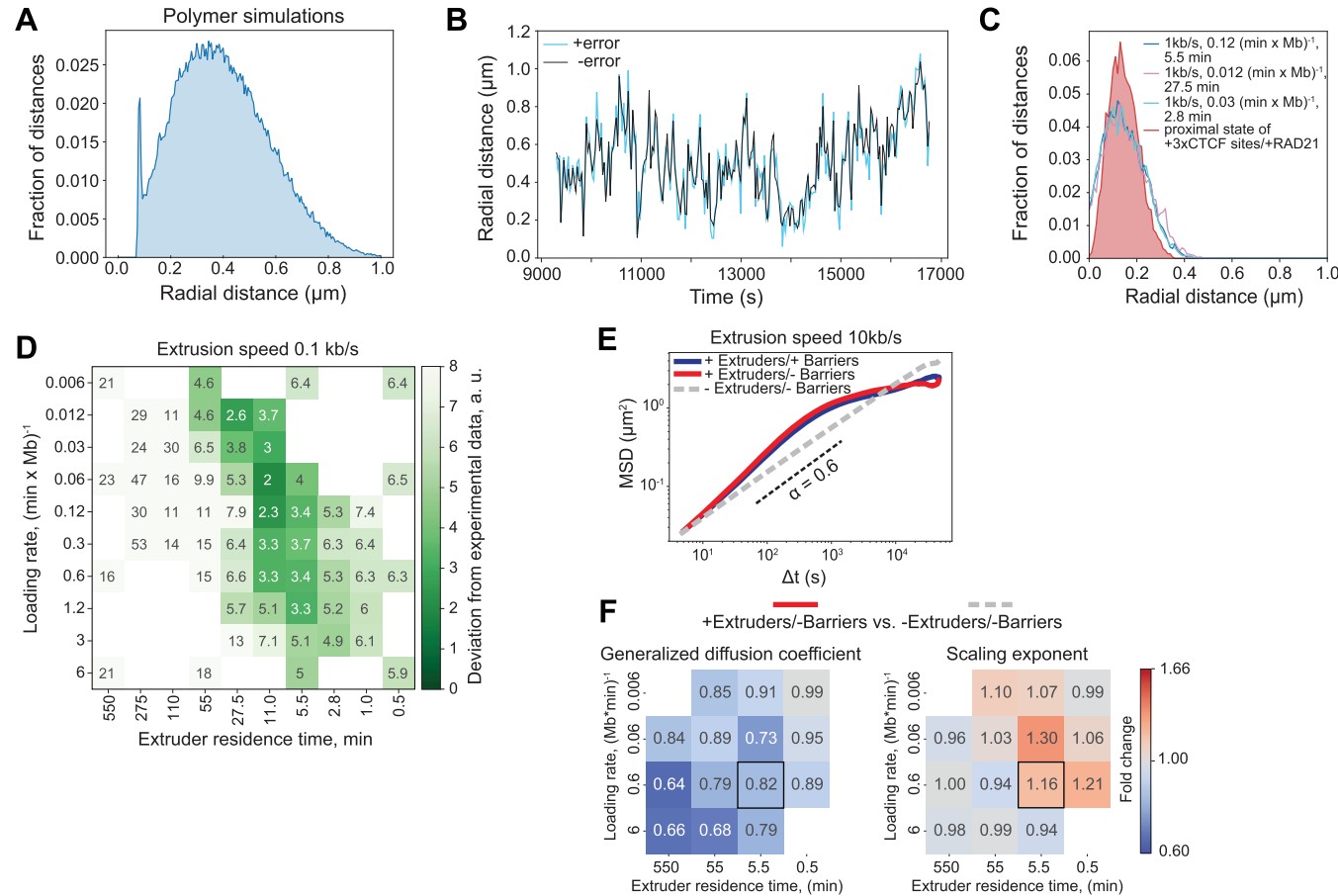

**Extended Data Fig. 9 | HMM analysis of simulations compared to experimental data. A.** Bimodal distribution of pairwise distances from simulations corresponding to the set of parameters with a loading rate of 0.06 (min × Mb)$^{-1}$, extruder residence time of 5.5 min, extruder speed of 1 kb/s, and in the absence of barriers. Data were sampled every 1 s and merged from 10 simulation runs. **B.** Representative radial distance trajectory of a simulated system with and without an additional error on the distance that is in the range of the experimental error. **C.** Radial distance distribution for the proximal state of the +3xCTCF sites/+RAD21 condition overlaid with the distributions of the proximal states from the three best matching parameters sets when comparing only the average radial distances. **D.** Heatmap showing the agreement of all simulated systems (for extrusion speed 0.1 kb/s) with the experimental data. The score is as described in Fig. 6A (see Methods). **E.** MSDs for three conditions for extruder residence time of 5.5 min, loading rate of 0.6 (Mb × min)$^{-1}$ and extrusion speed of 10 kb/s. Pairwise comparison for conditions indicated in the title of each pair of heatmaps. **F.** Heatmap showing the fold change of generalized diffusion coefficients (**D**) and scaling exponent ($\alpha$), and represents fold change between the conditions.

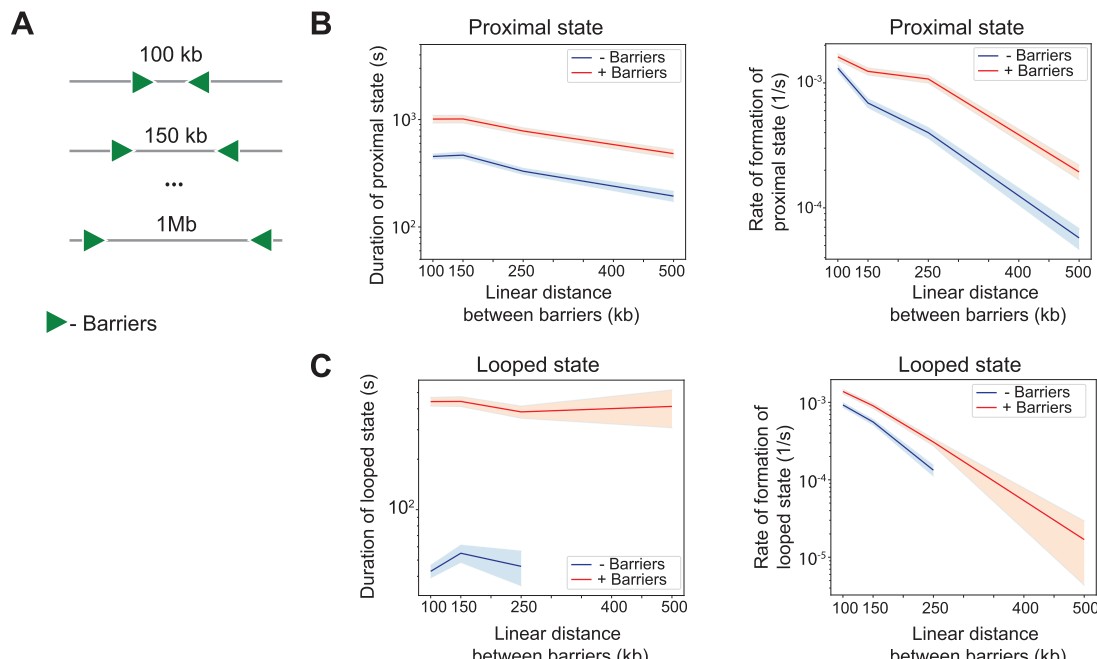

**Extended Data Fig. 10 | Polymer simulations of landscapes with two barriers at different distances. A**. Scheme of simulated polymers with varying distances between (optional) convergent loop extrusion barriers, corresponding to 100, 150, 250, 500, and 1000 kb. **B**. Duration (left) and rate of formation (right) of the HMM proximal state detected on simulated pairwise distances (after addition of experimental error) between monomers in the presence or absence of extrusion barriers, as a function of the intervening linear genomic distance. Lines are means, shaded areas are s.e.m. Note that the average duration of the HMM proximal state slightly decreases although the average duration of the underlying cohesin-mediated CTCF-CTCF interaction doesn't (see panel C). This is due to non-CTCF mediated interactions, which also contribute to the proximal state, and decrease with increasing genomic distance. **C**. Average duration (left) and rate of formation (right) of the looped state (that is cohesin-mediated CTCF-CTCF interaction) extracted from polymer simulations. Lines are means, shaded areas are s.e.m.

# Reporting Summary

## Statistics

For all statistical analyses, confirm that the following items are present in the figure legend, table legend, main text, or Methods section.

| n/a | Confirmed | |
|---|---|---|
| ☐ | ☒ | The exact sample size ($n$) for each experimental group/condition, given as a discrete number and unit of measurement |
| ☐ | ☒ | A statement on whether measurements were taken from distinct samples or whether the same sample was measured repeatedly |
| ☐ | ☒ | The statistical test(s) used AND whether they are one- or two-sided <br> *Only common tests should be described solely by name; describe more complex techniques in the Methods section.* |
| ☐ | ☒ | A description of all covariates tested |
| ☐ | ☒ | A description of any assumptions or corrections, such as tests of normality and adjustment for multiple comparisons |
| ☐ | ☒ | A full description of the statistical parameters including central tendency (e.g. means) or other basic estimates (e.g. regression coefficient) AND variation (e.g. standard deviation) or associated estimates of uncertainty (e.g. confidence intervals) |
| ☐ | ☒ | For null hypothesis testing, the test statistic (e.g. $F$, $t$, $r$) with confidence intervals, effect sizes, degrees of freedom and $P$ value noted <br> *Give P values as exact values whenever suitable.* |
| ☒ | ☐ | For Bayesian analysis, information on the choice of priors and Markov chain Monte Carlo settings |
| ☒ | ☐ | For hierarchical and complex designs, identification of the appropriate level for tests and full reporting of outcomes |
| ☒ | ☐ | Estimates of effect sizes (e.g. Cohen's $d$, Pearson's $r$), indicating how they were calculated |

*Our web collection on statistics for biologists contains articles on many of the points above.*

## Software and code

Policy information about availability of computer code

| Data collection | BD LSRII SORP Analyser was used for acquiring DAPI intensity by flow cytometry, BD Influx cell sorter was used for the FACS, MinION (protocol: SQL-CAS109) was used for Nanopore sequencing, Illumina Nextseq500 platform was used for Hi-C, Capture-C and integration site mapping, Illumina Hiseq2500 platform was used for 4C-seq. Odyssey infrared imaging system (Li-Cor Biosciences) was used for imaging Western Blot membranes. Live-cell imaging was performed on a Nikon Eclipse Ti-E inverted wide-field microscope with a Total Internal Reflection Microscopy iLAS2 module. Agarose gels were visualized using a Typhoon FLA 9500 scanner (GE Healthcare). |
|---|---|
| Data analysis | minimap2 (v. 2.17-r941), Snakemake (v. 3.13.3), IGV (v. 2.9.4), HiC-Pro (v. 2.11.4 for capture analysis v. 3.1.0 for Hi-C), Fiji (v. 2.0), TrackMate (v. 6.0.0), Trackpy (v 0.5.0), CellPose (v 0.6.5), deepBlink (0.1.1), scipy (v.1.4.1), Mustache(v. 1.0.1), coolpup.py (v. 0.9.2), PyMOL (v. 2.3.3), FlowJo (v10, BD Biosciences), BD FACSDiva (v8.0.1., BD Biosciences), Visiview (4.4.0.12, Visitron), Huygens Remote Manager (v3.8), QuasR (v1.36.0), csaw (v 1.30.1), GenomicRanges (v 1.48.0) <br> Custom codes can be found in <br> https://github.com/zhanyinx/SPT_analysis/ <br> https://github.com/polly-code/lammps_le/ <br> https://github.com/giorgettilab/Mach_et_al_chromosome_dynamics/ <br> https://github.com/zhanyinx/hmmlearn/ |

For manuscripts utilizing custom algorithms or software that are central to the research but not yet described in published literature, software must be made available to editors and reviewers. We strongly encourage code deposition in a community repository (e.g. GitHub). See the Nature Portfolio guidelines for submitting code & software for further information.

# Data

Policy information about availability of data

All manuscripts must include a data availability statement. This statement should provide the following information, where applicable:

- Accession codes, unique identifiers, or web links for publicly available datasets
- A description of any restrictions on data availability
- For clinical datasets or third party data, please ensure that the statement adheres to our policy

The image tracking data was uploaded to Zenodo (https://doi.org/10.5281/zenodo.7127868). All capture-C, Hi-C, 4C, integration site mapping sequencing fastq files generated in this study have been uploaded to the Gene Expression Omnibus (GEO) under accession GSE197238 (https://www.ncbi.nlm.nih.gov/geo/query/acc.cgi?acc=GSE197238). The following public databases were used: BSgenome.Mmusculus.UCSC.mm9 (https://bioconductor.org/packages/release/data/annotation/html/BSgenome.Mmusculus.UCSC.mm9.html ), Supplemental Information to Nora, Elphège P., et al. "Targeted degradation of CTCF decouples local insulation of chromosome domains from genomic compartmentalization." Cell 169.5 (2017): 930-944. Data for all plots for the Figures and Extended Data shown as well as original image files for gels and blots can be found in the Source Data section.

# Field-specific reporting

Please select the one below that is the best fit for your research. If you are not sure, read the appropriate sections before making your selection.

☒ Life sciences ☐ Behavioural & social sciences ☐ Ecological, evolutionary & environmental sciences

For a reference copy of the document with all sections, see nature.com/documents/nr-reporting-summary-flat.pdf

# Life sciences study design

All studies must disclose on these points even when the disclosure is negative.

| | |
|---|---|
| Sample size | No statistical methods were applied to predetermine sample size for live-cell imaging experiments. For live-cell imaging experiments of random TetO integrations, 3-4 biological replicates were performed with 1-2 technical replicates each. This resulted on average in 267 cells with 8402 trajectories analyzed per condition. For live-cell imaging of the dual-array cell lines, 3-7 biological replicates including 4 technical replicates each were performed resulting in on average 220 cells/condition.  For capture-C, Hi-C, 4C-seq, piggybac insertion site mapping and Nanopore sequencing with Cas9-guided adapter ligation 1 biological replicate was performed following the standard in the field on no. of reads sequenced. For flow cytometry measurements 2 biology replicates were performed recording >50,000 events for each condition. Western Blot analysis and genotyping PCR with subsequent agarose gel electrophoresis was performed with 1-2 biological and 2 technical replicates. Number of replicates was chosen based on standards in the field. |
| Data exclusions | No data was excluded from the analysis. |
| Replication | Live-cell imaging experiments were performed in 3-7 biological replicates and all replicates showed consistent results. For capture-C, Hi-C, 4C-seq, piggybac insertion site mapping and Nanopore sequencing with Cas9-guided adapter ligation 1 biological replicate was performed. For flow cytometry measurements 2 biology replicates were performed. Western Blot analysis and genotyping PCR with subsequent agarose gel electrophoresis was performed with 1-2 biological and 2 technical replicates. |
| Randomization | No randomization was performed as the study did not require sample allocation into different groups. Experimental groups were defined by the genotype of the cell line used and samples, i.e. cells measured, were chosen at random. |
| Blinding | Blinding was not possible for data collection in live-cell imaging experiments, as data acquisition required identification of the sample for further processing. Data analysis for live-cell imaging, capture-C, Hi-C, 4C-seq and Piggybac insertion site mapping were performed in a blinded manner. Blinding was not necessary for the other experiments since the results are quantitative and did not require subjective judgment or interpretation. |

# Reporting for specific materials, systems and methods

We require information from authors about some types of materials, experimental systems and methods used in many studies. Here, indicate whether each material, system or method listed is relevant to your study. If you are not sure if a list item applies to your research, read the appropriate section before selecting a response.

## Materials & experimental systems

| n/a | Involved in the study |
|-----|----------------------|
| ☐ | ☒ Antibodies |
| ☐ | ☒ Eukaryotic cell lines |
| ☒ | ☐ Palaeontology and archaeology |
| ☒ | ☐ Animals and other organisms |
| ☒ | ☐ Human research participants |
| ☒ | ☐ Clinical data |
| ☒ | ☐ Dual use research of concern |

## Methods

| n/a | Involved in the study |
|-----|----------------------|
| ☒ | ☐ ChIP-seq |
| ☐ | ☒ Flow cytometry |
| ☒ | ☐ MRI-based neuroimaging |

# Antibodies

| Antibodies used | rabbit polyclonal anti-CTCF antibody (Cat.No.: #2899, Lot:2, Cell signaling Technology)<br>rabbit polyclonal anti-WAPL antibody (Cat.No. 16370-1-AP, Lot: 00052432, Proteintech)<br>mouse monoclonal anti-alpha tubulin antibody (Cat. No. #3873, Lot: 15, Cell Signaling Technology)<br>rabbit polyclonal anti-Rad21 antibody (Cat.No. ab154769, Lot: GR3224138-1, Abcam) |
|-----------------|---|
| Validation | Anti-CTCF and anti-alpha-tubulin (CST): According to the manufacturer's website, both antibodies were validated for the use with mouse samples for Western Blotting. The manufacturer mentions the validation by SimpleChIP® Enzymatic Chromatin IP Kits for the anti-CTCF antibody. Anti-Rad21 (abcam): The manufacturer lists the antibody as being tested and suitable for the application in Western Blotting on mouse samples. Anti-WAPL (proteintech): The antibody was validated on samples from the same mouse cell lines in the following publication: Liu, N. Q. et al. "WAPL maintains a cohesin loading cycle to preserve cell-type-specific distal gene regulation." Nature Genetics 53, 100-109 (2021). Further validation is provided by this manuscript: The anti-CTCF, anti-Rad21 and anti-WAPL antibodies are validated by Western Blot in the CTCF/Rad21/WAPL-AID degron cell lines. In samples where the respective protein was degraded by IAA induction, no protein is detected, whereas in wild-type samples the protein is detected. |

# Eukaryotic cell lines

Policy information about cell lines

| Cell line source(s) | All cell lines for the dual-array imaging are based on E14 mouse embryonic stem cells (mESCs) provided by Edith Heard laboratory, EMBL, Heidelberg. E14 CTCF-AID-eGFP (clone EN52.9.1) were published in Nora, Elphège P., et al. "Targeted degradation of CTCF decouples local insulation of chromosome domains from genomic compartmentalization." Cell 169.5 (2017): 930-944. E14 WAPL-AID-eGFP and E14 RAD21-AID-eGFP Liu, N. Q. et al. "WAPL maintains a cohesin loading cycle to preserve cell-type-specific distal gene regulation." Nature Genetics 53, 100-109 (2021). E14 WAPL-AID-eGFP were provided by Elzo de Wit laboratory, NKI, Amsterdam. E14 CTCF-AID-eGFP and E14 Rad21-AID-eGFP were provided by laboratory of Elphège Nora, University of California, San Francisco. |
|---------------------|---|
| Authentication | Cell lines have been recurrently used by the authors in previous studies and therefore have not been authenticated. |
| Mycoplasma contamination | Cells were tested for mycoplasma contamination regularly and no contamination was detected. |
| Commonly misidentified lines<br>(See ICLAC register) | No commonly misidentified lines were used. |

# Flow Cytometry

## Plots

Confirm that:

☒ The axis labels state the marker and fluorochrome used (e.g. CD4-FITC).

☒ The axis scales are clearly visible. Include numbers along axes only for bottom left plot of group (a 'group' is an analysis of identical markers).

☒ All plots are contour plots with outliers or pseudocolor plots.

☒ A numerical value for number of cells or percentage (with statistics) is provided.

## Methodology

| Sample preparation | Cells were treated with either 500 µM auxin or 500 nM dTag-13 for the indicated time and then harvested with Accutase and re-suspend in 1x PBS. Cells were then fixed in 4% paraformaldehyde for 15 min at RT and stained with 5 µg/ml DAPI for 30 min at RT. |
|--------------------|---|
| Instrument | BD LSRII SORP Analyser (Becton Dickinson) |
| Software | BD FACSDiva™ Software v8.0.1, FlowJo (v10, BD Biosciences) |

| Cell population abundance | For each condition >50,000 cells were acquired. |
| --- | --- |
| Gating strategy | Forward scatter/Side scatter to discard big cells with high granularity; DAPI amplitude/DAPI height to discard doublets; DAPI amplitude/histogram to quantify cell cycle stage. |

☒ Tick this box to confirm that a figure exemplifying the gating strategy is provided in the Supplementary Information.

