## [Peer Review File · Nature Genetics]

Peer Review Information

Manuscript Title: Cohesin and CTCF control the dynamics of chromosome folding

Corresponding author name(s): Luca Giorgetti

Reviewer Comments & Decisions:

Decision Letter, initial version:
--

Dear Luca,

Your Article, entitled "Cohesin and CTCF control the dynamics of chromosome folding", has now been seen by 3 referees. You will see from their comments below that while they find your work of interest, some important points are raised. We are interested in the possibility of publishing your study in Nature Genetics, but would like to consider your response to these concerns in the form of a revised manuscript before we make a final decision on publication.

Reviewer #1 has submitted a very thoughtful and detailed review. They have some suggestions for improvement but these do not warrant further experiments.

Reviewer #2, like reviewer #1, thinks that this is high-quality work. However, while Reviewer #1 feels this is a landmark study and should be published soon, Reviewer #2 believes that most of the findings were previously reported or at least predicted, and therefore thinks that you should do more work to elevate the degree of novelty. For example, the reviewer suggests looking at the impact of transcriptional inhibition on looping dynamics.

Reviewer #3 is broadly positive about this work. They have some technical questions, and suggestions for further analysis but not necessarily new experiments.

As we discussed, a recent paper came out in Science from the Hansen group (ref. 21; bioRxiv version) that appears to have some overlapping findings. The reviewers did not highlight this as a major issue but we think that this merits a more detailed comparison/discussion.

We invite you to revise your manuscript taking into account all reviewer comments. While we encourage you to follow the reviewers' suggestions as much as possible, we think that a suitable revision could include only textual changes and additional in silico analyses, without any major wet lab experiments.

Please highlight all changes made in the manuscript text file. At this stage we will need you to upload a copy of the manuscript in MS Word .docx or similar editable format.

We are committed to providing a fair and constructive peer-review process. Do not hesitate to contact me if there are specific requests from the reviewers that you believe are technically impossible or

unlikely to yield a meaningful outcome. I would be happy to discuss the reviewers' comments in detail.

*2) If you have not done so already please begin to revise your manuscript so that it conforms to our Article format instructions, available [here](http://www.nature.com/ng/authors/article_types/index.html). Refer also to any guidelines provided in this letter.

[redacted]

We hope to receive your revised manuscript within 3 months. If you cannot send it within this time, please let us know.

Sincerely,

Tiago

Tiago Faial, PhD
Senior Editor
Nature Genetics
<https://orcid.org/0000-0003-0864-1200>

Reviewers' Comments:

Reviewer #1:
Remarks to the Author:
Overview

This work provides an estimation of the frequency and duration of chromatin interactions at one of the length scales most relevant to enhancer promoter communication, ~150 kb, both with and without CTCF anchors. While prior work has measured the 3D structure of the chromatin polymer at this length scale with greater detail and less measurement uncertainty through chromatin tracing in fixed cells, the time scales of the changes between these diverse structures is not known, but equally essential to our understanding of chromatin structure and cis-regulation. As such this work constitutes a major contribution to field of chromatin structure and cis-regulatory interactions.

On the whole I think the work is overall executed to a very high standard. I have some minor technical suggestions the authors may wish to keep in mind for future work, and some suggestions for improving the clarity of the presentation, and avoiding unnecessary confusion or dispute. In particular, I find claims in the text that, as I read them on the first pass, I thought not quite justified by the data. I believe the core problem is actually one of clarity, as described below. Rewriting these claims slightly will avoid unintentionally detracting from the value of the work. I support a speedy publication of this landmark work without delay of further experimentation.

Recommendations for improvement:

1) Introduction motivation "whether enhancer-promoter communication relies on stable physical contacts between regulatory sequences, or rather on stochastic and transient proximity events." I think this is a bit of a strawman motivation for the work, it unnecessarily weakens the appeal of the whole story.

Prior chromosome tracing experiments confirmed the predictions of dynamic models that TADs arise from complex ensembles. I do not believe anyone ever assumed these complex, highly diverse states represented stable structures. If the CTCF-loop or the TAD were to be stable complexes they would need to exist in all cells – not be structures that are stable the whole cell cycle long in 10% of cells and never occur in the others all cycle long. As with protein structures – if there is an extended

domain that bends back and forth a different way in each protein on the EM grid, the field assumes this is a 'floppy' 'labile' part of the protein that moves around dynamically with the protein, and can make transient contacts with other parts of its backbone – not that every protein is folded differently. Maybe these ensembles were all just rubbish, all different artefacts of the approach – except taken together, they recapitulate quite quantitatively the patterns seen in the Hi-C data. Thus, while in general ensembles snapshots of a group of different things don't suggest that those things ever move, I don't believe anyone was in serious doubt as to whether the interactions were truly "stable" vs "stochastic". Polymer physics models had been put forth to explain how either could happen (e.g. Nicodemi 2012, not to be confused with the later dynamic models of Nicodemi), along side dynamic ones, most notably based on loop extrusion. And despite some debate from the SIM data, the chromosome tracing data clearly supports the dynamic view of suggested by loop extrusion. Moreover, between all cohesin and CTCF degron experiments, the imaging experiments the loop extrusion model has received fairly widespread acceptance. Alternative models (Nicodemi) are equally built on stochastic, transient ensembles, even without invoking loop extrusion.

What isn't known, is how long, seconds, minutes, hours, that it takes for these states to interchange. This, in my opinion, is the more exciting, much more informative and detailed question the paper addresses, and the authors undermine this impact by talking about "stable" vs stochastic instead of real-time data.

2) I find the introduction to be unfair to prior work.

The authors do acknowledge that prior chromosome tracing data had argued the dynamic view, they neglect to cite some of the more recent work done at the much higher resolutions appropriate to enhancer promoter interactions. Specifically work on Sox2 at 5kb resolution, in both Mateo...Boettiger 2019 Nature, Huang...Ren Nat Gen 2021, fetal liver, also 5 kb res (Liu...Wang Nat Com 2020), not to mention yet higher resolution on abd-a/abd-B (Mateo...Boettiger Nature 2019, 2kb) and docs loci (Espinola...Nollmann Nat Gen 2021, 3kb). I think these works could be cited, and the motivation for measuring the lifetimes of these different contacts would be better established. Instead the current introduction proposes this question that no one really doubts (stochastic vs stable), and then has a dismissive sentence about a bunch of prior imaging work and skips over other more recent work entirely.

I also found it disappointing how the authors positioned their work related to other live imaging work in the field. This is an excellent work, I consider it to be a major advance, and I think it would sit better in the field if the authors could manage to work in a citation to other live imaging work. This could also help highlight some of the problems and complexities in the field that the authors work now addresses for the first time. For example there are notable similarities between the author's experiments and recent work by the Gregor and Jaynes labs (also published in Nat Gen incidentally, Chen...Gregor 2018). Both studies use live fluorescent imaging to study cis-interactions, both insert CTCF containing elements and show these elements mediate a difference in contact. (Yes, "homie" binds CTCF, and the insertion of homie at the site of the reporter gene, 140 kb distal from the 16 kb eve TAD, and it induces interactions with the CTCF marked borders of eve. Though it should be said it is likely in this Drosophila case that these CTCF marked sites depend more strongly on one or more of the other proteins that co-bind the regions, rather than CTCF alone, to induce these functions). There are noteworthy differences as well, such as tags of nascent RNA as a proxy for DNA position. The prior live cell work on Sox2 by Alexander and colleagues could also bear a citation (Alexander, eLife 2019). Both of those works have been used to support an 'action at a distance' interpretation. Whereas I think the careful controls done in the current work show that provocative interpretation to be a little

premature. At the very least these works lend support to the current understanding that spatial separations change dynamically, though few aside from Chen give a measure of the stability of the structure by failing to separate the noise/measurement error contributions from the physical structure as carefully as done here.

Finally, I think the description of the recent work from the Hansen lab could have been handled better. I found this sentence to be unfair as well "Recent analysis of a 500-kb CTCF loop connecting opposite TAD boundaries provided initial constraints on the duration and frequency of cohesin/CTCF mediated interactions. Yet the frequencies and durations of chromosomal interactions inside TADs and their relationship with the loop extrusion process have not been systematically investigated in living cells." As far as I can tell, both works do a similarly good job of putting constraints on the frequency and duration of cohesin/CTCF mediated interactions. Neither study is the last word as a systematic study of the loop extrusion process in living cells. This contrast is just setting your reader up for disappointment. Better to under-promise and over-deliver than over-sell and disappoint. There's no getting around the similarities of these two studies. I think the range of distances is helpful, and the range of loci is helpful, the field needs a lot more of this data. They clearly represent parallel work, they will most likely be largely cited together, and probably all-the-more frequently because there are two, and the whole field stronger for the reproducibility and depth of the pair, and will have more faith in the journals too. Just because it's published in C/N/S doesn't mean it's wrong, but the independent work is pretty sweet bonus all the same. Maybe the Hansen work would be better addressed in the discussion than in the intro.

I think it will take some work to get this introduction more exciting, fairer to prior work, and keep it brief and clear at the same time. But I think a work of this caliber deserves an even better introduction than what it has now.

3) Problematic claims about loop extrusion

"Thus solely relying on general estimates of realistic loop extrusion parameters and without any additional parameter fitting, a simple loop extrusion model with convergent barriers was able to reproduce the full range of dynamic properties observed in live-cell microscopy. This supports the notion that loop extrusion is largely responsible for our experimental observations"

I would recommend rewording this claim, it took me a long time to realize what the authors were actually claiming and I had an extensive list of further experiments and improvements to resolution that would be needed to claim what I thought the authors were claiming on the first pass. The introductory implication that this work would go vastly beyond the recent Hansen lab work to systematically measure in live cells the properties of loop extrusion ("loop extrusion process have not been systematically investigated in living cells") predisposed my misinterpretation of the claims. Below, I describe in more detail this confusion:

The core problem is it appears to be claiming a quantitative in vivo validation of loop extrusion and measurement of its kinetics, which is an over-interpretation of the data. On further reading/reflection of this ineffective wording, I think it is actually claiming almost nothing at all (I'll explain this below), which is to seriously under-sell the key results of the paper.

I think it would be much better to unpack what features of the data agree. If there was no measurement noise, perfect spatial resolution and temporal resolution, then the dynamical process in

which a motor protein walks processively down the chromatin extruding a loop should look different than a rouse polymer. It's not clear to me that any features of the data, the 'dynamic properties' observed, deviate significantly from the rouse polymer.

In any case it would be much more informative to show what models don't agree with the data, and to unpack what features of the data require what features of loop extrusion. For example the authors may do better to comment that, the fraction of time spent in the proximal state, the frequency it switches into that state, the duration of that state, and most importantly how and why those things change when CTCF is removed, cohesin is removed, or both are removed, is parsimoniously explained by loop extrusion. And equally important, is here measured for chromatin at the scale of the E-P interaction for the first time. With a note added in proof at the end that observations at the larger 500 kb distances, performed in parallel, are consistent with these data.

What the authors currently have written sounds off-puttingly grandiose, "all the dynamic microscopy data" and "a simple model of loop extrusion with no parameter optimization". Yet, it is actually, as a stand alone sentence, quite vacuous. One could say the same of the very first movies of single lacO sites. After all, there is nothing in "all that dynamic data" that is **inconsistent** with loop extrusion ((not very much dynamic data at all, but it is still all the dynamic data"). Sweeping conclusion of this nature have a tendency to sink in, and I think this one risks too much confusion.

I think if the authors were clearer about what they are not measuring and not concluding, it may also improve the work. For example, I see nothing in the data that rules out that the kinetics of chromatin motion are only those of the rouse polymer – a diffusive, energy minimizing process. There is no smoking gun that this is some energy burning, clearly motor driven, entropy reducing process to explain the data. The authors don't even show other non-extrusion models and how they would work. But I think they aren't actually making that claim (though I fear others may attribute such a claim given the wording of the current work). The time switching in and out of the proximal state, the duration in that state, these are real time frequency data that are useful whether or not they emerge as a result of loop extrusion. It is nice to know, but not very surprising given the limited spatial temporal resolution of the data, that these measurements are also consistent with loop extrusion. After all, the 150 kb separated loop anchors, which, according to the model, undergo multiple rounds of cohesin loading and unloading during the image window, are nearly indistinguishable, in the distribution of the 3D distances, from the fixed point dynamics, where the separation is driven by none of these extrusion dynamics. When the data has a substantial amount of measurement uncertainty relative to its dynamic range, and the simulation gets a similar amount of uncertainty built into it to capture the data it is not so surprising, nor conclusive, to say, look the data and model agree.

4) "Contact"

- Some of the field is going to get an over-interpreted idea that this work has measured the molecular scale interaction. I did appreciate the authors attempt to set this up as the "proximal state" before introducing contact as a short-hand, maybe it should not be used. On the other hand I'd much rather see these 140 nm distances interpreted as consistent with contact than as evidence of action at a distance, so maybe it should stay.

5) "Loop extrusion explains the decelerating effect of cohesin on chromosome dynamics"

- I found this assertion to be a bit too strong. I suspect the "deceleration" is actually just the consequence of having a many more short tethers distributed throughout the chromatin, so the fiber

has less wiggle room and move viscosity. It may be better to write "loop extrusion is consistent with". It would be interesting and straightforward to test alternative explanations by simulating the effect of adding tethers. If tethers alone are sufficient it would be premature to say the (active process of) loop extrusion per se that explains the observed dynamics.

6) "The finding that CTCF does not impact global chromosome mobility is at odds with 3C-based findings that sites bound by CTCF form site-specific interactions with increased contact probabilities."
 - I don't really think this is a true/fair representation of the 3C (Hi-C) based findings – the observation was that CTCF increases the interaction probability with adjacent CTCF sites, no one claimed it increases or affects interactions among one CTCF with all other CTCFs genome wide, or affected contact probabilities across the chromosome uniformly. Indeed the Hi-C normalization of "matrix balancing" explicitly assumes the opposite – by assuming 'equal visibility'. Here, the authors are introducing a very natural and very beautiful set of follow up experiments to their global analysis of random integration with a less than beautiful (confusing/misleading/simply false?) topic sentence.

7) Interestingly, distances in the proximal state inferred by HMM largely overlapped with those detected on perfectly colocalizing signals in control experiments (149 vs. 130 nm on average, respectively) (Fig. 5B right panel, cf. Suppl. Fig. S4K).

- This could be explained better – there is a really nice control here which deserves a word of explanation in the text, I would mention co-expression of tetR-tdTomato and TetR-eGFP.

Technical problems

1) Code availability

The github repositories on the Gioreggti Lab github site, listed in code availability do not appear to be publically available or privately accessible for review, though some analysis scripts referenced in other parts of the paper at the zhanyinx github site are accessible.

2) Data availability

I could find no mention of the tracking data deposition, which appear to be the core of the paper, though data for many peripheral experiments were deposited. If the data are not deposited yet, I recommend the authors use a repository such as Zenodo. I believe some Nature family journals also require this. In the opinion of this reviewer, these data should be available at review.

3) Fitting approach / Fit accuracy

Spot fitting is a generally a well posed physics problem where the knowledge of the wavelength, pixel size, NA, camera gain, dark current, photon flux, etc. add information to make this fit more accurate. There is a deep literature on the theory and a deep literature of the tools. Unlike, say, calling cell borders or something from microscopy images, I don't believe this is a good problem for deep learning off of images alone and I worry about the generalizability of the training sets. I know the authors have published it all already (in a work that I thought took a rather shallow view of the field and the tools). I'm not too worried about it in the context of this work, I think the images of the two-color Tet experiment already make it clear other sources of noise dominate, including the non-point-source nature of the tag.

4) Chromatic correction

The only description I found was this: "TetraSpeck™ Microspheres, 0.1 μm beads (Thermo Fisher Scientific, T7279) were imaged to allow for correction of chromatic aberrations during image processing and analysis", which I find inadequate as a description. Browsing the part of the github repository that was accessible it looks like the authors may be using an affine transform as the correction map. Maybe a second order polynomial might be better, it would at least be nice to know what was done, and how accurately that corrected the bead data (a plot would be awesome, but a number would be an improvement). Are the beads just on the coverglass, or are they distributed in 3D space? The chromatic aberrations typically have a significant dependence on the axial (z) position, which affects the correction in all dimensions, though this is not always taken into account. I think the field would improve if it were more standard practice to report these corrections at any rate. Though given the other sources of measurement error, even if it were possible that the aberration correction could be improved I do not think it would have a dramatic effect on the results. The chromatic aberration should be constant for a fixed position in the field of view, and from sup Fig 4, it looks like the offset changes.

5) Time scale conversions

"in order to convert the timescales between experiments and simulations, we measured the time needed for a bead (and 8 kb TetO array) to move for its own size ($95 \text{ nm} = 15 \text{ nm} \times \sqrt{40}$ = estimated size of a nucleosome x estimated number of nucleosomes in each 8kb segment, assuming that chromatin inside each 8kb behave as an ideal chain; this is in line with the previous estimations for a 3 kb segment [15])."

- Relation between simulation time scales and data: The authors have made a physicists reasonable back-of-the-envelope estimate based on some measured properties, but left out the corresponding back-of-the-envelope estimate of uncertainty. The problem of accurately and convincingly determining these time scale conversions is a project unto-itself, well beyond the scope of this paper, and certainly beyond the scope of the data presented here (I suspect the authors are themselves well aware of the limitation of their estimate, but to be concrete I will describe several issues below).

- An ideal chain of multiple segments and a bead don't behave the same. To model 8kb as 95 nm bead has a very different behavior than modeling it as a flexible chain. The high resolution FISH studies that the authors haven't cited (see comments about the intro) include measurements of the flexible configurations adopted by chromatin of this length, which sometimes has end to end distances of <50 nm and sometimes >300 nm as expected for a flexible chain.

- The movement measurements, given the relatively large time step (20s), the relatively large motion per time step (100 nm), the small total scales involved, and the substantial measurement uncertainty (0-300 nm spread for the repeat label), have substantial error bars.

- These are coarse grained langevin simulations. Given the lack of accurate measurements of the forces and energies involved and the large uncertainties on the measurements that are made, I am unconvinced the simulations "conversion to real time" is accurate within an order of magnitude or more.

- I think the whole field could benefit from a more careful treatment of the numbers. It is sloppiness like this that also leads the field to conclusion like "the enhancer and promoter are >200 nm apart, the cohesin ring is <40 nm, so it is impossible that these sequences are ever embraced in a cohesin ring". The language here should be softened, some acknowledgment of "estimation" and "approximation" should be given, and the take home impression should not be one of precision and cross-validation for an estimate made with so many problematic issues along the way. Such uncertainty will have little impact on the key conclusions of the effects of cohesin and ctcf on the frequencies/durations of switching in and out of the proximity state.

Issues with figures

1) Figure 2B

This systematic survey over parameters is nice. However, I expect to see smooth changes in the contact maps as a function of parameters like extrusion density, but I see little correspondence – indeed, I first thought these “contact maps” came from individual polymer examples. If the authors were to show population contact frequencies it would give a clear picture of the effects of the parameters on contact. The suggestion that these simulations are significantly undersampled (or maybe averaged correlated structures from the same simulation) has me concerned for the interpretation of the other results in this analysis as well.

2) Supp Fig S4L-M – is labeled “rMSD (um²)”. This is obviously a typo, rMSD being units of distance and MSD being distance squared. I believe the authors meant to write MSD, based on the actual unit measurements.

Also, the green lines in S4L and S4M to me both read as being the 150 cis +Rad21 +CTCF case, just juxtaposing that data against different comparison sets. But why then do the data look different? And why no error-bars in S4L? why do these green data lines have different max lengths for example?

Reviewer #2:

Remarks to the Author:

SUMMARY:

In this manuscript, authors used live imaging of genomic loci (labeled by TetO or LacI arrays) and physical modeling to dissect the role of Cohesin and CTCF in regulating chromosome dynamics. The main findings are 1) Cohesin loss increases the mobility of genomic loci globally with or without CTCF, consistent with the loop extrusion model prediction. 2) By imaging two loci within a neural TAD +/- a CTCF BE (boundary element), authors found that cohesin and CTCF acts together to constrain the radial mobility of chromatin 3) In addition, two distinct states were identified in CTCF BE(+) condition – a proximal state in which the 2 loci were less than 149nm apart close to the detection limit of the imaging and a distal state where 2 loci were separated by 288nm in average similar to the distance in the CTCF BE(-) condition. Using HHM to analyze the data in combination with physical modeling, they estimated that CTCF anchored loops last around 10mins in live cells. This reviewer really appreciates the interactive experimental and modeling approach taken by authors. The data quality is high. The experimental design/results are relatively straightforward. The main problem is that most results from the current study largely agree with previous loop exclusion simulation predictions or has been demonstrated (at least implied) previously (See detailed points below). Limited conceptual advances have been made. However, if authors are up for more challenges and would like to push their work further with the experimental suggestions below. I would be more than happy to review a revised manuscript.

Main points:

1) Cohesin loss eliminates TAD and loops but enhances compartmentalization. The loop extrusion model cannot explain the enhanced compartmentalization, suggesting additional mechanism must be at play. Regulatory activities within A and B compartments are fundamentally different. Authors here non-discriminatively analyzed loci within both active and inactive domains. One way to really advance

the fundamental understanding of the field is to study how chromatin dynamics in A/B compartment differs in both WT and Cohesin loss conditions. What if Cohesin loss have opposite effect on chromatin dynamics within A versus B? This is especially interesting when you consider recent emerging results showing that transcription factors with low complexity domains can form dynamic protein hubs in active domains. How would these type of regulations affect the looping or chromatin dynamics?

2) Another point is whether and how chromatin and looping dynamics is regulated by transcriptional activities. This point can be first tested by using chemical inhibitors (such as JQ1 and alpha amanitin). It would be even better if authors can target and track genes when they are in active and inactive states (MS2) and study whether Cohesin loss has different effect on chromatin mobility.

3) Authors always insist on is that their chromatin dynamics measurement reflects long-range transcriptional regulation. One main point from results in the past few years is that Cohesin-mediated genome organization and transcriptional regulation could be decoupled. Cohesin or CTCF loss only has minor effect on gene expression (PMID: 28985562). Conversely, transcriptional inhibition does not appear to significantly impact trans-chromosomal or long-range cis-chromosomal interactions (MID: 32616013). Thus, it is not appropriate for authors to equate chromatin looping mediated by Cohesin/CTCF to long-range transcription regulation. The loops could be more serving a structural role rather than a functional one. It is worth noting that high-resolution Micro-C experiment even suggests that enhancer-promoter interactions persist in the absence of Cohesin. It is important for authors to make the distinction in the text.

4) Does the size of TetO and LacO repeats affect chromatin dynamics? Authors should test this by decreasing and increasing the number of repeats in the array?

5) Previous loop extrusion simulation suggests that chromatin dynamics would increase due to the lack of constraint from Cohesin (PMID: 26538024; PMID: 29967174). Authors should discuss these papers.

6) Previous simulation and experimental results suggest that the lifetime of chromatin loops are in ~10s minute range (PMID: 28467304). Authors should discuss the paper.

7) The drift correction performed in the analysis (Figure 1D) is less than optimal. Specifically, one premise for using the ensemble average to correct is that the number of particles detected is the same at every time point. As soon as there are missing detections, particle trajectories will experience jitters and it fails. It is important for authors to show that they can consistently detect the same number of loci across time points.

Reviewer #3:

Remarks to the Author:

The manuscript of Mach et al. investigates the dynamics of loop extrusion using two-color time-lapse microscopy and computer simulations in a well-suited model system (mES cells). The study is timely and is highly interesting to a wide community interested in understanding chromosome structure, chromosome dynamics, and transcriptional regulation in eukaryotes. The study show that: (1) cohesin reduces the mobility of genomic loci, (2) this reduction is consistent with loop extrusion, (3) the action of loop-extrusion and of CTCF can modulate dynamic chromatin interactions within TADs, (4) presence of CTCF and cohesin increases the frequency CTCF-CTCF encounters, and their duration, (5) CTCF-CTCF loops within TADs can be quite stable (10-15') and occur frequently, at least when these loci are close in genomic distance (150kb).

The experiments are well described and analyzed, and the conclusions are well supported by data. However, I highlight a number of issues that should be addressed.

Major issues:

- The authors talk about 'deceleration' when referring to changes in dynamics in Figs. 1-2. I am not sure why they speak about deceleration when this term can be understood as a (negative) change in the derivative of the velocity. I don't think the authors measure acceleration in their figures.
- are the locations of the barriers used in the simulations (Fig. 2) the same as those cloned and characterized in Fig. 1?
- how were chromatic aberrations corrected? why are they so high provided that the signal-to-noise ratio from their images appears to be enough to reach nanometer localization precision? what limited further improvement? Is this limitation coming from spot detection? If it was the case, then the authors could use other localization methods to reduce the uncertainty in the measurement of radial distances, and thus detect more effectively CTCF loops, which in Fig. 6 are shown to occur less frequently and to be more short-lived than 'proximity events'.
- In Fig. 4F the authors use variance to characterize the distance explored by each trajectory, but wouldn't it be more interesting to perform jump analysis? remove baseline to reduce contribution of slow-rigid body motion of the two spots?
- why do you expect only two states in the HMM model used in Fig. 5 to determine proximal states? Presumably there are a continuous of polymer configurations? In Fig. 6, the authors show that the HMM model can be used to detect proximal states from simulations, but that these do not always correspond to looped states. Can the authors train the HMM model on the simulations to predict also looped states in their experimental data?
- Is the distribution of distances bimodal in Fig. 5A to justify use of 2 states in the HMM model?
- why does the rate of formation of the proximal state depends on the presence of 3xCTCF sites? (Fig. 5E) I would have expected only the duration of the proximal state to be affected. Evidence for one-sided looping? Can this be investigated computationally?
- Why does removal of barriers or extruders have a similar effect on the duration of proximal states? Does this indicate a basal looping ability of CTCF independent of cohesin?
- How do frequency and duration of CTCF loops change with genomic distance? The results in the paper (residence time/ frequency of encounters) should depend on the distance between CTCF sites. While 150kb is not a bad choice, convergent CTCF sites can be found at shorter distances and at much longer distances (e.g. loops between TAD borders). It would be useful to be able to extrapolate these results. Otherwise there is a risk that a reader would interpret that 'in average' or 'in general' CTCF loops can be relatively long lived (10-15') and occur several times during a cell cycle. These additional analysis would also be important as it would reflect the importance of how genomic distance can be used to modulate time-scales and frequencies of interaction.

Minor issues:

- What are the error bars in Figure 1I? what can be considered as statistically relevant change?

- Not clear what 'additional barriers' mean in zoom-in of Fig. 2A.
- What does "time needed for a monomer and TetO array to move their own sizes?" mean?
- Where is the definition of rMSD ?

Author Rebuttal to Initial comments

We thank the referees for their very constructive and helpful criticism. We have addressed their comments with new experiments and analyses. New and modified sentences in the main text have been marked in red in the revised manuscript.

In summary, these are the main changes:

1. We provide new live-cell measurements of looping dynamics in the presence of strong promoters flanking the LacO and TetO arrays (new **Suppl. Fig. S4,N and S5E-I**), and show that cohesin depletion has similar effects in the presence and absence of active transcription.
2. We report the results of jump analysis of dual-color looping experiments (new **Fig. 4G**).
3. We present a more exhaustive description of the correction of chromatic aberrations and the estimation of experimental uncertainty on 3D distances (new **Suppl. Fig. S4I**).
4. We provide new simulations describing how contact duration and frequency varies as a function of the genomic distance between convergent CTCF sites (new **Suppl. Fig. S8**).
5. We have rewritten the introduction and included citations to previous work that was inadvertently omitted in the previous version of the manuscript.
6. We rephrased several passages in the text to convey a more balanced view of the interpretation of the experimental data through polymer simulations.

Taken together, although the conclusions of the manuscript remain unchanged, we feel that our paper has been greatly improved thanks to the Reviewers' insightful suggestions.

Point-by-point responses to reviewers's comments

Reviewer #1:

Remarks to the Author:

Overview

This work provides an estimation of the frequency and duration of chromatin interactions at one of the length scales most relevant to enhancer promoter communication, ~150 kb, both with and without CTCF anchors. While prior work has measured the 3D structure of the chromatin polymer at this

length scale with greater detail and less measurement uncertainty through chromatin tracing in fixed cells, the time scales of the changes between these diverse structures is not known, but equally essential to our understanding of chromatin structure and cis-regulation. As such this work constitutes a major contribution to field of chromatin structure and cis-regulatory interactions.

On the whole I think the work is overall executed to a very high standard. I have some minor technical suggestions the authors may wish to keep in mind for future work, and some suggestions for improving the clarity of the presentation, and avoiding unnecessary confusion or dispute. In particular, I find claims in the text that, as I read them on the first pass, I thought not quite justified by the data. I believe the core problem is actually one of clarity, as described below. Rewriting these claims slightly will avoid unintentionally detracting from the value of the work. I support a speedy publication of this landmark work without delay of further experimentation.

We thank the Reviewer for their enthusiastic appreciation of our work, thorough assessment of our manuscript and insightful suggestions and advice.

Recommendations for improvement:

1) Introduction motivation “whether enhancer-promoter communication relies on stable physical contacts between regulatory sequences, or rather on stochastic and transient proximity events.” I think this is a bit of a strawman motivation for the work, it unnecessarily weakens the appeal of the whole story.

Prior chromosome tracing experiments confirmed the predictions of dynamic models that TADs arise from complex ensembles. I do not believe anyone ever assumed these complex, highly diverse states represented stable structures. If the CTCF-loop or the TAD were to be stable complexes they would need to exist in all cells – not be structures that are stable the whole cell cycle long in 10% of cells and never occur in the others all cycle long. As with protein structures – if there is an extended domain that bends back and forth a different way in each protein on the EM grid, the field assumes this is a ‘floppy’ ‘labile’ part of the protein that moves around dynamically with the protein, and can make transient contacts with other parts of its backbone – not that every protein is folded differently. Maybe these ensembles were all just rubbish, all different artefacts of the approach – except taken together, they recapitulate quite quantitatively the patterns seen in the Hi-C data. Thus, while in general ensembles snapshots of a group of different things don’t suggest that those things ever move, I don’t believe anyone was in serious doubt as to whether the interactions were truly “stable” vs “stochastic”. Polymer physics models had been put forth to explain how either could happen (e.g. Nicodemi 2012, not to be confused with the later dynamic models of Nicodemi), along side dynamic ones, most notably based on loop extrusion. And despite some debate from the SIM data, the chromosome tracing data clearly supports the dynamic view of suggested by loop extrusion. Moreover, between all cohesin and CTCF degron experiments, the imaging experiments the loop extrusion model has received fairly widespread acceptance. Alternative models (Nicodemi) are equally built on stochastic, transient ensembles, even without invoking loop extrusion.

What isn't known, is how long, seconds, minutes, hours, that it takes for these states to interchange. This, in my opinion, is the more exciting, much more informative and detailed question the paper addresses, and the authors undermine this impact by talking about "stable" vs stochastic instead of real-time data.

We completely agree with the Reviewer and apologize for the oversimplification. We indeed were among the first to suggest that chromosome structure within TADs is highly heterogeneous in single cells (Giorgetti et al, Cell 2014), and certainly shared the expectation that it should also be highly dynamic (see Tiana et al, Biophys J 2015). We agree with the Reviewer that this has been highly supported by tracing experiments and polymer models (see below) and have amended the introduction, abstract and references to convey a more balanced view of this topic. In the introduction, we have removed the sentence the Reviewer pointed at and replaced it with a more general statement that includes references to chromosome tracing papers and polymer simulations (page 2 in the revised manuscript).

2) I find the introduction to be unfair to prior work.

The authors do acknowledge that prior chromosome tracing data had argued the dynamic view, they neglect to cite some of the more recent work done at the much higher resolutions appropriate to enhancer promoter interactions. Specifically work on Sox2 at 5kb resolution, in both Mateo...Boettiger 2019 Nature, Huang...Ren Nat Gen 2021, fetal liver, also 5 kb res (Liu...Wang Nat Com 2020), not to mention yet higher resolution on abd-a/abd-B (Mateo...Boettiger Nature 2019, 2kb) and docs loci (Espinola...Nollmann Nat Gen 2021, 3kb). I think these works could be cited, and the motivation for measuring the lifetimes of these different contacts would be better established. Instead the current introduction proposes this question that no one really doubts (stochastic vs stable), and then has a dismissive sentence about a bunch of prior imaging work and skips over other more recent work entirely.

We indeed regret the lack of reference to chromosome tracing studies, which indeed contributed in a fundamental way to how we think about chromosome structure variability in single cells. We hope that the revised version of the introduction gives a more comprehensive view of the state of the art in the field.

I also found it disappointing how the authors positioned their work related to other live imaging work in the field. This is an excellent work, I consider it to be a major advance, and I think it would sit better in the field if the authors could manage to work in a citation to other live imaging work. This could also help highlight some of the problems and complexities in the field that the authors work now addresses for the first time. For example there are notable similarities between the author's experiments and recent work by the Gregor and Jaynes labs (also published in Nat Gen incidentally, Chen...Gregor 2018). Both studies use live fluorescent imaging to study cis-interactions, both insert CTCF containing elements and show these elements mediate a difference in contact. (Yes, "homie" binds CTCF, and the insertion of homie at the site of the reporter gene, 140 kb distal from the 16 kb eve TAD, and it induces interactions with the CTCF marked borders of eve. Though it should be said it is likely in this

Drosophila case that these CTCF marked sites depend more strongly on one or more of the other proteins that co-bind the regions, rather than CTCF alone, to induce these functions). There are noteworthy differences as well, such as tags of nascent RNA as a proxy for DNA position. The prior live cell work on Sox2 by Alexander and colleagues could also bear a citation (Alexander, eLife 2019). Both of those works have been used to support an 'action at a distance' interpretation. Whereas I think the careful controls done in the current work show that provocative interpretation to be a little premature. At the very least these works lend support to the current understanding that spatial separations change dynamically, though few aside from Chen give a measure of the stability of the structure by failing to separate the noise/measurement error contributions from the physical structure as carefully as done here.

We apologize for failing to acknowledge the Chen et al. and Alexander et al. papers. We now cite these studies in the introduction when describing previous evidence that chromosome structure is dynamic. In the discussion, we now also highlight that while the CTCF effect we observe is similar to that mediated by *homie* pairing in *Drosophila* (page 10 in the revised manuscript), our study shows that this is compatible with stalling of loop extrusion at CTCF sites. We have also attempted to emphasize that our ability to detect contacts within a certain spatial range is limited by residual chromatic aberrations that survive computational correction (new text at page 10, second paragraph).

Finally, I think the description of the recent work from the Hansen lab could have been handled better. I found this sentence to be unfair as well "Recent analysis of a 500-kb CTCF loop connecting opposite TAD boundaries provided initial constraints on the duration and frequency of cohesin/CTCF mediated interactions. Yet the frequencies and durations of chromosomal interactions inside TADs and their relationship with the loop extrusion process have not been systematically investigated in living cells." As far as I can tell, both works do a similarly good job of putting constraints on the frequency and duration of cohesin/CTCF mediated interactions. Neither study is the last word as a systematic study of the loop extrusion process in living cells. This contrast is just setting your reader up for disappointment. Better to under-promise and over-deliver than over-sell and disappoint. There's no getting around the similarities of these two studies. I think the range of distances is helpful, and the range of loci is helpful, the field needs a lot more of this data. They clearly represent parallel work, they will most likely be largely cited together, and probably all-the-more frequently because there are two, and the whole field stronger for the reproducibility and depth of the pair, and will have more faith in the journals too. Just because it's published in C/N/S doesn't mean it's wrong, but the independent work is pretty sweet bonus all the same. Maybe the Hansen work would be better addressed in the discussion than in the intro.

I think it will take some work to get this introduction more exciting, fairer to prior work, and keep it brief and clear at the same time. But I think a work of this caliber deserves an even better introduction than what it has now.

We have substantially rewritten the introduction and hope that the Reviewer agrees that it now presents a more balanced and fairer account of previous work.

3) Problematic claims about loop extrusion

“Thus solely relying on general estimates of realistic loop extrusion parameters and without any additional parameter fitting, a simple loop extrusion model with convergent barriers was able to reproduce the full range of dynamic properties observed in live-cell microscopy. This supports the notion that loop extrusion is largely responsible for our experimental observations”

I would recommend rewording this claim, it took me a long time to realize what the authors were actually claiming and I had an extensive list of further experiments and improvements to resolution that would be needed to claim what I thought the authors were claiming on the first pass. The introductory implication that this work would go vastly beyond the recent Hansen lab work to systematically measure in live cells the properties of loop extrusion (“loop extrusion process have not been systematically investigated in living cells”) predisposed my misinterpretation of the claims. Below, I describe in more detail this confusion:

The core problem is it appears to be claiming a quantitative in vivo validation of loop extrusion and measurement of its kinetics, which is an over-interpretation of the data. On further reading/reflection of this ineffective wording, I think it is actually claiming almost nothing at all (I'll explain this below), which is to seriously under-sell the key results of the paper.

I think it would be much better to unpack what features of the data agree. If there was no measurement noise, perfect spatial resolution and temporal resolution, then the dynamical process in which a motor protein walks processively down the chromatin extruding a loop should look different than a rouse polymer. It's not clear to me that any features of the data, the 'dynamic properties' observed, deviate significantly from the rouse polymer.

In any case it would be much more informative to show what models don't agree with the data, and to unpack what features of the data require what features of loop extrusion. For example the authors may do better to comment that, the fraction of time spent in the proximal state, the frequency it switches into that state, the duration of that state, and most importantly how and why those things change when CTCF is removed, cohesin is removed, or both are removed, is parsimoniously explained by loop extrusion. And equally important, is here measured for chromatin at the scale of the E-P interaction for the first time. With a note added in proof at the end that observations at the larger 500 kb distances, performed in parallel, are consistent with these data.

What the authors currently have written sounds off-puttingly grandiose, “all the dynamic microscopy data” and “a simple model of loop extrusion with no parameter optimization”. Yet, it is actually, as a stand alone sentence, quite vacuous. One could say the same of the very first movies of single lacO sites. After all, there is nothing in “all that dynamic data” that is **inconsistent** with loop extrusion ((not very much dynamic data at all, but it is still all the dynamic data”). Sweeping conclusion of this nature have a tendency to sink in, and I think this one risks too much confusion.

I think if the authors were clearer about what they are not measuring and not concluding, it may also improve the work. For example, I see nothing in the data that rules out that the kinetics of chromatin motion are only those of the Rouse polymer – a diffusive, energy minimizing process. There is no smoking gun that this is some energy burning, clearly motor driven, entropy reducing process to explain the data. The authors don't even show other non-extrusion models and how they would work. But I think they aren't actually making that claim (though I fear others may attribute such a claim given the wording of the current work). The time switching in and out of the proximal state, the duration in that state, these are real time frequency data that are useful whether or not they emerge as a result of loop extrusion. It is nice to know, but not very surprising given the limited spatial temporal resolution of the data, that these measurements are also consistent with loop extrusion. After all, the 150 kb separated loop anchors, which, according to the model, undergo multiple rounds of cohesin loading and unloading during the image window, are nearly indistinguishable, in the distribution of the 3D distances, from the fixed point dynamics, where the separation is driven by none of these extrusion dynamics. When the data has a substantial amount of measurement uncertainty relative to its dynamic range, and the simulation gets a similar amount of uncertainty built into it to capture the data it is not so surprising, nor conclusive, to say, look the data and model agree.

We thank the Reviewer for inviting us to reformulate inaccurate statements related to the role of loop extrusion in our interpretation of experimental data. The Reviewer is right: at the extrusion speeds corresponding to the maximal agreement between model and experiments, what dominates is not the extrusion process *per se* but rather the presence of physical constraints imposed on the polymer by the loop extruder (please see our response to the Reviewer's point 5 below). We therefore took their suggestion to heart and 1) removed the problematic sentence in the introduction the Reviewer alludes to (see new introduction, page 2 in the revised manuscript); 2) rephrased all claims that loop extrusion "explains" the ensemble of our observations, notably at page 9 where we replaced the previous strong claim as suggested by the Reviewer; see also new section heading page 4 and new text at page 5 in response to the Reviewer's point 5 below; 3) emphasized in the Discussion that these are the first estimates of contact durations and frequencies at the length scale of enhancer-promoter communication (page 10); 4) reinforced the link with the Hansen lab's results through new simulations that predict the duration and frequency of CTCF-mediated interactions a function of genomic distance (**Suppl. Fig. S8** and new text, last paragraph page 10). These simulations indeed account well for their observation that longer-range CTCF-CTCF interactions last around the same time while becoming substantially rarer. We hope that the Reviewer agrees that this gives a fairer and more balanced account of our model-based interpretation of experimental results.

4) "Contact"

- Some of the field is going to get an over-interpreted idea that this work has measured the molecular scale interaction. I did appreciate the authors attempt to set this up as the "proximal state" before introducing contact as a short-hand, maybe it should not be used. On the other hand I'd much rather see these 140 nm distances interpreted as consistent with contact than as evidence of action at a distance, so maybe it should stay.

We agree with the concern expressed by the Reviewer. We have indeed long debated whether or not we should identify the proximal state as 'contacts' in the manuscript, and how to avoid that this is misinterpreted by the field. We nevertheless think that the careful and hopefully clear definition we provide should prevent the reader from being misled. We also hope that the Reviewer agrees that the revised discussion sentence on experimental resolution and the potential detection of even shorterlived proximity events should also help in this sense (page 10, see point 2 above).

5) "Loop extrusion explains the decelerating effect of cohesin on chromosome dynamics"

- I found this assertion to be a bit too strong. I suspect the "deceleration" is actually just the consequence of having a many more short tethers distributed throughout the chromatin, so the fiber has less wiggle room and move viscosity. It may be better to write "loop extrusion is consistent with". It would be interesting and straightforward to test alternative explanations by simulating the effect of adding tethers. If tethers alone are sufficient it would be premature to say the (active process of) loop extrusion per se that explains the observed dynamics.

Motivated by the Reviewer's comment, we simulated polymer chains with varying numbers L of fixed harmonic links between pairs of beads chosen randomly and maintained fixed along the simulation. The pairs of beads were chosen from a power-law distribution of their 'genomic' distance with exponent -1. For each value of L , we calculated the MSD and averaged over random realizations of the static links and the randomly chosen beads. *Reviewer Figure 1* below shows examples of MSD obtained with different values of L on a chain of 1000 beads. We found that by varying L we could obtain different dynamic regimes with exponents ranging from 0.2 to 1. Thus polymer chains with fixed random links can produce a wide range of behaviors, including the regime (~ 0.6) that we measured experimentally. It is also worth to note that what determines this complex subdiffusive behavior is not simply the abundance of links (i.e., the "viscosity" mentioned by the referee). In fact, simulating the same chain with links drawn from a short-range Gaussian distribution only produces Rouse-like subdiffusion (cf. cyan curve in the *Reviewer Figure 1*, $L=100$).

We thus agree with the Reviewer that the agreement of the active-extruder simulations with the experimental data alone is not a formal argument in favor of the loop-extrusion model. We have therefore now amended the section heading as suggested by the Reviewer and also modified the final sentence in the section into "Polymer simulations thus strongly support the notion that the observed decrease in chromosome mobility and lack of substantial effects from strong CTCF motifs is a macroscopic manifestation of the physical constraints imposed by cohesin in living cells".

While we are very much interested in better analyzing and describing the polymer models described in this paragraph in the near future, we trust that the Reviewer agrees that this requires a longer-term theoretical commitment, and that the validity and significance of our study do not rely on such additional developments.

Reviewer Figure 1. MSD calculated in Langevin simulations of a 1000-bead polymer with static random constraints. Adjacent beads interact with harmonic springs and have excluded volume. Purple, red and green curves correspond to simulations with a number L of harmonic constraints between beads chosen at random with distribution which is a power-law of the genomic distance with exponent -1 . The curves are averaged over independent realizations of the random constraints and on the identity of the tracked bead. The cyan curve corresponds to a simulation with $L=100$ constraints with Gaussian distribution (mean=10 stdev=3). The dashed lines display linear fit in the log-log plot and the corresponding slope is indicated

6) “The finding that CTCF does not impact global chromosome mobility is at odds with 3C-based findings that sites bound by CTCF form site-specific interactions with increased contact probabilities.” - I don’t really think this is a true/fair representation of the 3C (Hi-C) based findings – the observation was that CTCF increases the interaction probability with adjacent CTCF sites, no one claimed it increases or affects interactions among one CTCF with all other CTCFs genome wide, or affected contact probabilities across the chromosome uniformly. Indeed the Hi-C normalization of “matrix balancing” explicitly assumes the opposite – by assuming ‘equal visibility’. Here, the authors are introducing a very natural and very beautiful set of follow up experiments to their global analysis of random integration with a less than beautiful (confusing/misleading/simply false?) topic sentence.

Thanks for pointing this out. We have removed this sentence from the new version of the manuscript.

7) Interestingly, distances in the proximal state inferred by HMM largely overlapped with those detected on perfectly colocalizing signals in control experiments (149 vs. 130 nm on average, respectively) (Fig. 5B right panel, cf. Suppl. Fig. S4K). - This could be explained better – there is a really nice control here which deserves a word of explanation in the text, I would mention co-expression of tetR-tdTomato and TetR-eGFP.

We thank the Reviewer for their appreciation of our control experiments and agree that they might deserve a few more words of explanation. We have now updated the description of these experiments as follows (page 8): “Interestingly, distances in the proximal state inferred by HMM largely overlapped with those detected on perfectly colocalizing signals in control experiments where TetR-eGFP and TetR-tdTomato were coexpressed and bound to the same set of randomly inserted TetO arrays (149 vs. 130 nm on average, respectively)”. We have also expanded the actual passage in the manuscript where these experiments were originally introduced (page 8).

Technical problems

1) Code availability

The github repositories on the Gioreggti Lab github site, listed in code availability do not appear to be publically available or privately accessible for review, though some analysis scripts referenced in other parts of the paper at the zhanyinx github site are accessible.

We are sorry for the confusion. All the repositories have been made open and available.

2) Data availability

I could find no mention of the tracking data deposition, which appear to be the core of the paper, though data for many peripheral experiments were deposited. If the data are not deposited yet, I recommend the authors use a repository such as Zenodo. I believe some Nature family journals also require this. In the opinion of this reviewer, these data should be available at review.

All tracking data have now been uploaded on zenodo (<https://doi.org/10.5281/zenodo.6627715>) and the link to the repository is now provided in the text in the “Data availability” section.

3) Fitting approach / Fit accuracy

Spot fitting is a generally a well posed physics problem where the knowledge of the wavelength, pixel size, NA, camera gain, dark current, photon flux, etc. add information to make this fit more accurate. There is a deep literature on the theory and a deep literature of the tools. Unlike, say, calling cell borders or something from microscopy images, I don't believe this is a good problem for deep learning off of images alone and I worry about the generalizability of the training sets. I know the authors have published it all already (in a work that I thought took a rather shallow view of the field and the tools). I'm not too worried about it in the context of this work, I think the images of the two-color Tet experiment already make it clear other sources of noise dominate, including the non-point-source nature of the tag.

We thank the Reviewer for raising this important point and would like to take the opportunity to clarify our spot detection and localization procedure. This consists of 1) spot detection using DeepBlink (deep learning) and 2) spot localization using 3D Gaussian fitting. The rationale for using DeepBlink instead of alternative spot detection methods is that our field of view typically contains approximately 25 mESC nuclei. Despite the fact that our mESC lines are clonal for the insertion of TetR-tdTomato and LacI-GFP, background nuclear fluorescence intensities in each cell can vary substantially (e.g. as a function of the cell size during cell cycle progression). This poses a problem for conventional threshold-dependent algorithms which perform unevenly across cells with different background intensities (see *Reviewer Figure 2* below), and additionally require substantial manual tuning of detection parameters on an image-by-image basis. These limitations are overcome using DeepBlink as no manual parameter tuning is required and it can cope well with imbalanced images (*Figure 2* below). To further increase the reliability of spot detection, for each dataset we used models trained on manually labeled images from the same dataset.

Importantly, our spot localisation does not rely on deep learning but rather on 3D gaussian fitting in an ROI around each spot coordinate detected using DeepBlink. Combined with other noise sources such as residual chromatic aberrations (after computational correction, see point 4 below) and non-pointlike sources, spot localization error contributes to the ~130nm uncertainty.

We have now substantially rewritten the Methods section to make sure these important details come across more clearly (page 38).

Reviewer Figure 2. Example image (max. Intensity projection of a Z-stack) showing the spot detection performance of deepBlink vs TrackMate. The two rows correspond to two different color saturations of the same image to highlight spots within cells with different backgrounds. From left to right: bona fide spots (left, green arrows), spots detected by deepBlink (middle panel, red crosses) and spots detected by TrackMate (right panel, red crosses). TrackMate tends to detect false positive spots in cells with a high background, which are filtered out by deepBlink.

4) Chromatic correction

The only description I found was this: “TetraSpeck™ Microspheres, 0.1 μm beads (Thermo Fisher Scientific, T7279) were imaged to allow for correction of chromatic aberrations during image processing and analysis”, which I find inadequate as a description. Browsing the part of the github repository that was accessible it looks like the authors may be using an affine transform as the correction map. Maybe a second order polynomial might be better, it would at least be nice to know what was done, and how accurately that corrected the bead data (a plot would be awesome, but a number would be an improvement). Are the beads just on the coverglass, or are they distributed in 3D space? The chromatic aberrations typically have a significant dependence on the axial (z) position, which affects the correction in all dimensions, though this is not always taken into account. I think the field would improve if it were more standard practice to report these corrections at any rate. Though given the other sources of measurement error, even if it were possible that the aberration correction could be improved I do not think it would have a dramatic effect on the results. The chromatic aberration should be constant for a fixed position in the field of view, and from sup Fig 4, it looks like the offset changes.

We apologize for the lack of clarity regarding the correction of chromatic aberration. This is a key technical point indeed, and we are grateful to the Reviewer for prompting us to revise the corresponding Methods section (‘Chromatic aberration correction of dual color data’) and include a better account of this in the main text as well (**Suppl. Fig. S4I**). In brief: we took 3D image stacks of beads adsorbed on MatTek dishes and imaged in 1xPBS at the beginning of every imaging session and used them to correct the corresponding set of movies. After detecting signals from single beads using deepBlink and determining their 3D location by Gaussian fitting, we first identified spots that are shared across channels by solving the linear assignment problem (LAP) using the euclidean distance between spots. We then used the common set of bead signals to compute a 3D roto-translation that we finally applied to xyz positions. This procedure corrects for x, y and z aberrations simultaneously (see *Reviewer Figure 3A* below). The same transformations accurately corrected chromatic aberrations in actual experiments in double-labeled mESC (“Control TetO” in **Suppl. Fig. 4J,K**), with the exception of a small residual systematic shift (~40 nm) along the z axis (see “TetO-LacO” case in *Reviewer Figure 3B* below and new **Suppl. Fig. S4I**), which is likely due to 3D image anisotropies that cannot be measured using “2D” bead images. This shift is however substantially smaller than the ~130nm uncertainty measured on overlapping TetR-GFP and -tdTomato signals and only leads to ~15% change on average pairwise distances in (see *Reviewer Figure 3C* below). In principle it would be possible to remove this small systematic shift, either by inferring a transformation from 3D bead samples (e.g. by incorporating beads in a gel or allowing them to adsorb to fixed cell samples), or alternatively by correcting single z coordinates of one of the two channels *a posteriori* (see for example Giorgetti*, Lajoie*, Carter*, Attia* et al., Nature 2016). The first strategy would require to re-image every sample in this manuscript and we hope the Reviewer agrees that this would exceed the scope of their request. We have thus applied the second method and systematically subtracted the ~40-nm offset from individual z coordinates of the tdTomato channel. This resulted in only minor quantitative differences in the actual key takeaway numbers without altering any of the conclusions of the paper (*Reviewer Figure 3D* below). While this procedure could be justified by the expectation that cell nuclei and chromosomes are randomly oriented and thus distance distributions should be centered around

zero in all three directions, it also introduces an element of arbitrariness as it assumes that all signal pairs are corrected by the same offset no matter where they are in the field of view. We thus hope that the Reviewer agrees that the original version of the correction should be retained, especially considering that we have added a new passage in the revised Methods section where we explain the effects of an additional *a posteriori* correction of the systematic z shift.

Reviewer Figure 3: Chromatic aberration correction using beads. A) Effect of correction on beads images. Uncorrected images (left panel) have biased (average distance different from zero) distances compared to corrected images (right panel). B) Distribution of pairwise distance for the beads, “control TetO” cell line (0kb), actual “TetO-LacO” cell line

before and after chromatic aberration correction. C) Effect of the residual error on z on the average pairwise distance across the lines. D) effect of correcting for the residual error along z on fraction of time spent in the proximal state (left panel) and duration of proximal state (right panel)

5) Time scale conversions

- “in order to convert the timescales between experiments and simulations, we measured the time need for a bead (and 8 kb TetO array) to move for its own size ($95\text{nm} - 15\text{nm} \times \sqrt{40}$ = estimated size of a nucleosome \times estimated number of nucleosomes in each 8kb segment, assuming that chromatin inside each 8 kb behave as an ideal chain; this is in line with the previous estimations for a 3kb segment [15])
- Relation between simulation time scales and data: The authors have made a physicists reasonable back-of-the-envelope estimate based on some measured properties, but left out the corresponding back-of-the-envelope estimate of uncertainty. The problem of accurately and convincingly determining these time scale conversions is a project unto-itself, well beyond the scope of this paper, and certainly beyond the scope of the data presented here (I suspect the authors are themselves well aware of the limitation of their estimate, but to be concrete I will describe several issues below). - An ideal chain of multiple segments and a bead don't behave the same. To model 8kb as 95 nm bead has a very different behavior than modeling it as a flexible chain. The high resolution FISH studies that the authors haven't cited (see comments about the intro) include measurements of the flexible configurations adopted by chromatin of this length, which sometimes has end to end distances of <50 nm and sometimes >300 nm as expected for a flexible chain.
- The movement measurements, given the relatively large time step (20s), the relatively large motion per time step (100 nm), the small total scales involved, and the substantial measurement uncertainty (0-300 nm spread for the repeat label), have substantial error bars.
- These are coarse grained langevin simulations. Given the lack of accurate measurements of the forces and energies involved and the large uncertainties on the measurements that are made, I am unconvinced the simulations “conversion to real time” is accurate within an order of magnitude or more.
- I think the whole field could benefit from a more careful treatment of the numbers. It is sloppiness like this that also leads the field to conclusion like “the enhancer and promoter are >200 nm apart, the cohesin ring is <40 nm, so it is impossible that these sequences are ever embraced in a cohesin ring”. The language here should be softened, some acknowledgment of “estimation” and “approximation” should be given, and the take home impression should not be one of precision and cross-validation for an estimate made with so many problematic issues along the way. Such uncertainty will have little impact on the key conclusions of the effects of cohesin and ctcf on the frequencies/durations of switching in and out of the proximity state.

We thank the Reviewer for raising this important point. We have modified the passage where timescale conversion was introduced as follows: “Simulation steps were approximated to real time units by matching the time needed for a monomer to move by its own diameter with the time required by the TetO array to move by its estimated mean physical size (see Methods)”. We have also modified

the Methods section ('Conversion of simulation timestep to real time' subheading) to accommodate a lengthier discussion of the approximations that we have implicitly made in the conversion.

Issues with figures

1) Figure 2B

This systematic survey over parameters is nice. However, I expect to see smooth changes in the contact maps as a function of parameters like extrusion density, but I see little correspondence – indeed, I first thought these “contact maps” came from individual polymer examples. If the authors were to show population contact frequencies it would give a clear picture of the effects of the parameters on contact. The suggestion that these simulations are significantly undersampled (or maybe averaged correlated structures from the same simulation) has me concerned for the interpretation of the other results in this analysis as well.

We regret that this issue might arise from the fact that contact maps in the original version of Fig. 2B reported a very zoomed-out view of contacts probabilities across the equivalent of 8 Mb, thus contributing to an impression of sparseness. The new version of this panel showing 2-Mb views should give a better visual account of how average looping patterns vary across the parameter space. Although we hope that the Reviewer agrees that these changes are smoother than they seemed in the previous version, we would nevertheless draw their attention to the fact that parameter sets are separated by a factor 10 in each direction and can thus be expected to lead to relatively different contact patterns.

We do not believe that simulations are undersampled. First, for each parameter set, contact maps are calculated by averaging over 10 independent simulations, each with a different initial conformation. Second, within each simulation run, conformations are recorded every 10^4 steps and the run lasts for 10^8 simulation steps (approx. 13 hours in real time under the approximations used to map computational time onto real time described in the Methods section). Conformations within a run become completely uncorrelated between 10^6 and 10^7 simulation steps (see *Reviewer Figure 4* below). We hope that the Reviewer agrees that these conditions are in line with state-of-the-art criteria in the field.

Reviewer Figure 4. A) Similarity between contact maps of single conformations within individual simulations used to calculate average contact maps for Fig. 2B. To estimate when conformations become uncorrelated, we measured the Spearman correlation of distance-corrected contact maps from single conformations inside a single simulation and plotted it as a function of the simulation time that separates each pair of conformations (blue line: average correlation, error bars: standard deviation). This was compared to baseline correlation levels measured over conformations drawn from independent simulations, which are uncorrelated by default (solid red line – mean baseline correlation, dashed red lines: +/-standard deviation). Conformations of the entire chain of 1'000 beads (8 Mb) become completely uncorrelated around 10^7 simulation steps. B) Same as in (A) but showing correlations between conformations of the 100 beads (800 kb) in the middle of the chain.

2) Supp Fig S4L-M – is labeled “rMSD (μm^2)”. This is obviously a typo, rMSD being units of distance and MSD being distance squared. I believe the authors meant to write MSD, based on the actual unit measurements.

Also, the green lines in S4L and S4M to me both read as being the 150 cis +Rad21 +CTCF case, just juxtaposing that data against different comparison sets. But why then do the data look different? And why no error-bars in S4L? why do these green data lines have different max lengths for example?

We apologize for the confusion. In the previous version of the manuscript, the abbreviation ‘rMSD’ stood for ‘radial mean squared displacement’, i.e. the MSD of the distance between two signals (as defined at page 4 in the original manuscript). This is effectively an MSD and has units of μm^2 . We realize however that this abbreviation could be confused with ‘root mean squared displacement’ and have replaced all instances of ‘rMSD’ in the text and plots with ‘radial MSD’.

We further apologize for the incorrect labels in plots S4L and N in the first version of the manuscript. Indeed, the green lines according to the labels should show the same dataset. However, since the simulations at varying distances were performed in the absence of extrusion barriers only, the plot

S4L should show the comparison between the experimental data in the absence of the CTCF sites but in the presence of cohesin. We apologize for the incorrect comparison and the missing error bars in our previous submission. We have revised the plot to show the aforementioned comparison. We have further added the error bars missing from the previous Suppl. Fig. S4M. The revised plots can be found in **Suppl. Fig. S5A,B**.

Reviewer #2:

Remarks to the Author:

SUMMARY:

In this manuscript, authors used live imaging of genomic loci (labeled by TetO or LacI arrays) and physical modeling to dissect the role of Cohesin and CTCF in regulating chromosome dynamics. The main findings are 1) Cohesin loss increases the mobility of genomic loci globally with or without CTCF, consistent with the loop extrusion model prediction. 2) By imaging two loci within a neural TAD +/- a CTCF BE (boundary element), authors found that cohesin and CTCF acts together to constrain the radial mobility of chromatin 3) In addition, two distinct states were identified in CTCF BE(+) condition – a proximal state in which the 2 loci were less than 149nm apart close to the detection limit of the imaging and a distal state where 2 loci were separated by 288nm in average similar to the distance in the CTCF BE(-) condition. Using HHM to analyze the data in combination with physical modeling, they estimated that CTCF anchored loops last around 10mins in live cells. This reviewer really appreciates the interactive experimental and modeling approach taken by authors. The data quality is high. The experimental design/results are relatively straightforward. The main problem is that most results from the current study largely agree with previous loop exclusion simulation predictions or has been demonstrated (at least implied) previously (See detailed points below). Limited conceptual advances have been made. However, if authors are up for more challenges and would like to push their work further with the experimental suggestions below. I would be more than happy to review a revised manuscript.

We thank the Reviewer for their appreciation of our work. We would like to respectfully disagree with their impression that our results are trivial or of limited conceptual advance. First, we would like to draw the Reviewer's attention to the fact that previous loop extrusion simulations have not provided clear predictions on whether loop extrusion should increase or decrease chromosome dynamics. Of the two theoretical papers the Reviewer refers to in point 5 below, one (PMID: 26538024) simulates the effects of *static* loop anchors at telomere regions and predicts that they should slow down the motion of the polymer; the other (PMID: 29967174) predicts that dynamic loop extrusion with supposedly realistic parameters for interphase mammalian chromosomes should rather increase chromosome mobility (cf. Fig. 4G in PMID: 29967174). None of these predictions has been ever tested experimentally. Our work now 1) shows that *dynamic* loop extrusion should generally lead to a decrease in chromosome motion (**Fig. 2**) unless extrusion speeds are non-physiologically high (**Suppl. Fig. 6E,F**); 2) proves experimentally that under physiological conditions cohesin effectively slows down chromosome mobility; and 3) characterizes for the first time the timescale of chromosome interactions inside TADs and how they are modified by the presence of cohesin and CTCF. Our study

thus provides the first unifying framework to understand the role of cohesin and loop extrusion in determining the dynamics of interphase chromosomes in mammals.

We would also like to draw the Reviewer's attention to the fact that our experimental design is far from being straightforward, and rather results from a combination of state-of-the-art genome engineering methods and live-cell imaging approaches. This is the first time that a large number of genomic viewpoints could actually be imaged in live, which was enabled by the repurposed piggyBac transposition strategy described in Fig. 1. Insertion of two operator arrays in *cis* within a highly controlled genomic environment is in itself a daunting genome engineering task. Live-cell dual-color 3D imaging of genomic locations in multiple conditions on a timescale of hours at high temporal resolution in mouse embryonic stem cells is at the cutting edge of quantitative microscopy approaches. We have nevertheless taken the Reviewer's comments and suggestions to heart and now provide new experiments and analyses that address some of their experimental requests, as described below in our point-to-point responses to their comments.

Main points:

1) Cohesin loss eliminates TAD and loops but enhances compartmentalization. The loop extrusion model cannot explain the enhanced compartmentalization, suggesting additional mechanism must be at play. Regulatory activities within A and B compartments are fundamentally different. Authors here non-discriminatively analyzed loci within both active and inactive domains. One way to really advance the fundamental understanding of the field is to study how chromatin dynamics in A/B compartment differs in both WT and Cohesin loss conditions. What if Cohesin loss have opposite effect on chromatin dynamics within A versus B? This is especially interesting when you consider recent emerging results showing that transcription factors with low complexity domains can form dynamic protein hubs in active domains. How would these type of regulations affect the looping or chromatin dynamics?

We sympathize with the Reviewer's opinion that these are very interesting questions, and that the experiments they suggest could shed some light into the interplay between loop extrusion and compartmentalisation. We however hope that they agree that addressing these questions would require new experiments that go far beyond the scope of our paper. Although we know the genomic coordinates of single TetO insertions (at least in the clonal cell lines where we performed 4C, see Suppl. Fig. 1H), these cannot be distinguished under the microscope in living cells because all TetO insertions are detected using the same fluorescent protein and are thus optically undistinguishable. The only way to identify single genomic locations would be to fix cells after live-cell microscopy, perform DNA FISH with locus-specific probes on the same samples, acquire FISH images, followed by image registration and identification of single DNA FISH signals with live-cell tracks. This presents daunting technical challenges, notably 1) Imaging of DNA FISH and live-cell imaging should be performed on two different microscopes; 2) only three or four sequences could be detected in a single round of DNA FISH, which would severely limit the statistics of such experiments unless 3) a multiplexed sequential DNA FISH approach is implemented. While we are very much committed to developing such methods in the near future, we trust that the Reviewer agrees that this requires a

long-term experimental commitment and that the validity and significance of our study do not rely on such additional developments.

We have nonetheless addressed the very much related questions of whether the transcriptional state of a locus alters its mobility and looping properties, and whether depletion of cohesin modifies such properties using an alternative experimental approach, which is described in our response to the Reviewer's point 2 below.

2) Another point is whether and how chromatin and looping dynamics is regulated by transcriptional activities. This point can be first tested by using chemical inhibitors (such as JQ1 and alpha amanitin). It would be even better if authors can target and track genes when they are in active and inactive states (MS2) and study whether Cohesin loss has different effect on chromatin mobility.

This is indeed another interesting point. The effect of transcription has in fact already been addressed in previous studies (Germier et al. PMID: 28978433 and Nozaki et al. PMID: 28712725) where small molecule inhibitors were used to study the global effects of transcription inhibition at the initiation as well as the elongation stage on chromosome dynamics. In both cases it was shown that transcription mildly constrains chromosome motion. Stimulated by the Reviewer's comment, we nevertheless now provide new experiments designed to move beyond previous studies and determine whether cohesin depletion differentially affects chromosome motion for loci undergoing active transcription. To this aim we integrated strong constitutive promoters (the well-characterized mouse 3-phosphoglycerate kinase (PGK) promoter that leads to stable expression over time and during differentiation as compared to commonly used viral promoters, see Adra et al. PMID: 3440520, Herbst et al. PMID: 22434137) next to the LacO and TetO operator arrays we used in dual-color looping experiments. We then measured the relative dynamics of the two loci in the presence or absence of RAD21 and compared them to those of cells without promoters. In line with previous studies, we found that the radial dynamics of the two loci was slightly slower in the presence of the promoters (see **Suppl. Fig. S5C**). Interestingly however, chromosome dynamics was affected to a similar extent by the loss of cohesin, which led in both cases to a slight increase in MSD overall. Pairwise distance distributions and interaction durations and frequencies of operator arrays flanked by promoters were indistinguishable from those without promoters, both in the presence and absence of cohesin (see **Suppl. Fig. S5E-I**). These new experiments thus suggest that cohesin affects pairwise interactions of transcribing and nontranscribing loci to the same extent.

3) Authors always insist on is that their chromatin dynamics measurement reflects long-range transcriptional regulation. One main point from results in the past few years is that Cohesin-mediated genome organization and transcriptional regulation could be decoupled. Cohesin or CTCF loss only has minor effect on gene expression (PMID: 28985562). Conversely, transcriptional inhibition does not appear to significantly impact trans-chromosomal or long-range cis-chromosomal interactions (MID: 32616013). Thus, it is not appropriate for authors to equate chromatin looping mediated by Cohesin/CTCF to long-range transcription regulation. The loops could be more serving a structural

role rather than a functional one. It is worth noting that high-resolution Micro-C experiment even suggests that enhancer-promoter interactions persist in the absence of Cohesin. It is important for authors to make the distinction in the text.

We disagree with the Reviewer's opinion that chromosome interactions are irrelevant for transcriptional regulation. It is true that global depletion of cohesin or CTCF has been shown to have moderate effects on gene expression, but there is wide consensus in the community that these results should be nuanced due to the fact that global impairment of trans-acting factors results by definition also in indirect effects; and that mRNA readout does not faithfully account for transcriptional changes in acute depletion experiments (see for example Balasubramani et al, PMID: 20969595). Recent experiments within controlled genomic regions have indeed shown that contact probabilities have a major impact on enhancer-promoter communication (e.g. Huang et al PMID: 34002095, Zuin et al PMID: 35418676). We however agree to tone down sentences that could be perceived as suggesting that our study directly proves a direct causal link between dynamic chromosome contacts and transcriptional control (see last sentence in the Abstract, last sentence in the introduction, page 2). We now also emphasize that we measure interactions at the *genomic length scale* of enhancer-promoter communication (as opposed to between enhancers and promoters directly).

4) Does the size of TetO and LacO repeats affect chromatin dynamics? Authors should test this by decreasing and increasing the number of repeats in the array?

This is indeed an interesting albeit technically demanding question which we could only address by cloning TetO and LacO arrays of different sizes, inserting them in mESC through piggyBac transposition and CRISPR-Cas9 assisted knock-in, deriving clonal cell lines, and performing live-cell microscopy followed by image analysis. We trust that the Reviewer agrees that this would require a long-term experimental commitment that would go beyond the scope of this study. Reassuringly, TetO and LacO arrays of different (larger) size have been previously used for live-imaging in mESC (e.g. Masui et al. PMID: 21529716: ~ 250 TetO sites; Khanna et al, PMID: 31235807: 360 TetO sites), and shown to undergo subdiffusive motion with generalized diffusion coefficients in the same range as those we measured here using 140 TetO sites per array ($\sim 0.01 \mu\text{m}^2\text{s}^{-\alpha}$ with $\alpha \sim 0.6$). It is also worth noticing that in our experience arrays with ~100 sites represent an optimal compromise between fluorescence signal-to-noise ratios in mESC and ease of insertion through piggyBac and CRISPRCas9 editing.

5) Previous loop extrusion simulation suggests that chromatin dynamics would increase due to the lack of constraint from Cohesin (PMID: 26538024; PMID: 29967174). Authors should discuss these papers.

We thank the Reviewer for this suggestion. We would however like to point out that simulations in PMID: 26538024 predicted that static loops (i.e. not loop *extrusion*) should slow down the motion of a

polymer; but in PMID: 29967174 the prediction is instead that loop extrusion should rather increase chromosome mobility (cf. Fig. 4G in PMID: 29967174). We thus now cite these two papers in the introduction (page 2) as examples that theoretical analysis has suggested that loop extrusion could in principle lead to both an increase and a decrease in chromosome motion.

6) Previous simulation and experimental results suggest that the lifetime of chromatin loops are in ~10s minute range (PMID: 28467304). Authors should discuss the paper.

We apologize for omitting this important citation concerning the estimated lifetime of cohesin on DNA. We did in fact cite Hansen et al. in the introduction but now also cite Hansen et al. PMID: 28467304 at page 9 (ref. 20) when comparing residence times extracted from our simulations with existing experimental estimations.

7) The drift correction performed in the analysis (Figure 1D) is less than optimal. Specifically, one premise for using the ensemble average to correct is that the number of particles detected is the same at every time point. As soon as there are missing detections, particle trajectories will experience jitters and it fails. It is important for authors to show that they can consistently detect the same number of loci across time points.

We thank the Reviewer for raising this point. We realize that the nuclear motion correction was not clearly explained in the previous version of the manuscript and have now expanded the corresponding Methods section ('Tracking and mean square displacement analysis of dual color data'). We definitely do share the reviewer's concern that missing spots can be problematic towards the estimation of the transformation used to correct for cell motion. That is why we implemented a method that ensures that exactly the same spots across two consecutive time points are used to estimate the transformation. This relies on spot matching (registration) based on the solution of the linear assignment problem (LAP) using the euclidean distance between spots as a measure of distance. This procedure does not use the same number of spots at every time point to infer each roto-translation, and the Reviewer is right that the accuracy of the motion correction depends on the number of spots. However, the method we implemented is robust to substantial variation in the number of spots and we hope that the following argument convinces the Reviewer that this is indeed the case. We generated sets of in silico "cells" with 20 synthetic particle trajectories each (that we generated using TrackLib (Gabriele et al. PMID: 35420890) undergoing subdiffusion dynamics with MSDs in the range of those we determined experimentally ($\sim 0.01 \mu\text{m}^2\text{s}^{-\alpha}$ with $\alpha \sim 0.5$). We then added a random cell drift and rotation to each set of trajectories (i.e. to each "cell") using gaussian distributions of drift step (in μm) and rotation angles (in radians) with average 0 and standard deviation 0.08. We finally downsampled the number of trajectories using the experimental distribution of the number of matched (registered) points across experimental trajectories, which is centered around 6 and never drops below 4 (see *Reviewer Figure 5* below, left panel). We then used the downsampled trajectories to compute 3D roto-translations and verified that this removes most of the contribution of the "cell" motion (right panel; 'gt' = ground truth). This suggests that our motion correction procedure

is robust to fluctuations in the number of spots used to infer the roto-translation, at least in the range we observed in our experimental data.

Reviewer Figure 5: Test of the motion correction method. Left: Distribution of the numbers of matched (registered) points per cell across consecutive time frames used to correct for nuclear motion in experimental live-cell movies. Right: Examples of motion correction performed in two distinct “cells” (Cell 1 and Cell 2). The average cell motion correction across all simulated “cells” is shown on the far right. Solid lines: average MSD; shaded areas: +/- standard deviation across simulated trajectories.

Reviewer #3:

Remarks to the Author:

The manuscript of Mach et al. investigates the dynamics of loop extrusion using two-color time-lapse microscopy and computer simulations in a well-suited model system (mES cells). The study is timely and is highly interesting to a wide community interested in understanding chromosome structure, chromosome dynamics, and transcriptional regulation in eukaryotes. The study shows that: (1) cohesin reduces the mobility of genomic loci, (2) this reduction is consistent with loop extrusion, (3) the action of loop-extrusion and of CTCF can modulate dynamic chromatin interactions within TADs, (4) presence of CTCF and cohesin increases the frequency CTCF-CTCF encounters, and their duration, (5) CTCF-CTCF loops within TADs can be quite stable (10-15') and occur frequently, at least when these loci are close in genomic distance (150kb).

The experiments are well described and analyzed, and the conclusions are well supported by data. However, I highlight a number of issues that should be addressed. Major issues:

- The authors talk about 'deceleration' when referring to changes in dynamics in Figs. 1-2. I am not sure why they speak about deceleration when this term can be understood as a (negative) change in the derivative of the velocity. I don't think the authors measure acceleration in their figures.

We apologize for the confusion. We have reworded these passages and systematically replaced 'deceleration' with 'slowing down' or 'reduced chromosome dynamics'.

- are the locations of the barriers used in the simulations (Fig. 2) the same as those cloned and characterized in Fig. 1?

We thank the Reviewer for giving us the opportunity to clarify this point. The genomic distribution and orientation of barriers that mimic *endogenous* CTCF sites have been sampled from a real 10-Mb genomic region on chromosome 15 as described in the main text (page 4). On top of these 'endogenous' barriers we however inserted *additional* barriers to mimic the effect of randomly inserted 3xCTCF-TetO arrays. In our experiments, these occur at a frequency of 1-2 per chromosome and are thus separated by tens of Mb. To make sure we would overemphasize and thus reveal any possible effects mediated by such rare additional CTCF sites in polymer simulations, we decided to introduce *additional loop extrusion barriers* separated by a distance (800 kb) that is smaller than the experimental distance between 3xCTCF-TetO arrays (but still much larger than the correlation length along the chain, which is of ~25-30 beads corresponding to 200-240 kb). We hope that the additions to the main text (page 4) give a clearer account of this strategy.

- how were chromatic aberrations corrected? why are they so high provided that the signal-to-noise ratio from their images appears to be enough to reach nanometer localization precision? what limited further improvement? Is this limitation coming from spot detection? If it was the case, then the authors could use other localization methods to reduce the uncertainty in the measurement of radial distances, and thus detect more effectively CTCF loops, which in Fig. 6 are shown to occur less frequently and to be more short-lived than 'proximity events'.

We thank the Reviewer for raising this point, which was also a reason of concern for Reviewer #1 (see their point 4 above). We realize that the correction of chromatic aberration and estimation of experimental uncertainty on 3D distances were not sufficiently explained in the previous version of the manuscript. This is indeed a key technical point, and we have now revised the corresponding Methods section ('Chromatic aberration correction of dual color data') and included a better account of this in the main text as well (**Suppl. Fig. S4I**). In brief: we took 3D image stacks of beads adsorbed on MatTek dishes, imaged them in 1xPBS at the beginning of every imaging session and used them to correct the corresponding set of movies. After detecting signals from single beads using deepBlink and determining their 3D location by Gaussian fitting, we first identified spots that are shared across channels by solving the linear assignment problem (LAP) using the euclidean distance between spots. We then used the common set of bead signals to compute a 3D roto-translation that we finally applied to xyz positions. This procedure corrects for x, y and z aberrations simultaneously (see *Reviewer Figure 6A* below). The same transformations accurately corrected chromatic aberrations in actual experiments in mESC where TetR-GFP and tdTomato were allowed to bind to the same randomly inserted TetO arrays ("Control TetO" line in **Suppl. Fig. 4J,K**). In these control experiments, the measured 3D distances between perfectly overlapping GFP and tdTomato signals were centered around 130 nm after beads-based correction of chromatic aberrations (see **Suppl. Fig. S4I,L**). This

is our very stringent, but we believe intellectually honest definition of experimental uncertainty on 3D distances. It arises from residual chromatic aberrations in the z direction that cannot be corrected by our procedure (which uses 2D bead images; see our response to Reviewer #1's point 4 above), as well as contributions from the non point-like nature of GFP and tdTomato signals arising from TetO and LacO arrays, which introduces error in 3D Gaussian fitting. We would however like to draw the Reviewer's attention to the fact that our 130-nm 3D uncertainty is in the very same range (and indeed on the low side) of those reported in recent landmark studies using dual-color live-cell imaging to study chromosome folding dynamics (Chen et al, PMID: 30038397, Gabriele et al PMID: 35420890) and is thus representative of state-of-the-art methods in the field.

Reviewer Figure 6: Chromatic aberration correction using beads. A) Effect of correction on beads images. Uncorrected images (left panel) have biased (average distance different from zero) distances compared to corrected images (right panel). B) Distribution of pairwise distance for the beads, “Control TetO” cell line (0kb), actual “TetO-LacO” cell line before and after chromatic aberration correction.

- In Fig. 4F the authors use variance to characterize the distance explored by each trajectory, but wouldn't it be more interesting to perform jump analysis ? remove baseline to reduce contribution of slow-rigid body motion of the two spots?

We thank the Reviewer for this suggestion. We now provide jump analysis in the new **Fig. 4G**. This analysis confirms that constraints imposed by cohesin and convergent CTCF sites reduce the radial movement of the two operator arrays. We would like to note however that since in the double-color experiments we measure radial distances between the TetO and LacO sites, the rigid-body baseline motion of the two spots (e.g. arising from the motion of the chromosomal territory, or the nucleus) is by definition already subtracted in these data.

- why do you expect only two states in the HMM model used in Fig. 5 to determine proximal states? Presumably there are a continuous of polymer configurations? In Fig. 6, the authors show that the HMM model can be used to detect proximal states from simulations, but that these do not always correspond to looped states. Can the authors train the HMM model on the simulations to predict also looped states in their experimental data?

The reason beyond the choice of a 2-state HMM is that we expect any polymer with site-specific interactions between two sites (such as those effectively mediated by cohesin stalling at convergent CTCF motifs) to be at least in first approximation described as a two-state system, corresponding to two free energy minima as a function of the mutual distance between the two sites: one (bound) determined by the interaction energy at the anchor sites and the other (unbound) stabilized by entropy. If the two free-energy minima are deep and well separated, the distribution of distances between converging CTCF sites is apparently bimodal (see example in the new **Suppl. Fig. S6A**). It should however be noted that in many cases the peaks corresponding to the two states could be highly overlapping and they could be hard to distinguish. This is the case in which HMMs are highly useful. Following the Reviewer's suggestion, we have trained the HMM on the simulation data and used it on the experimental data. The results (see *Reviewer Figure 7* below, for a single parameter set corresponding to the previous best agreement for illustration) of course change quantitatively, although slightly. However, this approach is problematic because using the exact same parameters in polymer models each with different extrusion parameters and thus different effective physical sizes, *and* on experimental data at the same time inevitably introduces errors which are difficult to estimate. Unless the Reviewer strongly disagrees, we would argue that it is preferable to keep the original 'selfconsistent' HMM approach where emission rates are determined separately for experimental and simulation data.

Reviewer Figure 7: Applying an HMM model trained on simulation data to predict the CTCF looped state in the experimental data. Left panel: Prediction of CTCF looped state based on the model compared to proximal state defined on experimental data directly. Right Panel: Contact duration changes slightly when comparing proximal and looped states.

- Is the distribution of distances bimodal in Fig. 5A to justify use of 2 states in the HMM model?

As discussed in response to the previous question, the requisite for using a 2-state HMM is not that the distributions of distances is bimodal but rather that we expect that the system has two free energy minima. This does not necessarily materialize as a clear bimodal distribution of distances with two distinguishable peaks. We hope that the Reviewer agrees that these assumptions are now better explained in the main text (page 8 and 9).

- why does the rate of formation of the proximal state depends on the presence of 3xCTCF sites? (Fig. 5E) I would have expected only the duration of the proximal state to be affected. Evidence for on-sided looping? Can this be investigated computationally?

The Reviewer is right, this is a counterintuitive finding which results from two effects. The first is that in the absence of 3xCTCF sites, two specific loci can become the base of a loop only when the loop is initiated right in the middle between the two loci, otherwise they never do (see *Reviewer Figure 8A* below). When convergent CTCF sites are present, the two locations can be brought together at a loop base no matter where the loop is initiated between the two locations (*Reviewer Figure 8B*). This results in an effective increase in contact frequency and indeed is equivalent to transient one-sided extrusion while one end of the loop stalls at one of the two barriers. The second effect that contributes to an increased frequency of transition into the proximal state is that due to the slow radial subdiffusion we observe in the experiments, the simulations and the theory (see **Suppl. Fig. S4A-B, S5A-C** and Supplementary Information). This results in distances remaining close to the proximal state range immediately after the resolution of a contact and 'recapture' in the proximal state is more frequent (either due to loop extrusion or baseline polymer motion).

Reviewer Figure 8: A) Two regions that are not bound by convergent CTCF sites can never be found together at the base of an extruded loop, unless the loop is initiated right in the middle of their genomic distance. B) In the presence of convergent CTCF sites, loops initiated no matter where inside the intervening region can bring the two regions together and effective unidirectional extrusion can occur as the loop is stalled at one end.

- Why does removal of barriers or extruders have a similar effect on the duration of proximal states? Does this indicate a basal looping ability of CTCF independent of cohesin?

We believe that this is an effect of the small residual amount of RAD21 upon dTag13 treatment (which we estimate to be <5% based on Western Blot quantification, see **Suppl. Fig. S4D**). In these conditions, removal of 3xCTCF sites is expected to lead to a decrease in contact duration that is similar to . In order to compare the simulation data to the experimental data set, we choose the data from the parameter set with a loading rate = 0.012 min x Mb and an extruder residence time = 5.5 min as a "low cohesin" case (as explained in the Methods section). Compared to the best fit parameter set, this parameter set has a decreased loading rate (by 25-fold) whilst keeping the unloading rate constant (extruder residence time). This therefore mimicked our experimental set-up with overall lower cohesin densities per Mb (**Suppl. Fig. S7E**). Simulations without the loop extruder did not yield any proximity events, so that duration and rate of formation of the proximal state could not be calculated. We therefore argue that removal of the barriers or extruders has a similar effect due to the remaining cohesin/extruder in the case of the removal of the barriers/CTCF sites.

- How do frequency and duration of CTCF loops change with genomic distance? The results in the paper (residence time/ frequency of encounters) should depend on the distance between CTCF sites. While 150kb is not a bad choice, convergent CTCF sites can be found at shorter distances and at much longer distances (e.g. loops between TAD borders). It would be useful to be able to extrapolate these results. Otherwise there is a risk that a reader would interpret that 'in average' or 'in general' CTCF loops can be relatively long lived (10-15') and occur several times during a cell

cycle. These additional analysis would also be important as it would reflect the importance of how genomic distance can be used to modulate time-scales and frequencies of interaction.

We totally agree with the Reviewer and thank them for the suggestion. We now provide new simulations where two converging extrusion barriers (or generic beads) are located at a distance equivalent to 100, 150, 250 and 500 kb (**Suppl. Fig. S8A-B**). These simulations reveal, as expected, that the duration of the 'true' looped state (i.e. the CTCF-CTCF interaction mediated by cohesin) does not vary with increasing genomic distance between the CTCF sites; but that its frequency substantially decreases. Interestingly this is reflected by the HMM calls after adding experimental uncertainty with the minor exception that the duration of the proximal state slightly decreases too. This is a consequence of the fact that 'recapture' frequency in the proximal state is higher when the genomic distance is smaller due to more frequent non-CTCF/cohesin mediated collisions of the chromatin fiber.

Minor issues:

- What are the error bars in Figure 1I? what can be considered as statistically relevant change?

Variation in generalized diffusion coefficients (D) and scaling exponents (α) across single trajectories is shown in **Suppl. Fig. S2E** in the manuscript. Student t-test (two-sided) on D and α distributions shows that changes in α are significant when depleting RAD21, while changes in D are significant when depleting either RAD21 or WAPL. Global depletion as well as removal of local CTCF sites do not affect α and D (also reported in **Suppl. Fig. S2E**). We now provide indications on statistical significance in the new version of **Fig. 1I**.

- Not clear what 'additional barriers' mean in zoom-in of Fig. 2A.

We apologize for the lack of clarity on this specific point and invite the Reviewer to refer to our response to their previous comment on barriers and their separation (page 20 in the rebuttal).

- What does "time needed for a monomer and TetO array to move their own sizes?" mean?

We apologize for the lack of clarity. As now hopefully better described in the main text (page 5) and in the Methods ('Conversion of simulation timestep to real time' subheading), we used computational and experimental MSD curves to compare the average time needed for a monomer in the simulation to travel by its own physical size with the average time needed for the xyz position of TetO signals to travel by the estimated average size of a 8-kb chromatin fragment (see also our response to Reviewer #1, point 5 above).

- Where is the definition of rMSD ?

We apologize for the possible confusion around the definition of “rMSD”. In the previous version of the manuscript, “rMSD” was meant as an abbreviation of “radial mean squared displacement” as explained in **Fig. 1D**, i.e. the MSD of the distance between two signals (as defined at page 4 in the original manuscript). We realize that this abbreviation can be easily confused with “root mean squared displacement” and have replaced all instances of “rMSD” in the text and plots with “radial MSD”.

Decision Letter, first revision:

Dear Luca,

Thank you for submitting your revised manuscript "Cohesin and CTCF control the dynamics of chromosome folding" (NG-A59831R). It has now been seen by the original referees and their comments are below. The reviewers find that the paper has improved in revision, and therefore we'll be happy in principle to publish it in Nature Genetics, pending minor revisions to comply with our editorial and formatting guidelines.

Since the current version of your manuscript is in a PDF format, please email us (CC: natgen@us.nature.com) a copy of the file in an editable format (Microsoft Word) - we cannot proceed with PDFs at this stage.

We will then be performing detailed checks on your paper and will send you a checklist detailing our editorial and formatting requirements soon. Please do not upload the final materials and make any revisions until you receive this additional information from us.

Thank you again for your interest in Nature Genetics. Please do not hesitate to contact me if you have any questions.

Congratulations!

Sincerely,

Tiago

Tiago Faial, PhD
Senior Editor
Nature Genetics
<https://orcid.org/0000-0003-0864-1200>

Reviewer #1 (Remarks to the Author):

The authors have addressed all of my concerns with care and precision. I commend them on the

excellent advances of this important study and recommend a speedy publication.
- Alistair Boettiger

Reviewer #2 (Remarks to the Author):

Authors have adequately addressed my previous concerns.

Reviewer #3 (Remarks to the Author):

The authors have very convincingly addressed all my comments. Congratulations on a beautiful work.

Author Rebuttal, first revision:

None

Final Decision Letter:

Dear Luca,

I am delighted to say that your manuscript entitled "Cohesin and CTCF control the dynamics of chromosome folding" has been accepted for publication in an upcoming issue of Nature Genetics.

Your paper will be published online after we receive your corrections and will appear in print in the next available issue. You can find out your date of online publication by contacting the Nature Press Office (press@nature.com) after sending your e-proof corrections. Now is the time to inform your Public Relations or Press Office about your paper, as they might be interested in promoting its publication. This will allow them time to prepare an accurate and satisfactory press release. Include your manuscript tracking number (NG-A59831R1) and the name of the journal, which they will need when they contact our Press Office.

Please note that *Nature Genetics* is a Transformative Journal (TJ). Authors may publish their research with us through the traditional subscription access route or make their paper immediately open access through payment of an article-processing charge (APC). Authors will not be required to make a final decision about access to their article until it has been accepted. [Find out more about Transformative Journals](https://www.springernature.com/gp/open-research/transformative-journals)

Authors may need to take specific actions to achieve [compliance](https://www.springernature.com/gp/open-research/funding/policy-compliance-faqs) with funder and institutional open access mandates. If your research is supported by a funder that requires immediate open access (e.g. according to [Plan S principles](https://www.springernature.com/gp/open-research/plan-s-compliance)) then you should select the gold OA route, and we will direct you to the compliant route where possible. For authors selecting the subscription publication route, the journal's standard licensing terms will need to be accepted, including [self-archiving-and-license-to-publish](https://www.nature.com/nature-portfolio/editorial-policies/self-archiving-and-license-to-publish). Those licensing terms will supersede any other terms that the author or any third party may assert apply to any version of the manuscript.

Please note that Nature Portfolio offers an immediate open access option only for papers that were first submitted after 1 January, 2021.

You can now use a single sign-on for all your accounts, view the status of all your manuscript

submissions and reviews, access usage statistics for your published articles and download a record of your refereeing activity for the Nature journals.

Sincerely,

Tiago

Tiago Faial, PhD
Chief Editor
Nature Genetics
<https://orcid.org/0000-0003-0864-1200>

Click here if you would like to recommend Nature Genetics to your librarian
<http://www.nature.com/subscriptions/recommend.html#forms>